# Annexin A7 enhances TIA1 axonal trafficking to counteract pathological aggregation in neurons

Yu Feng [1,3], Tongshu Luan[1,3], Zhenda Zhang[1], Wei Wang[1], Yuanyuan Chu[1], Sijia Wan[1], Xiaorong Pan [1], Jie Li[2], Yifan Liu[2], Yaqian Xu[1], Kun Dou[1] & Tong Wang [1✉]

## Abstract

Directed axonal trafficking of mRNA via ribonucleoprotein (RNP) complexes is essential for neuronal function and survival. However, mechanisms governing retrograde RNP transport remain poorly understood. Here, we reveal that Annexin A7 (ANXA7) promotes the recruitment of aggregation-prone T-cell intracellular antigen 1 (TIA1)-containing RNPs to cytoplasmic dynein, enabling their retrograde trafficking to the soma for degradation. Both persistent and transient $Ca^{2+}$ elevation disrupted this function of ANXA7, leading to the detachment of TIA1 granules from dynein, impairing their transport, and subsequently triggering pathological TIA1 aggregation within axons. Similarly, ANXA7 knockdown decouples TIA1 granules from dynein, preventing their transport and inducing pathological aggregation of TIA1, which culminates in axonopathy and neurodegeneration both in vitro and in vivo. Conversely, ANXA7 overexpression reinforces trafficking and counteracts aberrant aggregation of TIA1-containing RNPs in axons. We describe here a $Ca^{2+}$-regulated mechanism which modulates retrograde axonal trafficking of RNPs and prevents the formation of pathological aggregates in axons.

**Keywords** Axon Trafficking; Phase Separation; Protein Aggregates; Calcium Signaling; Dynein
**Subject Categories** Cell Adhesion, Polarity & Cytoskeleton; Neuroscience; RNA Biology

## Introduction

Neurons are exceptionally long and polarized cells, with the axons of projection neurons extending up to 1 m (Cavanagh, 1984). The functions and survival of these neurons are critically reliant on the axon trafficking of proteins and mRNAs to support the demands of remote axonal compartments (Das et al, 2019; Das et al, 2021; Sleigh et al, 2019; Xiong and Sheng, 2024). mRNAs are packaged with RNA-binding proteins (RBPs) into membrane-less granules known as ribonucleoprotein complexes (RNPs), which serve as the transport units for axon trafficking (Abouward and Schiavo, 2021; Abraham and Fainzilber, 2022; Das et al, 2021). RNPs are transported by molecular motors, primarily kinesin and dynein, either through direct linkage to the motor via adapter proteins (Baumann et al, 2020; Geng et al, 2024; McClintock et al, 2018; Sladewski et al, 2018) or indirectly by tethering to the outer surface of membrane-bound organelles (Cioni et al, 2019; Liao et al, 2019; Quentin et al, 2023), enabling their transport along microtubule tracks (Abouward and Schiavo, 2021; Das et al, 2019; Turner-Bridger et al, 2018). Although motor-dependent trafficking is crucial for ensuring the correct localization and function of RNPs within neurons (Abouward and Schiavo, 2021; Abraham and Fainzilber, 2022; Dalla Costa et al, 2021; Das et al, 2021), the mechanisms underlying dynein-mediated retrograde transport of RNPs in mammalian axons remain largely unknown.

In addition to mRNA mislocation, accumulating evidence reveals that perturbations in axon trafficking also lead to toxic accumulation of RBPs within neurons (Dalla Costa et al, 2021; Sleigh et al, 2019). Specifically, some RBPs are capable of self-assembly into irreversible condensates that potentially play a key pathogenic role (Dalla Costa et al, 2021; Sleigh et al, 2019). These abnormal aggregates of certain RBPs are considered causal factors for axonopathy and neurodegeneration in diseases such as frontotemporal dementia (FTD) and amyotrophic lateral sclerosis (ALS) (Luan et al, 2024; Naskar et al, 2023; Xue et al, 2020). One prominent RBP involved in forming pathogenical aggregation is T-cell intracellular antigen 1 (TIA1), which possesses RNA recognition motifs (RRMs) at the N-terminus and a prion-like domain (PrLD) at its C-terminus that mediates its prion-like self-aggregation (Gilks et al, 2004). In response to axonal injury, TIA1 rapidly forms large RNPs near lesion sites via liquid-liquid phase separation (LLPS), repressing the translation of certain mRNAs and thus suppressing the axon regeneration in a PrLD-dependent manner (Andrusiak et al, 2019; Sahoo et al, 2018). Mutations in the PrLD of TIA1 can cause the formation of insoluble (pathological) aggregates and are associated with several neurodegenerative diseases, including FTD and ALS (Mackenzie et al, 2017), and Welander distal myopathy (WDM) (Hackman et al, 2013). Furthermore, interaction of wild-type TIA1 with physiological tau/mRNA can lead to toxic tau aggregate formation in neurons (Ash et al, 2021). Reducing TIA1 levels in vivo counteracts the tauopathy, protects against neurodegeneration, and prolongs the

[1]School of Life Science and Technology, ShanghaiTech University, Shanghai 201210, China. [2]Division of Chemistry and Physical Biology, School of Physical Science and Technology, ShanghaiTech University, Shanghai 201210, China. [3]These authors contributed equally: Yu Feng, Tongshu Luan. ✉E-mail: wangtong@shanghaitech.edu.cn

survival of P301S Tau mice, a model for Alzheimer's disease (Apicco et al, 2018). Despite these findings, mechanisms controlling the dynamics of TIA1-containing RNPs in axons remain elusive.

To address this gap, we tracked TIA1 granule movement in unidirectional axons of live neurons cultured in microfluidic devices, and revealed that TIA1 granules predominantly move retrogradely. Mass spectrometry analysis of TIA1 interactors from rat brain lysates identified recruitment of TIA1 to cytoplasmic dynein intermediate chain (DIC), facilitated by Annexin A7 (ANXA7). This complex mediates the retrograde transport of RNPs back to lysosomes in the soma. In vitro and axonal trafficking assays demonstrated that elevated $Ca^{2+}$ disrupts ANXA7's ability to recruit TIA1 granules to dynein. ANXA7 knockdown decoupled TIA1 from dynein, impairing its trafficking and leading to the axonal accumulation of its mRNA cargoes, such as *Ryk*, ultimately leading to the formation of large pathological TIA1 granule aggregates in axons. Conversely, ANXA7 overexpression enhanced trafficking efficiency, reducing TIA1 granule aggregates in axons. In vivo, ANXA7 knockdown in the mouse cortex led to corticospinal tract axonopathy, neurodegeneration, and motor deficits, all of which were rescued by expressing an shRNA-resistant ANXA7 variant. Our study uncovers a direct dynein-dependent mechanism driving the retrograde trafficking of TIA1-RNPs in axons, which counteracts the pathological aggregation and axonopathies, offering potential avenues for neurodegenerative diseases.

## Results

### TIA1 forms RNPs that undergo retrograde trafficking in axons

Known as one of the core proteins composing stress granules (SGs), TIA1 protein suppresses the mRNA translation in many types of cells, incuding the developing neural stem cells (Byres et al, 2021; Díaz-Muñoz et al, 2017; Dixon et al, 2003; López de Silanes et al, 2005; Piecyk et al, 2000), mediated by its PrLD, the LLPS of TIA1 controls the local translation of axonal mRNAs, and thus inhibits the regeneration capacity of injured axons (Andrusiak et al, 2019; Sahoo et al, 2018). Besides, LLPS-mediated aggregation of TIA1 also controls tauopathy, thus promoting the degeneration of axons (Apicco et al, 2018; Ash et al, 2021). Given these crucial roles of TIA1 in axon, whether its dynamics is regulated in axons remains largely unknown.

To address this question, in axons of rat hippocampal neurons cultured for 8 days (DIV8), we examined the distribution of endogenous TIA1 and G3BP, which label stress granules, or SQSTM1/p62, which labels ubiquitinated protein aggregates, and found that the subcellular localizations of these molecules are different (Appendix Fig. S1A). TIA1 overlaps with the latter two aggregation markers only in the expanded and beading regions of axons, but not along the unexpanded axon shafts (Appendix Fig. S1A'). These observations were further supported by results from lattice SIM, revealing the limited co-localization between TIA1 and either G3BP or p62 in unexpanded axon shafts (Appendix Fig. S1B,B'). These data agree with previous reports (Andrusiak et al, 2019; Sahoo et al, 2018), suggesting that TIA1 granules may have unknown functions in axon shafts.

To explore the potential TIA1 functions in axons, in DIV8 rat hippocampal neurons, we co-expressed EGFP-TIA1 and CY5-UTP, which label total RNA (Cioni et al, 2019), to mark the RNPs formed by TIA1 and RNA (Figs. 1A and EV1A). We found that in axons of living neurons, CY5-UTP RNPs are highly mobile with retrograde and anterograde directions (Fig. 1B; Movie EV1). Interestingly, EGFP–TIA1 overlaps with RNPs moving in the retrograde direction, as reflected by a significantly higher ratio of TIA1-positive RNPs in retrograde transport (Fig. 1B'). Next, we employed a microfluidic device to separate the axons of cultured hippocampal neurons from the somatodendritic part, as previously described (Wang et al, 2016; Wang and Meunier, 2022), which allowed sparse labeling of axons co-expressing TIA1-mCherry and the cytosolic shape marker EGFP amid a dense background of untransfected axons (Fig. 1C). These TIA1-mCherry-positive neurons could be visualized among the much larger number of untransfected βIII-tubulin-positive neurons (Appendix Fig. S1C). In the unidirectional axon bundles formed within the axon channels, directional trafficking of TIA1-granules could be traced and analysed (Fig. 1D). We observed that a substantial number of TIA1-mCherry granules undergo retrograde transport along axons (Fig. 1D; Movie EV2). When categorizing the trajectories of these granules into stationary ($-0.05\,\mu m/s < speed < 0.05\,\mu m/s$), retrograde (speed $\leq -0.05\,\mu m/s$), and anterograde (speed $\geq 0.05\,\mu m/s$) (Fig. 1D'; Appendix Fig. S1D), we found that $34.03 \pm 2.08\%$ of trajectories were retrograde, significantly higher than the $21.26 \pm 1.89\%$ of trajectories that were anterograde (Fig. 1D"). These data suggest that most mobile TIA1 granules are retrogradely transported within the axons of living neurons.

The most well-known mechanism for RNP axon trafficking is indirect tethering to membrane-bound organelles (Cioni et al, 2019; Liao et al, 2019; Quentin et al, 2023). We next explored whether TIA1 granules co-transport with retrograde membranous carriers in axons, using a pulse-chase labeling assay (Fig. EV1B), which labels various membranous organelles originating from axon terminal (Wang and Meunier, 2022). Fluorescently tagged CTB labels signaling endosomes (Wang et al, 2016) (Fig. EV1C; Movie EV3), the heavy chain of Botulinum neurotoxin (BoNT/A-Hc) labels autophagosomes and multivesicular bodies (MVBs) derived from synaptic vesicles (Wang et al, 2015) (Fig. EV1D; Movie EV3), LysoTracker labels lysosomes (Wang et al, 2020) (Fig. 1E; Movie EV3), MitoTrakcer labels mitochondria (Fig. EV1E; Movie EV3), and EGFP-Rab5 labels early endosomes (Fig. EV1F; Movie EV3). We found that TIA1 granules show limited co-trafficking with these membraneous organelles (Fig. 1E',F'). Notably, the retrograde motor cytoplasmic dynein, labeled by its neuron-specific isoform of intermediate chain (DIC1B-mRFP) (Ha et al, 2008), demonstrated the highest co-trafficking percentage with TIA1 granules in live axons (Fig. 1F,F'; Movie EV4). Consistently, in fixed axons of neurons, endogenous TIA1 and these membrane axon carriers also showed limited co-localization, with the highest overlap observed with dynein (Fig. EV1G-G"'). These data suggest that retrograde trafficking of TIA1 granules in axons is likely driven by dynein, rather than relying exclusively on tethering to membranous carriers. Moreover, knockdown of endogenous DIC1B using two independent shRNA constructs (shDIC1B-1# and 2#) (Fig. EV1H) significantly impaired the retrograde trafficking of TIA1-mCherry granules in neurons (Fig. 1G,G'). Similarly, disruption of microtubule tracks with nocodazole markedly reduced the axonal

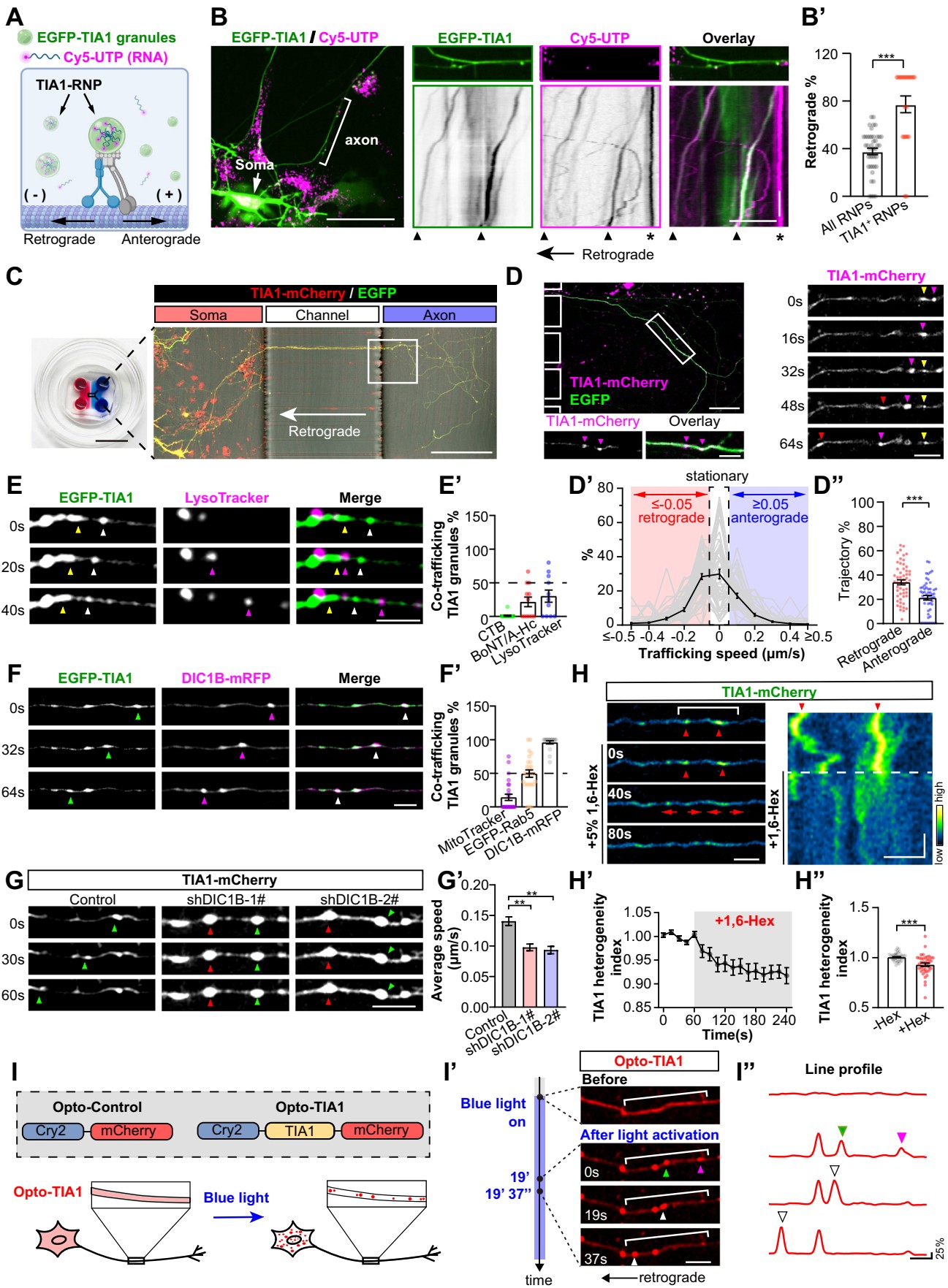

**Figure 1.  TIA1 forms membrane-less RNPs that undergo retrograde trafficking in axons.**

(A) Schematic diagram illustrating RNP labeling in neurons co-transfected with EGFP-TIA1 and Cy5-UTP. (B) Live-cell imaging of DIV8 rat hippocampal neurons reveals directed axonal trafficking of RNPs. Kymographs of bracketed axons show retrograde (triangles) and stationary (asterisks) RNPs. Scale bars = 50 μm (left), 20 μm (right); y-axis = 180 s. (B') Percentage of TIA1-positive RNPs and all RNPs undergoing retrograde trafficking ($n$ = 22, 42 neurons derived from four biological replicates, $P < 0.0001$). (C) Schematic of the microfluidic device. Scale bars = 1 cm (left), 200 μm (right). (D) Dynamics of TIA1-mCherry granules in the axons of DIV8 neurons. Magnified boxed regions on the right show distinct TIA1 granules, marked by colored arrowheads. Scale bars = 50 μm (left top) and 10 μm (left bottom, right). (D') Distribution profile of average trafficking speeds for TIA1 granules: retrograde ($\leq -0.05$ μm/s), anterograde ($\geq 0.05$ μm/s), stationary ($-0.05$ to $0.05$ μm/s). (D") Percentage of retrograde and anterograde TIA1 granules ($n$ = 53 axons from four biological replicates, $P < 0.0001$). (E–F') Key frames from time-lapse images showing co-trafficking of TIA1 granules with LysoTracker (E) and DIC1B-mRFP (F) in axons. Arrowheads in different colors mark distinct mobile particles. Scale bar = 5 μm. (E') Quantification of TIA1 granules co-trafficking with axon-derived CTB, BoNT/A-Hc, or LysoTracker ($n$ = 14, 12, 11 axons from three biological replicates). (F') Quantification of TIA1 granules co-trafficking with whole cell–applied MitoTracker, or with co-expressed EGFP-Rab5 and DIC1B-mRFP ($n$ = 25, 25 and 18 axons from at least 3 biological replicates). (G) Key frames from time-lapse images showing TIA1-mCherry axon trafficking in DIC1B knockdown neurons. Scale bars = 10 μm. (G') Speed quantification of (G') ($n$ = 216, 51 and 54 granules from four biological replicates. Control vs. shDIC1B-1#: $P = 0.0044$; Control vs. shDIC1B-2#: $P = 0.0011$). (H) Key frames from time-lapse images showing axonal TIA1 granules (indicated by arrowheads). Disassembly of these granules is marked by red arrows. Right: Kymograph of the indicated axonal segment. Scale bar = 5 μm; y-axis = 10 s. (H'-H") Quantification shows TIA1 intensity heterogeneity along axons (H') before and 180 s post-1,6-Hex (H") ($n$ = 40 axons from three biological replicates, $P < 0.0001$). (I) Design of Opto-Control and Opto-TIA1 constructs (top). Schematics illustrating light-induced formation of Opto-TIA1 granules in neurons (bottom). (I') Key frames from time-lapse images showing movement of Opto-TIA1 granules in axons. The timeline on the left indicates the onset of light activation. Intensity profiles of the bracketed area shown on the right. Moving and fusing granules are indicated by arrowheads. Scale bars = 10 μm (left), 5 μm (right); y-axis = 25% (Normalized to ($F_{Max}$-$F_0$)). Data represent mean ± SEM; two-tailed unpaired $t$-test in (B', D", H"); one-way ANOVA in (G'). Source data are available online for this figure.

transport of TIA1 granules (Fig. EV1I,I'). Together, these results demonstrate that TIA1-granule trafficking in axons depends on the microtubule-based retrograde motor dynein.

Next, we examined whether the axonal TIA1 granules are membrane-less, by employing two different approaches. First, we used the 5% 1,6-Hexanediol (1,6-Hex) to treat DIV12 neurons, with the dynamics of the TIA1-mCherry monitored under the live-imaging microscopy. We found that the addition of 1,6-Hex causes significant disassembly of TIA1-mCherry granules (Fig. 1H), reflected by the significantly reduced heterogeneity index (Fig. 1H',H"), which is defined in Appendix Fig. S1E reflects the extent of uneven distribution of fluorescent signals along the thin axons. Second, we used the optoDroplet system (Shin et al, 2017) to construct the light-induced Opto-TIA1, which forms droplets when exposed to blue laser (Fig. 1I). The earliest light-induced Opto-TIA1 droplets were observed ~120 s after light activation (Appendix Fig. S1F; Movie EV5), leading to a significantly raised heterogeneity index (Appendix Fig. S1F'). Some of these light-induced Opto-TIA1 granules underwent rapid fusion, as indicated by the merging of two droplets (Fig. 1I',I"; Movie EV6; magenta- and green-arrowheads) into a larger granule (white-arrowheads). Interestingly, this fused granule subsequently underwent retrograde axon trafficking (white-arrowheads). Results from these two approaches demonstrate that axonal TIA1 granules are indeed membrane-less droplets formed via LLPS.

In summary, these co-trafficking data suggest that the majority of TIA1 granules in axons are retrogradely transported as membrane-less RNPs that may primarily associate with dynein rather than rely on tethering to membranous organelles, although potential interactions with other membrane-bound compartments cannot be fully excluded. These TIA1-containing RNPs may have novel functions in axons, distinct from their known SG-related roles.

## ANXA7 enhances the interaction between TIA1 and dynein

To elucidate the mechanisms underlying the direct dynein-driven retrograde trafficking of TIA1 granules in axons, we first compared

known TIA1 interactors with those of cytoplasmic dynein intermediate chain 1B (DIC1B), the primary homolog of the dynein intermediate chain 1 in neurons (Ha et al, 2008), using data from the BioGRID database. This comparison revealed four shared proteins: ANXA7, CUL3, C2orf44, and SMN1 (Fig. 2A; Dataset EV1). Next, we expressed recombinant TIA1 in *E. coli* and analysed its interactome using purified GST-TIA1 from P14 rat brain lysates. Following tryptic digestion of GST bead-bound proteins, the resulting peptides were subjected to liquid chromatography-tandem mass spectrometry (LC–MS/MS). Using specific screening criteria (see Methods), we identified 63 proteins enriched in GST-TIA1 compared to GST alone (Fig. 2B; Dataset EV2). Gene Ontology (GO) and Kyoto Encyclopedia of Genes and Genomes (KEGG) analysis revealed that these TIA1-associated proteins were enriched in processes related to cytoplasmic stress granule assembly, translation, RNA processing and RNA binding and are associated with neurodegenerative diseases (Appendix Fig. S2A; Dataset EV3), consistent with TIA1's established functions (Del Gatto-Konczak et al, 2000; Gilks et al, 2004; Kedersha et al, 1999; López de Silanes et al, 2005; Piecyk et al, 2000; Yamasaki et al, 2007). Among the TIA1 interactors, CUL3 and ANXA7 were detected (Fig. 2B). Given ANXA7's roles in facilitating intracellular trafficking of various organelles (Creutz, 1992; Li et al, 2013; Sønder et al, 2019; Wang et al, 2010) and its confirmed presence in GST-TIA1 pull-downs from P14 rat brain lysates (Fig. 2B,C), we focused on examining whether ANXA7 acts as the linker between TIA1 and dynein.

Next, we conducted a pull-down assay using recombinant GST-ANXA7 and P14 rat brain homogenates, with GST-peptide as the negative control. This identified both dynein intermediate chain (DYNC1i1, DIC1) and TIA1 as ANXA7-associated proteins (Fig. 2D; Dataset EV2), confirmed by western blot using specific antibodies (Fig. 2E). GO term and KEGG pathway enrichment analyses of the ANXA7 interactome revealed significant enrichment in pathways related to microtubule-dependent transport, membrane-less organelle (MLO) assembly, RNP granules, vesicle trafficking, distal axon function, $Ca^{2+}$-dependent activities, and mRNA binding (Fig. EV2A; Dataset EV3), suggesting a role for

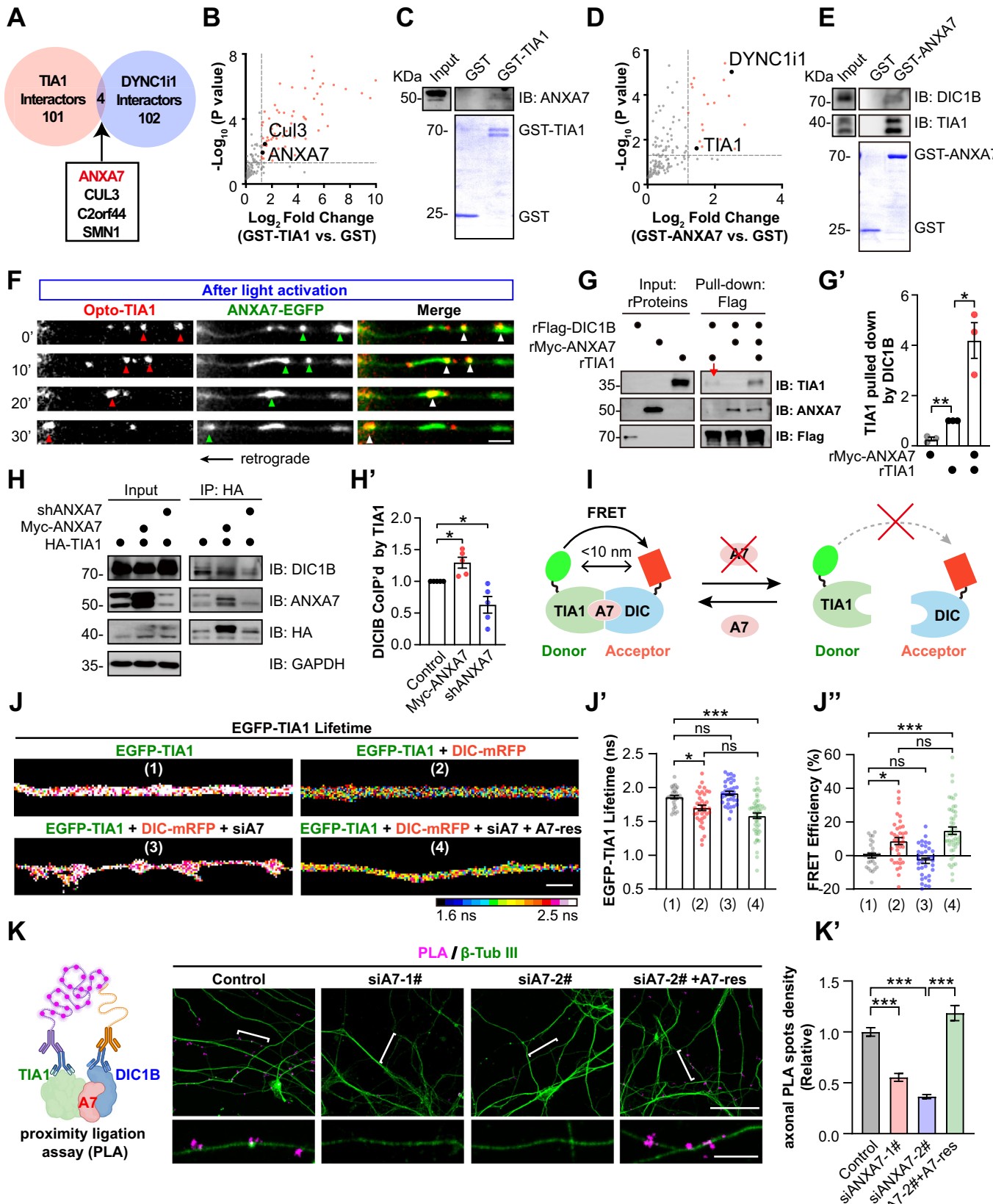

**Figure 2.   ANXA7 enhances the interaction between TIA1 and dynein in neurons.**

(A) Venn diagram showing four shared interactors between TIA1 and DIC1B interactomes (DYNC1i1 in BioGRID, see Methods and Dataset EV1). (B) Mass spectrometry analysis of proteins interacting with GST-TIA1 in rat brain lysates, using GST tag as a control. Red dots indicate significantly enhanced interactors ($p < 0.05$ and $log_2$ fold change >1.2). Data from three replicates; statistical significance assessed by paired *t*-test. (C) Immunoblots of ANXA7 in proteins pulled down by GST-TIA1 from rat brain. (D) Mass spectrometry analysis of proteins interacting with GST-ANXA7 in rat brain lysates, using GST tag as a control. Red dots indicate significantly enhanced interactors ($p < 0.05$ and $log_2$ fold change >1.2). Data from three replicates; statistical significance assessed by paired *t*-test. (E) Immunoblots of TIA1 and DIC1B in proteins pulled down by GST-ANXA7 from rat brain. (F) Key frames from time-lapse images showing retrograde co-trafficking of light-induced Opto-TIA1 (red) and ANXA7-EGFP (green) granules in DIV9 rat hippocampal neurons. Scale bar = 2 μm. Arrowheads indicate co-trafficking. (G) Purified recombinant Myc-ANXA7 (rMyc-ANXA7) protein enhances rTIA1 and rFlag-DIC1B interaction, shown by increased rTIA1 pulled down by rFlag-DIC1B. The arrow indicates the weak interaction between TIA1 and DIC1B observed in the absence of rANXA7. (G') Quantification of (G), data from three biological replicates (Myc-ANXA7 vs. TIA1: $P = 0.0088$; TIA1 vs. TIA1 + Myc-ANXA7: $P = 0.0463$). (H) Co-IP assay examining the interaction between endogenous DIC1B and HA-tagged TIA1 using anti-HA magnetic beads in DIV11 rat cortical neurons. The interaction is studied under endogenous ANXA7 knockdown (shANXA7) or Myc-ANXA7 overexpression conditions. (H') Quantifying TIA1-DIC1B interaction from (H) shows the effects of different ANXA7 levels ($n = 5$ technical replicates from four biological replicates. Control vs. Myc-ANXA7: $P = 0.0267$; Control vs. shANXA7: $P = 0.0474$). (I) Schematic diagram of FLIM-FRET to examine the affinity between EGFP-TIA1 (donor) and DIC1B-mRFP (acceptor) under varying levels of ANXA7 (A7). (J) Represented images showing color-coded EGFP-TIA1 lifetime in axon shafts of transfected neurons, with lifetime (J') and FRET efficiency (J") quantified and compared across indicated groups. Scale bar = 2 μm ($n = 29, 37, 35,$ and 49 axons from four biological replicates. (1) vs. (2): $P = 0.0266$; (1) vs. (4): $P < 0.0001$). (K) Left: schematic illustrating PLA detection of endogenous TIA1 and DIC1B interaction. Right: representative confocal images showing TIA1/DIC1B PLA signals in neurons with varying ANXA7 (A7) levels; bracketed axons are enlarged below. Scale bars = 50 μm (top), 10 μm (bottom). (K') Quantification of axonal PLA density from (K) ($n = 172, 90, 172,$ and 90 ROIs from six biological replicates. All $P < 0.0001$). Data represent mean ± SEM; one-sample *t*-test in (G', H'); one-way ANOVA in (J', J", K'). Source data are available online for this figure.

ANXA7 in RNP axonal trafficking. KEGG pathway analysis also highlighted neurodegenerative disease pathways, including ALS, Parkinson's disease (PD), and Huntington's disease (HD)—all characterized by early axon degeneration (Luan et al, 2024). Supporting this hypothesis, Myc-ANXA7 co-immunoprecipitated (co-IP'd) with HA-TIA1 when co-expressed in HEK293T cells (Appendix Fig. S2B). Additionally, co-expressed HA-DIC1B co-IP'd with Flag-TIA1, likely facilitated by endogenous ANXA7 in HEK293T cells. Notably, overexpression of Myc-ANXA7 significantly increased the amount of HA-DIC1B co-IP'd with Flag-TIA1 (Appendix Fig. S2C,C'). These findings suggest that ANXA7 may function as a crucial adapter molecule, enhancing the interaction between TIA1 granules and dynein.

We explored the expression patterns of endogenous TIA1, ANXA7, and DIC1B in mouse brains using specific antibodies and confocal microscopy. High expression levels of all three proteins were observed in both the cortex and hippocampus, with significant co-localization in neuronal cell bodies of the motor cortex and hippocampal CA3 region (Fig. EV2B). Subcellular distribution in the axons of DIV12 cultured hippocampal neurons was examined using Lattice SIM, revealing significant co-localization of ANXA7, TIA1, and DIC1B within the axon shafts (Fig. EV2C,C'). Additionally, light-induced Opto-TIA1 droplets demonstrated co-trafficking in the retrograde direction with the overexpressed ANXA7-EGFP granules in axons (Fig. 2F; Movie EV7). Time-lapse Lattice SIM imaging of EGFP-TIA1 and ANXA7-mCherry in living neurons showed their co-trafficking along the axon shafts, also highly overlapped in newly formed TIA1/ANXA7 dual positive granules (Fig. EV2D; Movie EV8). These findings suggest that TIA1, ANXA7, and DIC1B likely coexist within the same complex, facilitating the retrograde trafficking of TIA1 granules in axons.

To elucidate whether ANXA7 functions as an obligatory linker or as an affinity enhancer between TIA1 and dynein, we expressed and purified recombinant TIA1, Flag-DIC1B, and Myc-ANXA7 proteins in *E. coli* (Appendix Fig. S2D). Using an in vitro pull-down assay, in which recombinant Flag-DIC1B was immobilized on anti-Flag antibody-coated beads and incubated with purified TIA1 in the presence or absence of Myc-ANXA7 (Appendix Fig. S2E; see

Methods), we found that only a very low amount of TIA1 pulled down with DIC1B in the absence of ANXA7, whereas the addition of Myc-ANXA7 significantly increased the amount of TIA1 pulled down with DIC1B by approximately four-fold (Fig. 2G,G'; Appendix Fig. S2F). These data demonstrate that ANXA7 strongly enhances the weak interaction between TIA1 and dynein in vitro. To validate this role of ANXA7 in neurons, we modulated endogenous ANXA7 levels in DIV11 rat cortical neurons expressing HA-tagged TIA1, and found that reducing ANXA7 expression (shANXA7; Fig. EV2E) significantly decreased the DIC1B-TIA1 interaction, while overexpression of Myc-ANXA7 enhanced the interaction (Fig. 2H,H'). Furthermore, using Lattice SIM, we assessed the impact of modulating ANXA7 expression on TIA1 recruitment to dynein in the axons of cultured neurons. We found that overexpression of ANXA7-EGFP significantly increased the co-localization of endogenous TIA1 with DIC1B, whereas knockdown of ANXA7 (shA7 1# and 2#) reduced their co-localization (Fig. EV2F,F'). These results demonstrate that ANXA7 significantly promotes the recruitment of TIA1 granules to dynein in axons.

To investigate whether ANXA7 regulates TIA1 recruitment to dynein in living neurons, we employed fluorescence lifetime imaging microscopy (FLIM). The close proximity between EGFP-TIA1 (donor) and DIC1B-mRFP (acceptor) would lead to fluorescence resonance energy transfer (FRET), thereby reducing the fluorescence lifetime of EGFP-TIA1, an indication of their interaction. Manipulating ANXA7 levels should affect the FLIM-FRET efficiency between EGFP-TIA1 and DIC1B-mRFP if ANXA7 acts as a linker (Fig. 2I). We conducted experiments by knocking down ANXA7 using shRNA (shA7-1#, shA7-2#) and overexpressing a Myc-ANXA7 mutant (Myc-ANXA7-res) resistant to ANXA7 knockdown (Appendix Fig. S2G). In axons of hippocampal neurons co-expressing EGFP-TIA1 and DIC1B-mRFP (Fig. 2J, (2)), the fluorescence lifetime of EGFP-TIA1 (Fig. 2J') was significantly reduced, and FRET efficiency (Fig. 2J") increased compared to neurons expressing EGFP-TIA1 alone (Fig. 2J, (1)), indicating FRET between EGFP-TIA1 and DIC1B-mRFP. Conversely, ANXA7 knockdown (Fig. 2J, (3)) significantly prolonged the fluorescence lifetime of EGFP-TIA1 (Fig. 2J') and decreased FRET

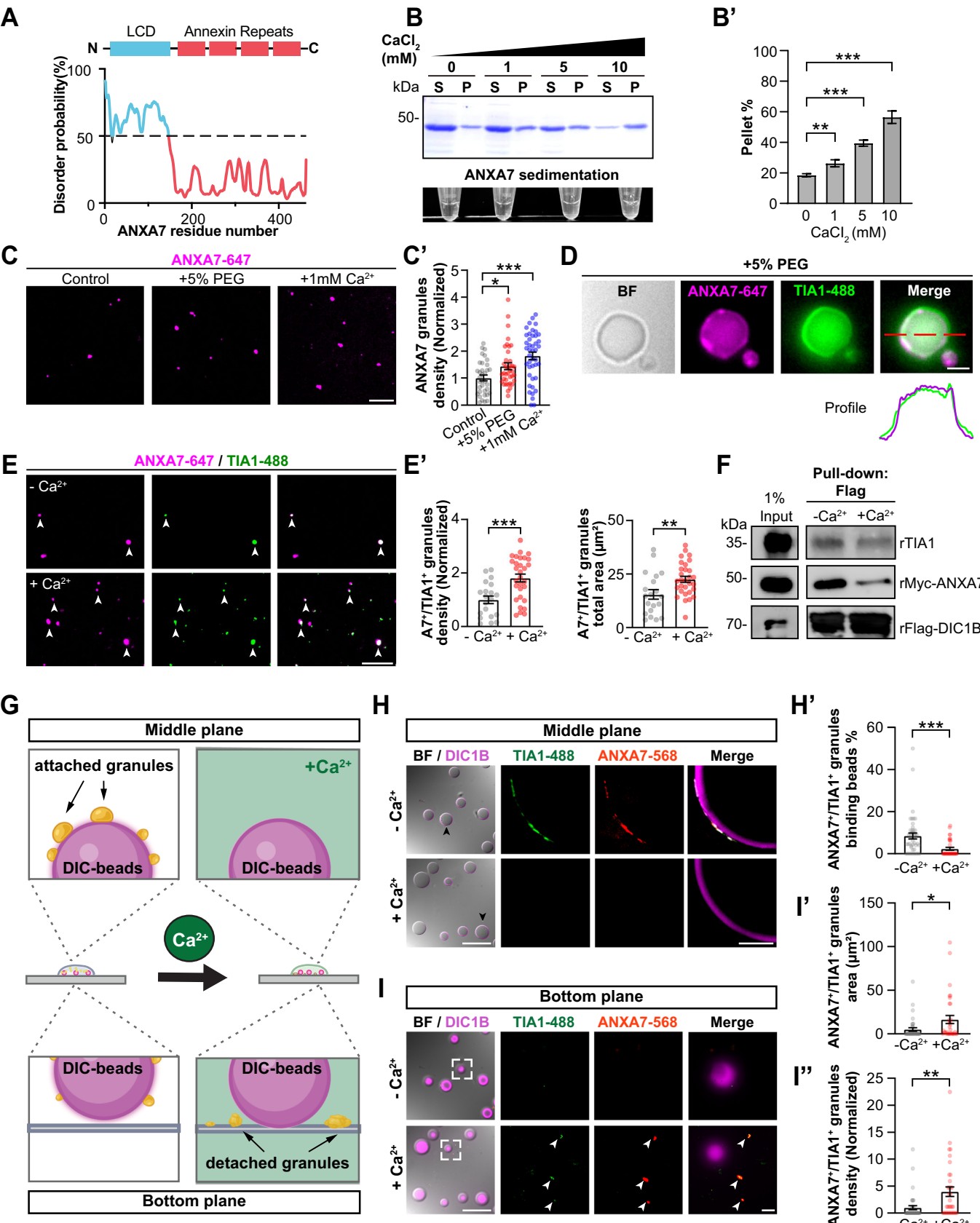

◀ **Figure 3. Ca²⁺ promotes ANXA7 aggregation, inhibiting the recruitment of TIA1 to dynein.**

(A) Schematic of ANXA7 domain structure with PrDOS analysis. (B) Top: In vitro sedimentation assay detected by SDS-PAGE showing the distribution of purified ANXA7 (5 µM) between supernatant (S) and pellet (P) at the indicated Ca²⁺ concentrations. Bottom: appearance of ANXA7 solutions in tubes, showing emulsification caused by LLPS. (B') Quantification of (B) ($n = 11$ technical replicates from 4 biological replicates. 0 mM vs. 1 mM: $P = 0.0062$; 0 mM vs. 5 mM: $P < 0.0001$; 0 mM vs. 10 mM: $P < 0.0001$). (C) In vitro phase separation of ANXA7-647 induced by PEG or Ca²⁺. Scale bar = 10 µm. (C') Quantification of (C) ($n = 35, 38, 43$ ROIs from three biological replicates. Control vs. +5% PEG: $P = 0.0355$; Control vs. 1 mM Ca²⁺: $P < 0.0001$). (D) In vitro droplet formation assay demonstrating co-existence of purified ANXA7-647 and TIA1-488 proteins in droplets induced by 5% PEG. Right: intensity profile along the dashed line in the merged image. Scale bar = 2 µm. (E) Confocal microscopy showing increased phase separation of ANXA7-647 and TIA1-488 (both 5 µM) with 1 mM Ca²⁺ addition. Dual-positive droplets are indicated by arrowheads. Scale bar = 10 µm. (E') Quantifications of (E) ($n = 20, 30$ ROIs from three biological replicates. Density -Ca²⁺ vs. +Ca²⁺: $P = 0.0004$; total area -Ca²⁺ vs. +Ca²⁺: $P = 0.007$). (F) In vitro interaction assay showing the effect of 1 mM Ca²⁺ on the interaction between rTIA1, rMyc-ANXA7, and rFlag-DIC1B, with rFlag-DIC1B pulled down using anti-Flag beads. (G) Schematic diagram of the confocal microscopy-based LLPS assay. See also the Methods for details. (H, I) Representative confocal images of the middle plane (H) or the bottom plane (I) of the DIC-coated beads with or without 1 mM Ca²⁺, showing the ANXA7⁺/TIA1⁺ granules attached (H) or detached (I) to beads. The arrowed or boxed regions in the bright field (BF) channel were further amplified in the right panels. Scale bars = 200 µm (left), 20 µm (right). (H') Quantification of the percentage of ANXA7⁺/TIA1⁺ granules binding beads in the medial plane shown in (H) ($n = 46, 35$ ROIs from three biological replicates. $P = 0.0007$). (I, I') Quantifications of the total area (I) and density (I') of ANXA7⁺/TIA1⁺ condensates in the bottom plane, as shown in (I) ($n = 39, 33$ ROIs from three biological replicates. Area -Ca²⁺ vs. +Ca²⁺: $P = 0.0181$; density -Ca²⁺ vs. +Ca²⁺: $P = 0.0023$). Data represent mean ± SEM; two-tailed unpaired $t$-test in (B', E', H', I', I''); one-way ANOVA in (C'). Source data are available online for this figure.

efficiency (Fig. 2J''), suggesting a weakened TIA1-DIC1B interaction. Remarkably, co-expression of Myc-ANXA7-res with siA7 (Fig. 2J, (4)) restored the EGFP-TIA1 lifetime and FRET efficiency to near baseline levels (Fig. 2J',J''), demonstrating that ANXA7 is crucial for maintaining the TIA1-DIC1B interaction. Consistently, a similar trend of altered FLIM-FRET efficiency changes were also observed in soma upon ANXA7 level manipulation (Fig. EV2G-G'').

Next, we performed proximity ligation assays (PLA) in cultured hippocampal neurons to examine the interactions among endogenous DIC1B, ANXA7, and TIA1 using their specific antibodies (Appendix Fig. S2H). PLA signals revealed that TIA1 interacts with ANXA7, and ANXA7 with DIC1B in axons, since ANXA7 knockdown led to a marked reduction in PLA signals for both TIA1–ANXA7 and ANXA7–DIC1B pairs (Fig. EV2H–I'), supporting the existence of direct physical interactions between each pair. Moreover, knockdown of endogenous ANXA7 significantly reduced the PLA signal between TIA1 and DIC1B in axons. Importantly, this reduction was rescued by co-expression of an shANXA7-resistant ANXA7 mutant (A7-res), which restored the PLA signal to control levels (Fig. 2K,K'). These results demonstrate that ANXA7 plays a critical role in promoting the association between TIA1 and dynein in neurons.

## Ca²⁺ elevation disrupts ANXA7-mediated TIA1 recruitment to dynein

ANXA7 contains N-terminal low-complexity domains (LCDs) that potentially mediate LLPS, as illustrated in Fig. 3A. In vitro LLPS assays demonstrated that purified ANXA7 protein forms droplets in a concentration-dependent manner (Fig. EV3A,A'). Droplet formation was significantly enhanced by the addition of the crowding agent PEG-8000 (Fig. EV3A,A''), aligning with previous findings (Yu et al, 2023). The C-terminal region of ANXA7 harbors Ca²⁺-sensitive Annexin repeats (Gerke et al, 2024), which, at millimolar Ca²⁺ concentration, have been shown to trigger LLPS and promote droplet aggregation near plasma membrane (PM) lesions in cancer cells (Gerke et al, 2024; Sønder et al, 2019). We then investigated whether intracellular Ca²⁺ affects ANXA7 function in recruiting TIA1 to dynein.

First, we assessed whether Ca²⁺ influences the LLPS of ANXA7 by measuring the sedimentation of purified ANXA7 proteins across a range of Ca²⁺ concentrations (0–10 mM). We found elevated Ca²⁺ levels significantly increased the segregation of ANXA7 into the pellet fraction (Fig. 3B,B'), accompanied by visible emulsification in the tubes (Fig. 3B, bottom), indicating robust Ca²⁺-induced LLPS. In contrast, altering Na⁺ concentrations (0–500 mM; Fig. EV3B,B') or Mg²⁺ concentrations (0–10 mM; Fig. EV3C,C') had no detectable effect on ANXA7 sedimentation. These data indicate that Ca²⁺ elevation specifically triggers ANXA7 LLPS, with the effect of 1 mM Ca²⁺ being comparable to that of 5% PEG in vitro (Fig. 3C,C'). In the presence of purified TIA1, ANXA7 droplets induced by either 5% PEG or 1 mM Ca²⁺ extensively overlapped with TIA1 droplets (Fig. 3D,E). Notably, TIA1 alone formed droplets in vitro with PEG (Fig. EV3D–F'), but not in response to Ca²⁺ in the absence of ANXA7 (Fig. EV3F,F''-G'), suggesting that the Ca²⁺-triggered formation of TIA1 and ANXA7 droplets (A7⁺/TIA1⁺ droplets) is dependent on ANXA7 (Fig. 3E,E'). These findings indicate that Ca²⁺ elevation induces ANXA7 LLPS, subsequently facilitating TIA1 condensation into the same droplets.

Next, we evaluated the impact of Ca²⁺ elevation on ANXA7's ability to recruit TIA1 to dynein. Our in vitro pull-down assay showed that 1 mM Ca²⁺ significantly reduced the amount of both purified rMyc-ANXA7 and rTIA1 proteins pulled down with rFlag-DIC1B (Fig. 3F). To further evaluate the impact of Ca²⁺ on recruitment efficiency, we employed a confocal microscopy-based LLPS assay (see Methods). This allowed us to visualize and quantify A7⁺/TIA1⁺ droplets attached or detached from DIC1B-coated beads across two optical planes: the middle plane, showing the intersection of DIC1B-coated beads with attached A7⁺/TIA1⁺ droplets (Fig. 3G, top), and the bottom plane, showing detached condensates (Fig. 3G, bottom). We found that 1 mM Ca²⁺ significantly reduced the attachment of A7⁺/TIA1⁺ granules to DIC1B-coated beads in the middle plane (Fig. 3H,H'), while significantly increasing the size and number of detached A7⁺/TIA1⁺ droplets in the bottom plane (Fig. 3I-I''). These results indicate that Ca²⁺ elevation not only induces the formation of ANXA7⁺/TIA1⁺ droplets but also promotes their detachment from dynein.

## Disruption of ANXA7-mediated trafficking causes TIA1 aggregation in axons

To assess the effect of ANXA7-mediated TIA1 axonal transport under elevated intracellular Ca²⁺ conditions, we used KCl depolarization to

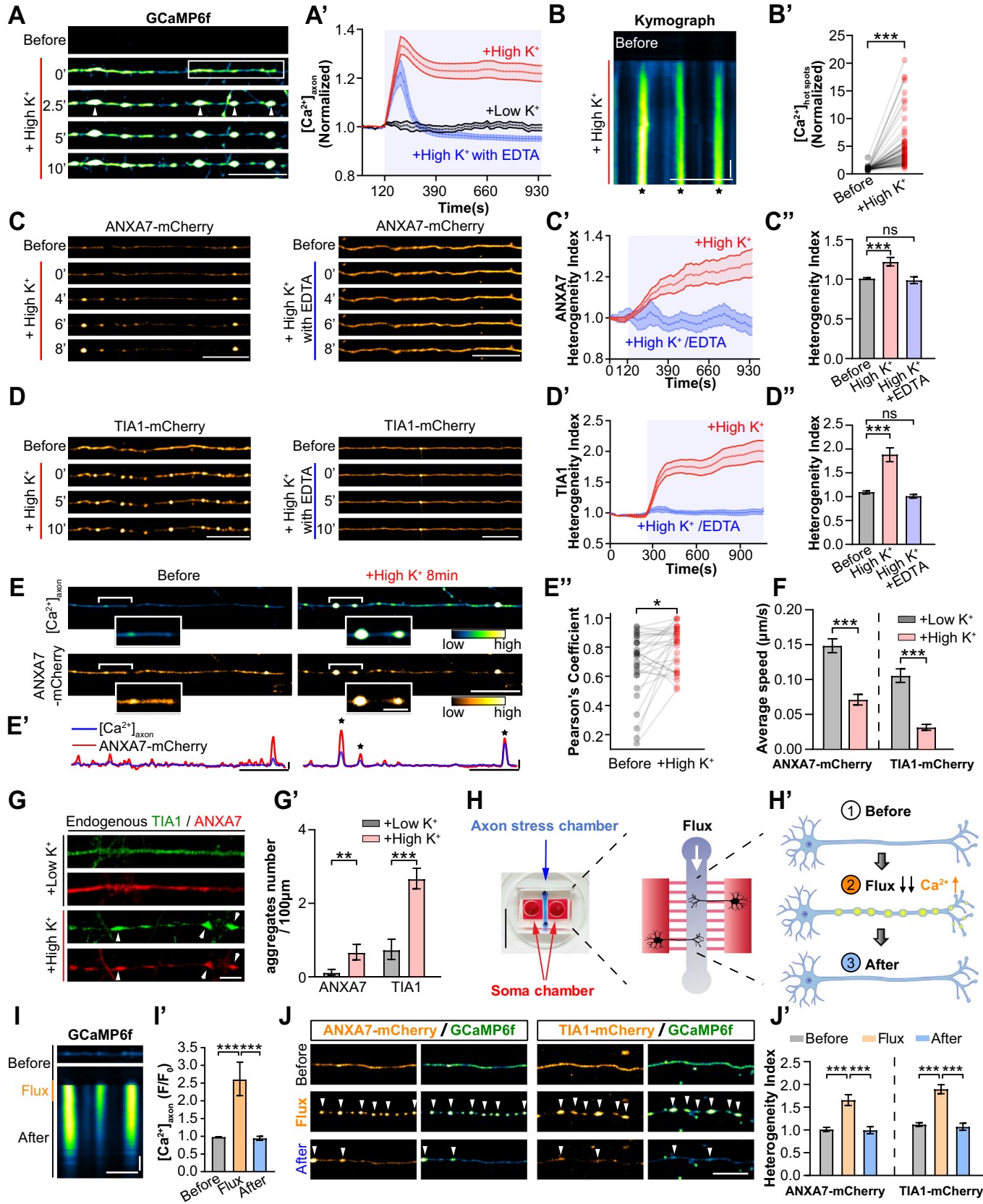

**Figure 4. Focal Ca²⁺ elevation causes ANXA7 and TIA1 aggregation in axons.**

(A) Key frames from time-lapse images of axons of DIV13 rat hippocampal neurons expressing the Ca²⁺ sensor GCaMP6f, Ca²⁺ elevating "hot spots" indicated with arrows. Scale bar = 20 μm. (A') Quantification of $[Ca^{2+}]_{axon}$ under indicated stimulations (Low K⁺: $n = 37$; High K⁺: $n = 72$; High K⁺ with EDTA: $n = 56$ axons from three biological replicates). (B) Kymographs of the boxed axon segment in (A), asterisks denote hot spots. Scale Bar = 20 μm; y-axis = 100 s. (B') Quantification of the "hot spots" Ca²⁺ concentration before and after 10 min of high K⁺ stimulation ($n = 70$ spots from three biological replicates. $P < 0.0001$). (C) Key frames from live imaging of ANXA7-mCherry distribution in axons before and after addition of high K⁺ or high K⁺/EDTA (0.5 mM). Scale bar = 20 μm. (C'–C'') Quantification of ANXA7-mCherry intensity heterogeneity over time (C') and after stimulation 8 min (C'') ($n = 55, 55, 36$ axons from four biological replicates. Before vs. High K⁺: $P < 0.0001$, ns non-significant). (D) Key frames from live imaging of TIA1-mCherry distribution in axons before and after addition of high K⁺ or high K⁺/EDTA (0.5 mM). Scale bar = 20 μm. (D'–D'') Quantification of TIA1-mCherry intensity heterogeneity over time (D') and after stimulation 8 min (D'') ($n = 55, 55, 56$ axons from five biological replicates. Before vs. High K⁺: $P < 0.0001$, ns non-significant). (E) Dual-color images of neurons co-expressing GCaMP6f and ANXA7-mCherry, bracketing axons magnified in insets. Scale bar = 5 μm (left), 20 μm (right). (E') Line profiles illustrating fluorescence intensity fluctuations from (E), hot spots denoted by asterisks. Scale bar = 20 μm; y-axis = 25% (Normalized to ($F_{Max}-F_0$)). (E'') Pearson's coefficient showing the correlation between Ca²⁺ hot spots and ANXA7-mCherry localizations before and after high K⁺ stimulation for 8 min ($n = 30$ axons from three biological replicates. Before vs. High K⁺: $P = 0.0106$). (F) Quantification of the average speed of TIA1 and ANXA7-mCherry granules under indicated conditions (ANXA7: $n = 241$ and $134$ granules; TIA1: $n = 236$ and $127$ granules; all from three biological replicates. ANXA7: $P < 0.0001$; TIA1: $P < 0.0001$). (G) Confocal images of endogenous TIA1 and ANXA7 distribution in axons of DIV12 rat hippocampal neurons after 10 min low or high K⁺ stimulation. Scale bar = 10 μm. (G') Quantification of (G), showing the number of TIA1 or ANXA7 granules per 100-μm axon of indicated stimulations ($n = 48, 37$ axons from three biological replicates. ANXA7: $P = 0.0098$; TIA1: $P < 0.0001$). (H) Schematic of the AoC device. Neurons were seeded in the soma chambers (red), and axons extended into the central stress chamber (blue). Scale bar = 1 cm. (H') Three steps of flux-induced stress: ① Before, ② Flux—culture medium injected into the stress chamber at 50 μL/min for 180 s to induce focal axonal Ca²⁺ elevation, and ③ After—recovery of Ca²⁺ levels. (I) Kymograph of a fluxed axon. Scale bar: 5 μm; y-axis: 120 s. (I') Quantification of $[Ca^{2+}]_{axon}$ during the Before, Flux and After phases ($n = 30$ axons from three biological replicates. Before vs. Flux: $P = 0.0001$; Flux vs. After: $P < 0.0001$). (J) Changes in ANXA7-mCherry and TIA1-mCherry aggregation in axons during the Before, Flux, and After phases. Scale Bar = 10 μm. (J') Quantification of (J) ($n = 30$ axons from three biological replicates. All $P < 0.0001$). Data represent mean ± SEM; two-tailed unpaired t-test in (F, G'); two-tailed paired t-test in (B', E''); one-way ANOVA in (C'', D', I', J'). Source data are available online for this figure.

induce significant and persistent Ca²⁺ elevation in cultured live neurons(Ouardouz et al, 2003; Xia et al, 1996). We found that application of 56 mM KCl (high K⁺) to DIV11-14 hippocampal neurons expressing the Ca²⁺ sensor GCaMP6f led to rapid and global increase in Ca²⁺ levels, represented by the sharp rise in GCaMP6f fluorescence intensity in both somatodendrites (Appendix Fig. S3A-B'; Movie EV9) and axons (Fig. 4A). Specifically, the Ca²⁺ intensity in the axon shaft ($[Ca^{2+}]_{axon}$) increased by 33 ± 3.7% and remained elevated for over ten minutes (Fig. 4A'). Within expanded axonal regions or "hot spots" (Fig. 4B, asterisks), $[Ca^{2+}]_{axon}$ levels rose by ~4.79 ± 0.43-fold (Fig. 4B'). In contrast, 5.6 mM KCl (Low K⁺) did not significantly affect Ca²⁺ levels in either somatodendritic (Appendix Fig. S3B,B') or axonal regions (Fig. 4A'; Appendix Fig. S3C-C''). Co-application of the Ca²⁺ chelator EDTA completely abolished the High K⁺-induced elevation (Fig. 4A'; Appendix Fig. S3D-D''), confirming its specificity.

Having established a live-cell model of depolarization-induced, persistent Ca²⁺ elevation in axonal hot spots, we next examined its effect on ANXA7-mediated axonal trafficking of TIA1. ANXA7-mCherry accumulated in these regions, coinciding with Ca²⁺ elevation (Fig. 4C-C''). Notably, TIA1-mCherry also significantly accumulated in expanded axonal regions following depolarization (Fig. 4D-D''), and both ANXA7 and TIA1 aggregation were abolished by EDTA treatment, as reflected by a reduced heterogeneity index (Fig. 4C–D''). In neurons co-expressing GCaMP6f and ANXA7-mCherry, co-localization of Ca²⁺ elevation and ANXA7-mCherry accumulation was observed (Fig. 4E; Movie EV10), confirmed by overlapping peaks in fluorescence line profiles (Fig. 4E', asterisks) and a significant increase in Pearson's coefficient (Fig. 4E''). The aggregation of ANXA7 in these Ca²⁺-elevated axonal hot spots resembled the Ca²⁺-induced ANXA7 aggregates previously seen in cancer cells (Sønder et al, 2019). To further investigate the effect of Ca²⁺ elevation on TIA1 granule trafficking, we measured the movement speed of TIA1-mCherry and ANXA7-mCherry granules, and found that 10 min after depolarization, the trafficking speed of both granules significantly decreased (Fig. 4F). Additionally, 10 min of high K⁺ depolarization led to a significant increase in both the number and size of

endogenous TIA1 and ANXA7 granules in DIV11 hippocampal neurons compared to the low K⁺-treated group (Fig. 4G,G'; Appendix Fig. S3E,E'). These results indicate that depolarization-induced persistent Ca²⁺ elevation in axonal "hot spots" promotes ANXA7 aggregation, impedes ANXA7-mediated TIA1 trafficking, and triggers TIA1 aggregation in axons.

In addition to the impact of persistent and global Ca²⁺ elevation on ANXA7 and TIA1 aggregation in axons, we also investigated whether transient and local Ca²⁺ elevation affects this process. Using a custom-designed microfluidic Axon-on-a-Chip (AoC) to apply mild, precisely controlled mechanical stress to axons, we induced a local, transient Ca²⁺ rise in the stressed axons (Pan et al, 2024; Pan et al, 2022). As shown in Fig. 4H, a microflux of culture medium (50 μL/min for 180 s) was delivered to axons of DIV8 neurons (Fig. 4H', step "② Flux"), eliciting a transient and spatially restricted Ca²⁺ increase. We confirmed that the flux-induced Ca²⁺ rise was rapid and reversible, peaking at ~2–3× baseline before returning to resting levels within 10 min (Fig. 4I,I', "After"), demonstrating the model's reproducibility. During the transient Ca²⁺ elevation, we monitored the dynamics of ANXA7-mCherry and TIA1-mCherry alongside Ca²⁺ levels. Both proteins rapidly accumulated at Ca²⁺-elevated foci (Fig. 4J, arrowheads; Movie EV11), reflected by a significant increase in the heterogeneity index (Fig. 4J', orange bars). When the elevated Ca²⁺ returned to baseline, these aggregates dispersed and the heterogeneity index returned to baseline (Fig. 4J', blue bars). These results provide spatial and temporal evidence that Ca²⁺ elevation drives ANXA7 and TIA1 condensation in axons.

## Identify the fate and RNA composition of retrograde RNPs

To specifically label axon-derived retrograde RNPs, we adapted the RNA-Select Green Fluorescent Cell Stain (RNA-Select), a cell-permeable, RNA-specific dye that rapidly stains intracellular RNA. In neurons cultured in microfluidic devices that physically separate axons from somas, we selectively applied RNA-Select to the axonal

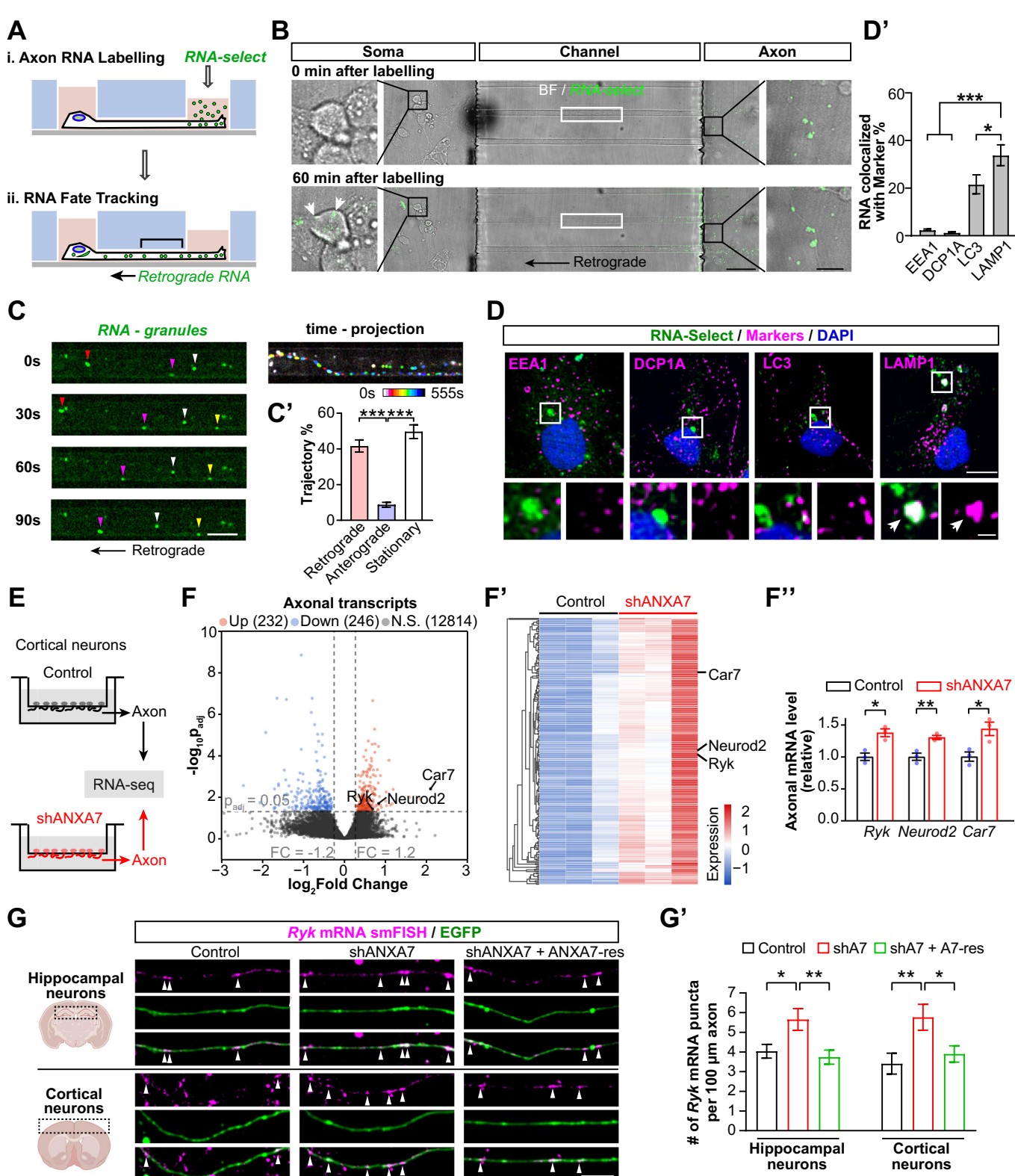

**A** i. Axon RNA Labelling

ii. RNA Fate Tracking

**B**

| Soma | Channel | Axon |

0 min after labelling

60 min after labelling

BF / *RNA-select*

Retrograde

**C** *RNA - granules*

0s / 30s / 60s / 90s

Retrograde

time - projection

0s — 555s

**C'**

Trajectory %

Retrograde / Anterograde / Stationary

*** *** ***

**D'**

RNA colocalized with Marker %

EEA1 / DCP1A / LC3 / LAMP1

*** *

**D** RNA-Select / Markers / DAPI

EEA1 / DCP1A / LC3 / LAMP1

**E**

Cortical neurons

Control → Axon

RNA-seq

shANXA7 → Axon

**F** Axonal transcripts

Up (232)  Down (246)  N.S. (12814)

$-\log_{10} p_{adj}$

$p_{adj} = 0.05$

FC = -1.2   FC = 1.2

Car7
Ryk   Neurod2

$\log_2$Fold Change

**F'**

Control    shANXA7

Car7

Neurod2
Ryk

Expression

**F''**

Axonal mRNA level (relative)

Control   shANXA7

*   **   *

Ryk   Neurod2   Car7

**G** *Ryk* mRNA smFISH / EGFP

Control / shANXA7 / shANXA7 + ANXA7-res

Hippocampal neurons

Cortical neurons

**G'**

Control   shA7   shA7 + A7-res

# of *Ryk* mRNA puncta per 100 μm axon

*   **        **   *

Hippocampal neurons / Cortical neurons

compartments to establish a two-step axonal RNA pulse-chase assay (Fig. 5A): (i) axonal RNA labeling, and (ii) tracking of labeled transcripts in both axons and somas, similar to our previously developed retrograde tracing method (Wang and Meunier, 2022). We observed that labeled RNA initially accumulated in distal axons and gradually appeared in the soma (Fig. 5B). Within axon channels, trajectories of RNA-Select-labeled RNPs were clearly visualized (Fig. 5C; Movie EV12). Tracking their movement revealed that most particles were either retrograde or stationary (Fig. 5C'), confirming their retrograde nature. To evaluate their

◄  **Figure 5.   Identify the fate and RNA composition of retrograde RNPs in axons.**

(A) Workflow of pulse-chase labeling of axon-derived retrograde RNAs in neurons cultured in microfluidic devices: (i) RNA-select dye was added to the axon terminal chambers to label axonal RNAs; (ii) the fate of axon-derived RNPs was traced in the axon and soma compartments. (B) RNA-select labeling was observed in axon terminals immediately after dye addition (top), with accumulation in soma compartments detected after 60 min (bottom). Black boxed regions are enlarged at the sides; white boxed regions were used for live imaging of axon trafficking as shown in (C). Scale bars = 50 μm (left), 10 μm (right). (C) Left: key frames from time-lapse images showing labeled retrograde RNA granules in axons, scale bar = 10 μm. Right: projection of RNA granules trajectories over 555 s, color-coded by time. (C') Quantification of transport direction of RNA granules (n = 28 axons from three biological replicates). (D) Co-localization of axon-derived RNPs with the indicated subcellular markers. Scale bars = 10 μm (top), 2 μm (bottom). (D') Quantification of (D) (n = 24, 44, 26, 41 neurons from three biological replicates). (E) Workflow for isolating and analysing axonal RNA transcripts affected by ANXA7 knockdown in cortical neurons cultured in Boyden chambers. (F) Volcano plot of axonal RNA transcripts of ANXA7 knockdown (shANXA7-1#) neurons versus control neurons. Colored dots indicate significantly upregulated (red, adjusted P < 0.05 and FC >1.2) and downregulated (blue, adjusted P < 0.05 and FC <−1.2) transcripts. Differential expression was calculated using DESeq2; P values were adjusted by Benjamini–Hochberg false discovery rate (FDR) method. (F') Clustering of RNA transcripts increased in axons of ANXA7 knockdown neurons compared to controls. Heatmap shows relative expression levels across datasets. Three indicated genes were selected for validation. (F″) RT-qPCR analysis of Ryk, Neurod2, and Car7 mRNA levels in RNA isolated from axons of ANXA7 knockdown (shANXA7-1#) versus control cortical neurons cultured in Boyden chambers (n = 3 independent biological replicates). (G) Representative images of Ryk mRNA smFISH puncta in axons of DIV12 rat hippocampal (top) and cortical (bottom) neurons with indicated ANXA7 levels. The shRNA used was shANXA7-1#. Arrowheads indicate Ryk mRNA smFISH puncta. Scale bar = 10 μm. (G') Quantification of (G) (For Hippocampal neurons, n = 55, 55, 55 axons; for Cortical neurons, n = 37, 35, 37 axons, both from three biological replicates). Data represent mean ± SEM; two-tailed unpaired t-test in (F″); one-way ANOVA in (C', D', G'). *P < 0.05, **P < 0.01, ***P < 0.001. See appendix for exact P values. Source data are available online for this figure.

fate, neurons were fixed 1 h post-labeling, followed by co-localization analysis in the soma. The majority of retrogradely transported RNA localized to degradative compartments, with 33.8% colocalizing with lysosomes (LAMP1) and 21.6% with autophagosomes (LC3), while only a small fraction overlapped with early endosomes (2.5% EEA1) or processing bodies (1.4% DCP1A) (Fig. 5D,D'). These findings suggest a "garbage disposal" role for axon-derived RNA transported to the soma.

To further characterize the mRNA composition of retrograde RNPs in axons, we cultured ANXA7 knockdown and control neurons in Boyden chambers (Fig. 5E) to isolate and extract RNA from axons, as described in (Doron-Mandel et al, 2021). We then sequenced and compared the axonal mRNAs from both groups (Fig. 5F,F'), identifying 232 transcripts with significantly increased abundance after ANXA7 knockdown. We validated three transcripts with known neuronal functions, Ryk, Neurod2, and Car7, by reverse transcription-quantitative PCR (RT-qPCR) (Fig. 5F″). Since Ryk mRNA had previously been identified as a cargo of TIA1 (ENCODE: ENCSR057DWB and ENCSR623VEQ) and encodes a receptor for Wnt5a that mediates repulsion of corticospinal tract axons (Duan et al, 2017; Hollis et al, 2016), we focused on Ryk mRNA. We next examined its abundance in axons of cultured hippocampal and cortical neurons using single molecule FISH (smFISH) (Femino et al, 1998; Jin et al, 2025; Raj et al, 2008) and found that, in axons of both neuron types, knockdown of ANXA7 significantly increased the density of Ryk mRNA puncta compared to controls (Fig. 5G,G'). Expression of ANXA7-res restored axonal Ryk mRNA levels to those of controls, confirming the role of ANXA7 in regulating Ryk mRNA axonal abundance.

Together, these results indicate that ANXA7-dependent retrograde axon trafficking delivers specific axon-derived mRNAs, such as Ryk, to degradative compartments in the soma, contributing to the dynamic regulation of axon growth in response to morphogens.

## ANXA7-mediated TIA1 trafficking is crucial for maintaining axon integrity

With evidence indicating that ANXA7-mediated retrograde trafficking of RNPs contributes to the degradation, we next investigated whether manipulating this mechanism affects axon

health, we conducted gain- and loss-of-function experiments in DIV12 cultured hippocampal neurons (HNs) and cortical upper motor neurons (UMNs, identified by CTIP2-positive staining; Fig. EV4A). In both HNs (Fig. 6A, top) and UMNs (Fig. 6A, bottom; Appendix Fig. S4A), altering ANXA7 expression significantly affected the extent of TIA1 granule aggregation within axons. Overexpression reduced the accumulation of large granules (≥2 μm²) (Figs. 6A, arrows and EV4B, boxed columns), decreasing both their percentage (Fig. EV4B) and density (Fig. 6A'). In contrast, knockdown of endogenous ANXA7 using shANXA7 led to a significant increase in large TIA1 granules within axons (Fig. 6A, arrows), elevating both their percentage (Fig. EV4B) and density (Fig. 6A'). Notably, overexpression of an shRNA-resistant ANXA7 variant (shA7 + A7-res) reversed the phenotype caused by ANXA7 knockdown, reducing the size of TIA1 granules (Figs. EV4B and 6A,A'). Consistent results were observed when examining co-expressed TIA1-mCherry granules, showing a reduced number of large granules in ANXA7-overexpressing HNs, and an increase in large TIA1-mCherry puncta upon ANXA7 knockdown (Fig. EV4C,C'). These findings indicate that ANXA7 is critical for preventing TIA1 aggregation within axons.

Next, we determined how ANXA7 prevents the formation of large TIA1 aggregates in axons of HNs. Since phase separation capacity is closely related to the molecular mobility of RNA-binding proteins (Shin and Brangwynne, 2017), we assessed whether ANXA7 modulates the overall mobility of TIA1 molecules in axons by measuring fluorescence recovery after photobleaching (FRAP) of EGFP-TIA1 along long axonal segments (~30 μm) (Fig. 6B; Movie EV13). FRAP rates were enhanced in ANXA7-overexpressing axons but reduced in ANXA7 knockdown axons (Fig. 6B'). The FRAP efficiency across long axonal segments reflects the combined effects of trafficking- and diffusion-dependent mobility of TIA1 molecules within the axon. To distinguish the contributions of these two mechanisms, we first evaluated TIA1 granule trafficking by automatically tracking TIA1-mCherry granules and comparing their average speed along axons (Fig. 6C; Movie EV14). ANXA7 knockdown significantly increased the ratio of immobile TIA1 granules (Fig. EV4D, blue shaded), whereas ANXA7 overexpression resulted in a higher ratio of mobile granules (Fig. EV4D, red shaded). The average speed of retrograde

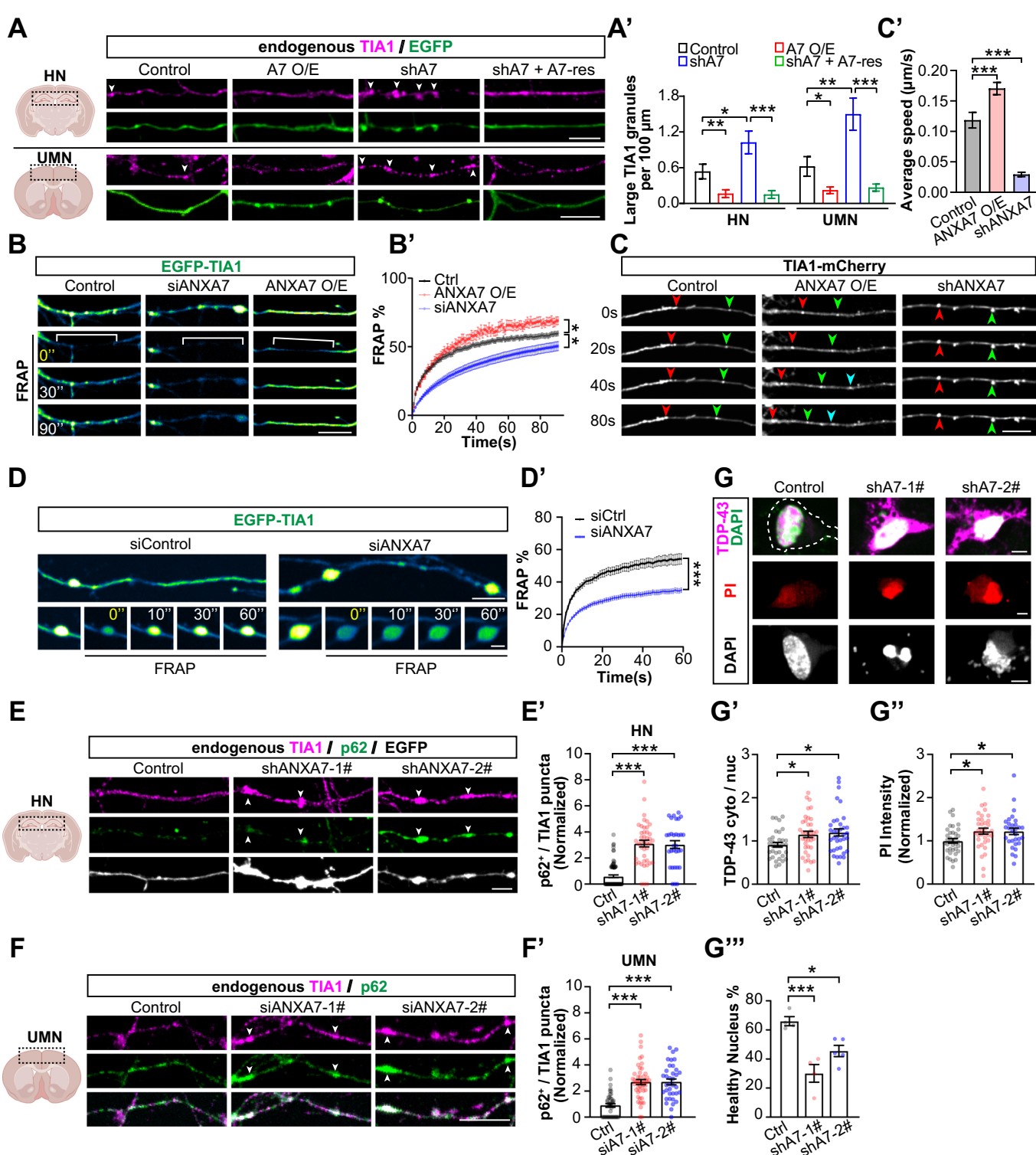

TIA1 granules was faster in ANXA7-overexpressing axons and slower in those with reduced ANXA7 (Fig. 6C'). Similarly, the trafficking efficiency of light-induced Opto-TIA1 granules was significantly reduced in ANXA7 knockdown neurons (Fig. EV4E,E'), although their formation capacity remained unaffected

(Fig. EV4E,E''). These results suggest that ANXA7 facilitates active axonal trafficking of TIA1 granules.

We then examined the mobility of TIA1 molecules within the large, immobile granules frequently observed in ANXA7 knockdown HNs, using FRAP assay (Fig. 6D; Movie EV15), and found

**Figure 6.   ANXA7 is essential for TIA1 axon trafficking and phase separation in neurons.**

(A) Representative images of TIA1 granules in axons of DIV12 rat hippocampal neuron (HN, top) and cortical upper motor neuron (UMN, bottom) under different ANXA7 expression levels. Arrowheads indicate TIA1 granules. Scale bar = 10 μm. (A') Quantification of axonal density of large TIA1 granules (≥ 2 μm², circularity 0.6–1) from (A) (for HN, $n$ = 47, 46, 47, 44 neurons; for UMN, $n$ = 61, 53, 51, 55 neurons, both from three biological replicates). (B) Key frames from time-lapse images of EGFP-TIA1 FRAP in axons. Brackets indicate photobleached segments. Scale bar = 10 μm. (B') Quantification of FRAP curves from (B) ($n$ = 90, 27, 69 axons from three biological replicates). (C) Key frames from time-lapse images showing retrograde trafficking of EGFP-TIA1 granules. Different colored arrowheads denote distinct granules. Scale bar = 10 μm. (C') Average speed of retrograde EGFP-TIA1 granules from (C) ($n$ = 244, 238, 236 granules from three biological replicates). (D) Key frames from time-lapse images of FRAP for stationary EGFP-TIA1 granules in axons. Scale bars = 15 μm (top), 5 μm (bottom). (D') Quantification of FRAP curves from (D) ($n$ = 160, 199 granules from three biological replicates). (E) Distribution of endogenous TIA1 with p62 in axon of HNs. EGFP depicts axon morphology. Scale bar = 10 μm. (E') Quantification from (E) showing the number of TIA1 puncta co-localized with p62 per 100-μm axon. ($n$ = 79, 42, 31 neurons from three biological replicates). (F) Distribution of endogenous TIA1 with p62 in axons of UMN. Scale bar = 10 μm. (F') Quantification from (F) showing the number of TIA1 puncta co-localized with p62 per 100-μm axon. ($n$ = 43, 47, 36 neurons from three biological replicates). (G) Top: distribution of endogenous TIA1 and TDP-43 in the cytoplasm and nucleus of neurons of indicated groups. Dashed line in "Control" outlines soma morphology. Middle: propidium iodide (PI) staining identifying dead cells. Bottom: DAPI staining showing nuclear morphology in DIV9 HN of indicated groups. All scale bars = 5 μm. (G'–G‴) Quantification of (G): TIA1/TDP-43 localization (G') ($n$ = 33, 37, 36 neurons from three biological replicates), PI intensity (G″) ($n$ = 36, 35, 30 neurons from three biological replicates), and healthy nucleus percentage (G‴) ($n$ = 4, 4, 5, $n$ representing independent biological replicates). Data represent mean ± SEM; two-tailed unpaired $t$-test in (A', D'); one-way ANOVA in (B', C', E', F', G', G″, G‴); *$P$ < 0.05, **$P$ < 0.01, ***$P$ < 0.001. See appendix for exact $P$ values. Source data are available online for this figure.

that EGFP-TIA1 mobility within these granules was dramatically reduced compared to control neurons (Fig. 6D'). This indicates that these larger, immobile TIA1 granules, induced by ANXA7 downregulation, resemble condensed aggregates. Notably, axons of HNs with ANXA7 knockdown using two distinct shRNA sequences (shANXA7-1# and shANXA7-2#) exhibited a significant increase in TIA1 aggregates (Fig. EV4F,F'), along with multiple axonal swellings and bead-like morphologies (Fig. EV4F, arrowheads; EV4F″), indicative of focal axonal swellings (FAS), a hallmark of axonopathy (Coleman, 2005; Geula et al, 2008; Nikić et al, 2011). IF staining revealed that TIA1 aggregates in FAS regions significantly overlap with SQSTM1/p62 and TDP-43, which are markers of pathological aggregations (Fig. 6E,E'; Fig. EV4G,G') (Gao et al, 2018; Ling et al, 2013; Pankiv et al, 2007). Similar pathological phenotypes were also observed along the axons of CTIP2-positive UMNs (Fig. 6F,F'; Appendix Fig. S4B). Additionally, ANXA7 downregulation significantly increased the cytoplasmic distribution of TDP-43 from the nucleus (Fig. 6G,G') and resulted in a higher proportion of HNs with unhealthy nuclei (Fig. 6G,G″,G‴), suggesting exacerbated neurodegeneration in the absence of ANXA7.

Collectively, these findings from primary HNs and UMNs demonstrate that the loss of ANXA7-mediated TIA1 axon trafficking promotes the aggregation of large TIA1 granules, leading to axonopathy and neurodegeneration. This underscores the crucial role of ANXA7 in maintaining axon integrity by ensuring the proper recruitment of TIA1 granules to dynein.

## ANXA7 downregulation induces neurodegeneration in the mouse motor cortex

To further explore the in vivo function of the ANXA7-mediated mechanism in axon integrity, we downregulated ANXA7 expression in neurons of the motor cortex of neonatal mice by intracerebroventricular injection (ICV) of two effective shRNA sequences (3# or 4#) (Fig. EV5A) delivered via AAV-U6-shANXA7-hSyn-EGFP at postnatal day 1 (P1) neonatal mice. To rule out off-target effects, we performed rescue experiments by co-expressing an shRNA-resistant rat ANXA7 homolog with shANXA7-4#, effectively restoring ANXA7 expression in targeted neurons (Fig. EV5B). AAV-hSyn-EGFP alone served as control

(Fig. 7A). Then, as illustrated in Fig. 7B, from P56 to P58, mice underwent three rotarod training sessions, followed by a latency-to-fall test on P59, before being sacrificed on P60. We found that ANXA7 knockdown mice exhibited significantly impaired motor capacity, evidenced by shorter latency to fall on the rotarod compared to control mice. This deficit was rescued by co-expression of ANXA7-res (Fig. 7B'). Motor information was delivered from the UMNs in layer V of the motor cortex to spinal motor neurons via the long-range axon projections in corticospinal tract (CST), located laterally in the spinal cord (Fig. 7C). We observed that ANXA7 knockdown led to a thinner layer V in the motor cortex (M1 and M2 regions (Franklin and Paxinos, 2001)), which was rescued by co-expression of ANXA7-res (Fig. 7D,D'), suggesting loss of neurons. Morphological analysis of axons in the lateral CST revealed a higher percentage of projecting axons exhibiting typical FAS morphology (Fig. 7E,E'), indicative of axon degeneration. This degenerative phenotype was rescued by co-expression of ANXA7-res (Fig. 7E,E'). We also examined TIA1 distribution in the soma of ANXA7-knockdown neurons in layer V, finding increased cytosolic TIA1 aggregates co-localized with SQSTM1/p62 and TDP-43 (Figs. 7F,F'; EV5C,C'). Notably, ANXA7-res significantly reduced these pathological co-aggregates of TIA1 and SQSTM1/p62 (Fig. 7F,F'), further supporting a role for ANXA7 in preventing aberrant TIA1 aggregation.

Furthermore, we found that in the motor cortex of ANXA7 knockdown mice, the number of activated microglia, indicated by the density of Iba1⁺ cells, was significantly increased compared to controls (Fig. 7G,G'), reflecting the neurodegeneration in these regions (Lull and Block, 2010). This microglial activation was markedly reduced by co-expression of ANXA7-res (Fig. 7G,G'), further supporting ANXA7's role in preventing neurodegeneration. These results demonstrate that ANXA7 knockdown in the mouse motor cortex leads to aberrant TIA1 aggregation in neurons, resulting in axonopathy and neurodegeneration in vivo. Importantly, restoring ANXA7 expression effectively rescues these pathological phenotypes, reinforcing the conclusion that ANXA7-mediated trafficking of TIA1 granules is crucial for maintaining the integrity of long projection neurons in the CNS.

In conclusion, our study uncovers a dynein-driven mechanism that mediates the retrograde axon transport of TIA1-containing RNPs to degradative compartments in CNS neurons, a process

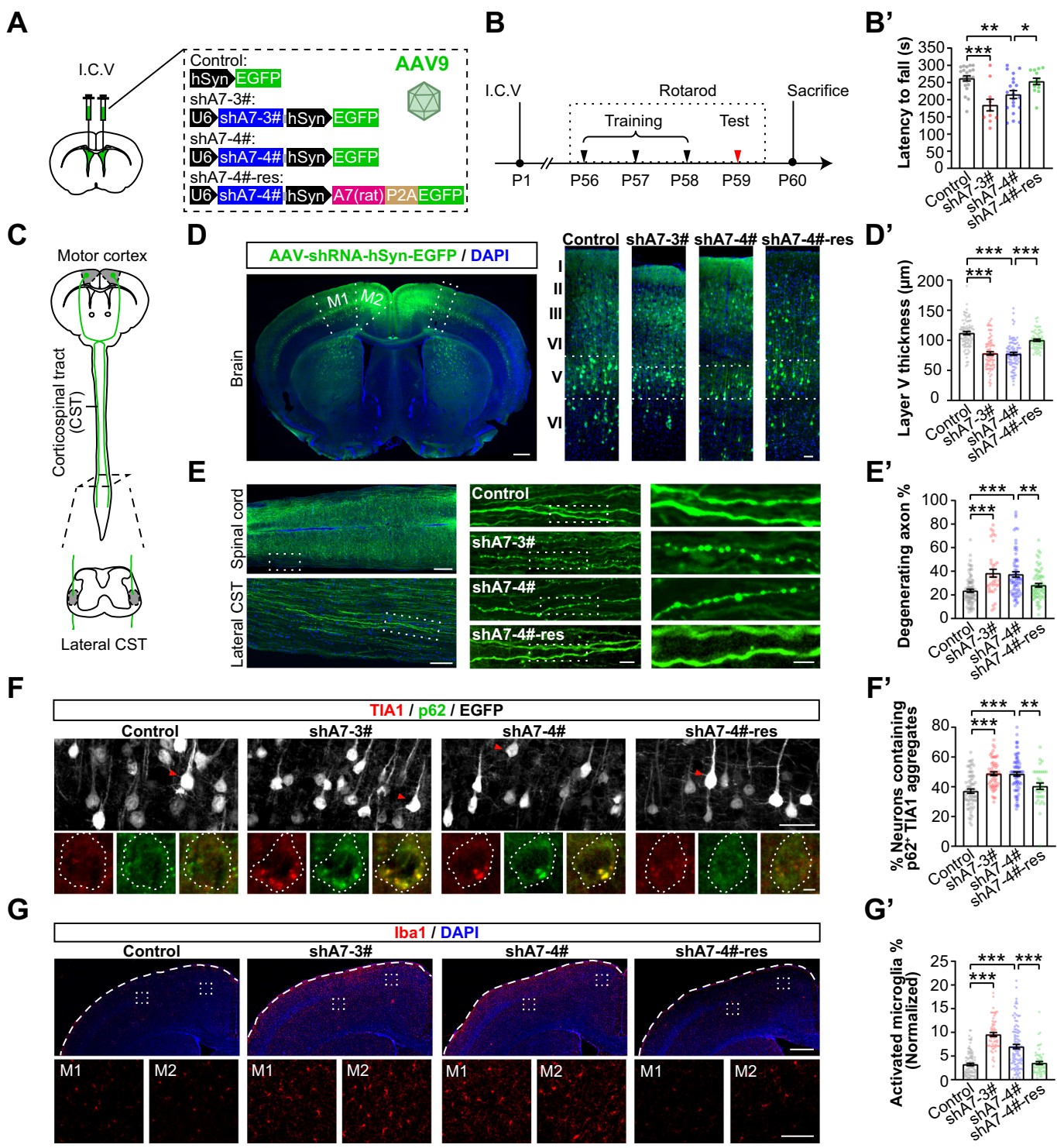

crucial for maintaining axon health. As illustrated in Fig. 8, ANXA7 serves as a pivotal enhancer that promotes the recruitment of TIA1 granules to dynein, facilitating their transport to lysosomes in the soma. Axon Ca²⁺ overload disrupts this interaction, leading to the detachment of ANXA7 and TIA1 granules from dynein and resulting in pathological TIA1 aggregation within focal axonal

regions. Similarly, knockdown of ANXA7 uncouples TIA1 granules from dynein, impairing their axon trafficking and causing aberrant aggregation, which ultimately triggers axonopathy and neurodegeneration. Conversely, ANXA7 overexpression enhances the axon trafficking efficiency, reduces TIA1 aggregates, and restoring ANXA7 expression in knockdown neurons rescues

**Figure 7.  Downregulation of ANXA7 causes neurodegeneration in mice motor cortex.**

(A) Diagram of intra-cerebroventricular (ICV) injection sites in P1 mouse pups for delivering shANXA7 (shA7-3# and 4#) or shA7-4#-res. (B) Experimental timeline. P1: ICV of AAVs; P56-P59: Rotarod training and test; P60: sacrifice and tissue IF staining. (B') Rotarod probe test results showing latency to fall (n = 22, 11, 21, 13 mice). (C) Diagram of upper motor neuron projection pathways, showing somas in cortical layer V and descending axons in the corticospinal tract (CST). (D) Confocal images of P60 mouse motor cortex, with the M1 and M2 indicated in the right panel and thickness of layer V marked between dashed lines in the left panels. Scale bars = 500 μm (left), 50 μm (right). (D') Quantification of layer V thickness in the cortex (n = 96, 83, 92, 62 ROIs from 5, 3, 5, 3 mice). (E) Confocal images of lateral CST in P60 mouse spinal cord showing descending axons of infected cortical neurons. Magnified boxed regions detail individual axon morphology. Scale bars = 500 μm (left top), 100 μm (left bottom), 20 μm (middle), and 10 μm (right). (E') Percentage of degenerating axons with beading morphology (n = 77, 36, 71, 46 ROIs, from 5, 3, 5, 3 mice). (F) Confocal images of P60 mouse cortex showing TIA1 and p62 IF in layer V neurons. Infected neurons marked by EGFP expression, the neurons pointed by red arrows are amplified in lower pannels. Dotted lines depict soma shapes. Scale bars = 50 μm (top), 5 μm (bottom). (F') Percentage of neurons containing p62$^+$ TIA1 aggregates (n = 69, 55, 74, 34 ROIs, from 5, 3, 5, 3 mice). (G) Confocal images of P60 mouse motor cortex showing the density of activated microglia, detected by Iba1 staining. Boxed areas are amplified in the bottom panels. Scale bars = 500 μm (top) and 100 μm (bottom). (G') Percentage of Iba1$^+$ activated microglia (n = 100, 60, 100, 60 ROIs, from 5, 3, 5, 3 mice). Data represent mean ± SEM; one-way ANOVA in (B', D', E', F', G'); *P < 0.05, **P < 0.01, ***P < 0.001. See appendix for exact P values. Source data are available online for this figure.

neurodegenerative phenotypes in mice, highlighting the therapeutic potential of targeting this pathway to clear pathological aggregates and alleviating neurodegeneration.

## Discussion

The directed axon trafficking system delivers RNPs to meet the dynamic demands for proteins and mRNAs in polarized, extended neurons (Abouward and Schiavo, 2021; Abraham and Fainzilber, 2022; Müntjes et al, 2021). Some of these RNPs contain RBPs with PrLDs that are prone to forming toxic fibrils, leading to neurodegeneration (Apicco et al, 2018; Ash et al, 2021). Not surprisingly, defects in axon trafficking closely associate with the abnormal aggregation of these fibril-forming RBPs (Fernandopulle et al, 2021; Sleigh et al, 2019). Nevertheless, it remains unknown whether the axon trafficking of RNPs plays a direct role in preventing RBP aggregation in axons, which is an early indicator and causal factor of neurodegeneration. In this study, we reveal that ANXA7 enhances a dynein-driven RNP transport mechanism related with the degradation of these RNPs, thereby counteracting TIA1 aggregation in axons and providing a potential strategy to target and eliminate pathogenic aggregates underlying neurodegenerative diseases.

### ANXA7 promotes the recruitment of TIA1-containing RNPs to dynein

Two main mechanisms underlie the bidirectional trafficking of RNPs: the indirect mechanism, in which RNPs are tethered to membranous organelles such as lysosomes (Liao et al, 2019) and endosomes (Baumann et al, 2012; Cioni et al, 2019), and the direct mechanism, where the RBP components of RNPs are linked directly to motors via adapter proteins. For instance, adenomatous polyposis coli (APC) binds kinesin-associated protein 3 (KAP3), an adapter for the anterograde motor kinesin-2 (Baumann et al, 2020); ZBP1/PAT1 links β-actin mRNA to kinesin-1 (Wu et al, 2020); nucleolin–GAR motifs engage multiple kinesins (Doron-Mandel et al, 2021); and splicing factor proline/glutamine-rich (SFPQ) binds kinesin light chain 1 (KLC1) of kinesin family member 5A (KIF5A) (Fukuda et al, 2021). Particularly, recent advances in RNP transport in axons were summarized in (Abraham and Fainzilber, 2022). However, although the retrograde motor dynein has been

shown to drive RBP trafficking in cultured *Drosophila* S2 cells or embryo lysates (McClintock et al, 2018; Sladewski et al, 2018), the mechanism underlying dynein-driven axon transport of RNPs in mammalian neurons remains elusive.

In neurons expressing EGFP-TIA1 and RNA labeled with CY5-UTP, we unexpectedly observed that, unlike the entire pool of CY5-UTP-labeled RNPs, which exhibited bidirectional transport, TIA1-containing RNPs predominantly moved in a retrograde direction (Fig. 1B,B'). This finding was validated in unidirectional axons of neurons cultured in microfluidic devices (Fig. 1C,D''). This retrograde bias suggested a link to dynein, the motor driving retrograde transport from axon tips to the soma (Cason and Holzbaur, 2022). Through mass spectrometry of TIA1 interactors from mouse brain lysates, we identified ANXA7, a Ca$^{2+}$-regulated protein, that interacts with both TIA1 and the dynein subunit DIC1B. The absence of DIC1B from the GST–TIA1 interactome likely reflects that the TIA1-DIC1B interaction depends heavily on ANXA7 acting as an affinity enhancer. At endogenous ANXA7 levels, this association is too transient or weak to allow efficient retention of DIC1B on GST–TIA1 beads, causing it to fall below the detection threshold of mass spectrometry. Whereas live-cell confocal microscopy and SIM revealed significant co-localization and co-transportation of TIA1, ANXA7, and dynein in primary neurons. Further, co-IP and in vitro pull-down experiments demonstrated that ANXA7 markedly strengthens the interaction between TIA1 and DIC1B. Moreover, FLIM-FRET experiments in live neurons showed ANXA7 expression controls their interaction. Finally, PLA assay confirmed that the interaction between endogenous TIA1 and dynein is controlled by ANXA7 levels in neurons.

Previous studies found that TIA1 binds to 3'UTRs of mRNAs to repress translation in various cell types (Díaz-Muñoz et al, 2017; Dixon et al, 2003; López de Silanes et al, 2005; Piecyk et al, 2000), yet the function of retrograde TIA1-RNPs within axons remained unclear. Using an axonal RNA pulse-chase assay in microfluidic devices, we demonstrated that these axon-derived RNPs are actively transported back to the soma and predominantly sorted to degradative compartments such as lysosomes and autophagosomes, suggesting a clearance pathway for axonal RNPs. Additionally, transcriptomic profiling of RNA enriched in axons of ANXA7-knockdown neurons revealed selective accumulation of specific

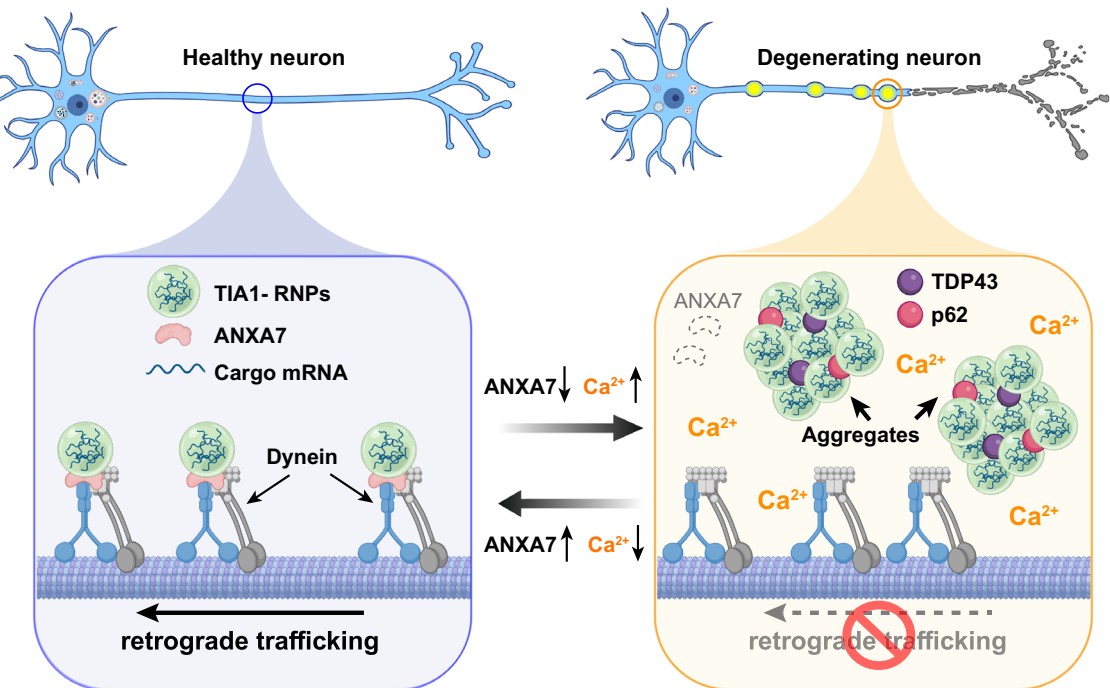

**Figure 8. ANXA7 enhances TIA1 axon transport to counteract pathological aggregation in neurons.**

Left: ANXA7 promotes the recruitment of TIA1 granules to dynein, facilitating their retrograde axon trafficking. By delivering RNPs back to the soma for degradation, this mechanism is essential for maintaining axon integrity and function. Right: Focal $Ca^{2+}$ elevation or ANXA7 knockdown disrupts ANXA7's function as an affinity enhancer, causing TIA1-RNPs to detach from dynein and accumulate in focal axonal regions where $Ca^{2+}$ surges persist. These pathological aggregates drive axonopathy and neurodegeneration both in vitro and in vivo. Conversely, restoring ANXA7 expression rescues transport defects and prevents aberrant TIA1 aggregation, highlighting the therapeutic potential of targeting this pathway in neurodegenerative diseases.

mRNAs, notably *Ryk*, which is a known TIA1 cargo encoding a Wnt5a receptor essential for corticospinal tract guidance (Duan et al, 2017; Hollis et al, 2016). Elevated *Ryk* abundance in axons of ANXA7-knockdown neurons was confirmed by RT-qPCR and smFISH, and reversed by ANXA7 re-expression, highlighting ANXA7's crucial role in the retrograde clearance of TIA1-associated transcripts.Together, these findings indicate that ANXA7-mediated retrograde transport dynamically regulates turnover of specific RNPs in axons, unveiling a critical mechanism linking RNP trafficking to neuronal homeostasis and health (Abraham and Fainzilber, 2022; Jung et al, 2023).

## $Ca^{2+}$ overload inhibits the ANXA7-mediated recruitment of TIA1 to dynein

With an N-terminal proline-rich LCD domain, the ANXA7 undergoes LLPS, which is triggered by millimolar of $Ca^{2+}$ elevation in cancer cells (Yu et al, 2023). Meanwhile ANXA7 is a $Ca^{2+}$-sensitive protein with C-terminal Annexin repeats, which mediates its $Ca^{2+}$-triggered phospholipid binding to the plasma membrane (PM) (Gerke et al, 2024; Sønder et al, 2019; Yu et al, 2023). Highly expressed in neurons, ANXA7 is known as a positive regulator of synaptic vesicle release and postsynaptic *N*-methyl-D-aspartate (NMDA) receptor trafficking (Li et al, 2018). But whether AXNA7 plays any role in axon trafficking, and whether such function is under $Ca^{2+}$ regulation, remains unknown. In axons of cultured

neurons, using two different live-cell models, we found that both the persistent, whole cell $Ca^{2+}$-elevation induced by depolarization and the transient, axon-specific $Ca^{2+}$-elevation induced by mild mechanical stress could cause localized $Ca^{2+}$ rises in axonal "hot spots," which co-localized with focal aggregation of ANXA7 (Fig. 4A–G'). Such $Ca^{2+}$-induced aggregation is distinct from the $Ca^{2+}$-triggered responses of ANXA11, which mediates $Ca^{2+}$-dependent tethering of RNPs to lysosomes (Liao et al, 2019). In contrast, ANXA7 responds to $Ca^{2+}$ surges by rapidly forming droplets that are recruited to the ruptured plasma membrane (PM). This recruitment facilitates the ESCRT III-dependent repair process in cancer cells (Sønder et al, 2019). Our observation of $Ca^{2+}$-induced ANXA7 focal aggregation aligns with these findings, showing focal aggregation near the PM in axons. However, further research is needed to explore the specific functions of these $Ca^{2+}$-induced ANXA7 aggregates in axons.

Under pathological conditions, local $Ca^{2+}$ levels can transiently rise to millimolar concentrations, similar to extracellular $Ca^{2+}$ levels, likely due to the rupture of the plasma membrane (Sønder et al, 2019). To model the acute effects of such localized $Ca^{2+}$ surges on the LLPS of ANXA family proteins, in vitro experiments commonly apply millimolar $Ca^{2+}$ concentrations (Liao et al, 2019; Sønder et al, 2019; Yu et al, 2023). Guided by the $Ca^{2+}$ ranges used in these studies, we examined the impact of $Ca^{2+}$ elevations (0–10 mM) on ANXA7 phase separation. We found that $Ca^{2+}$ elevation enhanced the LLPS of ANXA7, leading to the formation of ANXA7 droplets,

which engaged the TIA1 into themselves. Significantly, employing a confocal microscopy-based in vitro approach, we found that $Ca^{2+}$ causes the detachment of small $ANXA7^+/TIA1^+$ droplets from the DIC1B-coated on beads, and forming large aggregate-like sediments at the bottom of the dish. Consistently, by experiments in neurons, axon trafficking is found to be dramatically inhibited, leading to axonal aggregation of both TIA1 and ANXA7. However, since the precise domain involved in the interaction between ANXA7 and DIC1B is not yet identified, the exact mechanism behind their detachment remains unclear. Moreover, knockdown of endogenous ANXA7 leads to aggregation of TIA1 in axons, which in turn causes the axonopathy in CST of mice spinal cord and neurodegeneration. Importantly, restoring ANXA7 expression via a resistant rat ANXA7 variant rescued these in vivo phenotypes. This finding is consistent with the established role of TIA1 granules in promoting degeneration in Tau P301S mice (Apicco et al, 2018; Ash et al, 2021) and their regeneration-suppressing effects in both worm (Andrusiak et al, 2019) and rodent neurons (Sahoo et al, 2018), supporting the notion that TIA1 aggregation is pathogenic for axon health. Therefore, we detected not only a axon trafficking mechanism of RNP, but also a $Ca^{2+}$ overload-triggered pathological mechanism underlying axonopathy.

## Upregulation of ANXA7 represses TIA1 aggregates in axons

In living axons, we found that increasing the level of ANXA7 enhances the dynamics of axonal TIA1 granules. Conversely, knocking down ANXA7 results in more immobile and condensed TIA1 droplets, underscoring the crucial role of ANXA7 in controlling the LLPS dynamics of TIA1 granules in axons. Consistently, knockdown of endogenous ANXA7 leads to pathological TIA1 aggregates, which, in turn, cause axonopathy and neurodegeneration both in vitro and in vivo. Protein aggregation within axons is recognized as a pathogenic driver of neurodegenerative diseases, including ALS, FTD, and WDM (Chiti and Dobson, 2017; Luan et al, 2024; Ross and Poirier, 2004). Specifically, mutations in *TIA1* have been linked to WDM (Hackman et al, 2013), FTD, and ALS (Gu et al, 2018; Mackenzie et al, 2017; Yuan et al, 2018), and multisystem proteinopathy (MSP) (Lee et al, 2018). Noticeably, all of these pathological *TIA1* mutations are located in its PrLD, which facilitates the formation of toxic TIA1 aggregates (Lee et al, 2018; Mackenzie et al, 2017; Sekiyama et al, 2022). Additionally, LLPS of wild-type TIA1 has been found to promote the phase separation and toxic oligomerisation of tau, exacerbating tauopathies (Apicco et al, 2018). Reducing TIA1 levels can inhibit the accumulation of tau oligomers and improve neuronal survival in tauopathy mouse models (Apicco et al, 2018; Ash et al, 2021). In line with this, a recent study reported that TIA1 is upregulated in ALS patients and that knockdown of TIA1 alleviates neurodegeneration in C9orf72 mutant ALS mice (Wei et al, 2025). These works further underscore the pathogenic role of TIA1 aggregates in neurodegenerative diseases.

Therefore, our finding that the overexpression of ANXA7 alleviates the formation of TIA1 aggregates is highly promising. This suggests that boosting ANXA7 levels could represent a potentially effective therapeutic strategy for treating TIA1 aggregation-related neurodegenerative diseases.

# Methods

## Reagents and tools table

| Reagent/resource | Reference or source | Identifier or catalog number |
| --- | --- | --- |
| **Experimental models** | | |
| C57BL/6J (*M. musculus*) | Shanghai Jihui Laboratory Animal Care | C57BL/6JShjh |
| Sprague-Dawley (*Rattus norvegicus*) | Shanghai Jihui Laboratory Animal Care | Shjh:SD |
| HEK293T | ATCC | |
| **Recombinant DNA** | | |
| Lifeact-GFP | Gift from Prof. Roland Wedlich Soldner | |
| DIC1B-mRFP | Gift from Prof. K. Kevin Pfister | |
| pAAV-hSyn-EGFP, pHelper, pPHP.S | Gifts from Prof. Zhenge Luo | |
| DNA encoding TIA1 | Gift from Prof. Yichang Jia | |
| DNA encoding Cry2 | Gift from Prof. Hanhui Ma | |
| pGP-CMV-GCaMP6f | Addgene | Cat# 40755 |
| CMV-TIA1-mCherry | This study | |
| pRK5-HA-TIA1 | This study | |
| CMV-EGFP-TIA1 | This study | |
| All ANXA7-related plasmids | This study | |
| **Antibodies** | | |
| TIA1 Antibody (G-3) | Santa Cruz | Cat#sc-166247; RRID: AB_2201545 |
| TIA1 Polyclonal antibody | Proteintech | Cat#12133-2-AP; RRID: AB_2201427 |
| Annexin VII (A-1) | Santa Cruz | Cat# sc-17815; RRID: AB_626681 |
| Annexin A7 Polyclonal antibody | Proteintech | Cat#10154-2-AP; RRID: AB_2227386 |
| DYNC1I1 Polyclonal antibody | Proteintech | Cat#13808-1-AP; RRID: AB_2093492 |
| HA-Tag (C29F4) Rabbit mAb | Cell Signaling Technology | Cat#3724; RRID: AB_1549585 |
| MYC tag Monoclonal antibody | Proteintech | Cat#60003-2-Ig; RRID: AB_2734122 |
| MYC tag Polyclonal antibody | Proteintech | Cat#16286-1-AP; RRID: AB_11182162 |
| DYKDDDDK tag Polyclonal antibody | Proteintech | Cat#20543-1-AP; RRID: AB_11232216 |
| G3BP1 Antibody (H-10) | Santa Cruz | Cat#sc-365338; RRID: AB_10846950 |

| Reagent/resource | Reference or source | Identifier or catalog number |
|---|---|---|
| SQSTM1/p62 Rabbit pAb | ABclonal | Cat#A11247; RRID: AB_2758476 |
| Anti-β-Tubulin III Antibody | Sigma-Aldrich | Cat#AB9354; RRID: AB_570918 |
| Rab5 (C8B1) Rabbit mAb | Cell Signaling Technology | Cat#3547; RRID: AB_2300649 |
| LC3B (E5Q2K) Mouse mAb | Cell Signaling Technology | Cat#83506; RRID: AB_2800018 |
| LC3B (D11) XP Rabbit mAb | Cell Signaling Technology | Cat#3868; RRID: AB_2137707 |
| Anti-LAMP1 antibody [LY1C6] | Abcam | Cat#ab13523; RRID: AB_300425 |
| LAMP1 (D2D11) XP Rabbit mAb | Cell Signaling Technology | Cat#9091; RRID: AB_2687579 |
| EEA1 (C45B10) Rabbit mAb | Cell Signaling Technology | Cat#3288; RRID: AB_2096811 |
| DCP1A Polyclonal antibody | Proteintech | Cat#22373-1-AP; RRID: AB_2879092 |
| TDP-43 (D9R3L) Rabbit mAb | Cell Signaling Technology | Cat#89789; RRID: AB_2800143 |
| Anti Iba1, Rabbit (for Immunocytochemistry) | FUJIFILM | Cat#019-19741; RRID: AB_839504 |
| STAU1 Polyclonal antibody | Proteintech | Cat#14225-1-AP; RRID: AB_2302744 |
| FMRP Antibody | Cell Signaling Technology | Cat#4317; RRID: AB_1903978 |
| Bcl-11B (D6F1) XP Rabbit mAb | Cell Signaling Technology | Cat#12120; RRID: AB_2797823 |
| anti-p62/ SQSTM1 (C-terminus) guinea pig polyclonal, serum | PROGEN | Cat#GP62-C; RRID: AB_2687531 |
| Goat anti-Mouse IgG (H + L) Cross-Adsorbed Secondary Antibody, Alexa Fluor 488 | Invitrogen | Cat#A-11001; RRID: AB_2534069 |
| Goat anti-Mouse IgG (H + L) Highly Cross-Adsorbed Secondary Antibody, Alexa Fluor 568 | Invitrogen | Cat#A-11031; RRID: AB_144696 |
| Goat anti-Mouse IgG (H + L) Highly Cross-Adsorbed Secondary Antibody, Alexa Fluor Plus 647 | Invitrogen | Cat#A32728; RRID: AB_2633277 |

| Reagent/resource | Reference or source | Identifier or catalog number |
|---|---|---|
| Goat anti-Rabbit IgG (H + L) Highly Cross-Adsorbed Secondary Antibody, Alexa Fluor Plus 488 | Invitrogen | Cat#A32731; RRID: AB_2633280 |
| Goat anti-Rabbit IgG (H + L) Cross-Adsorbed Secondary Antibody, Alexa Fluor 568 | Invitrogen | Cat#A-11011; RRID: AB_143157 |
| Goat anti-Rabbit IgG (H + L) Cross-Adsorbed Secondary Antibody, Alexa Fluor 647 | Invitrogen | Cat#A-21244; RRID: AB_2535812 |
| Goat anti-Guinea Pig IgG (H + L) Highly Cross-Adsorbed Secondary Antibody, Alexa Fluor 647 | Invitrogen | Cat# A-21450; RRID: AB_2535867 |
| Alexa Fluor 488 AffiniPure Donkey Anti-Chicken IgY (IgG) (H + L) | Jackson ImmunoResearch Labs | Cat# 703-545-155; RRID: AB_2340375 |
| Anti-mouse IgG for IP (HRP) | Abcam | Cat#ab131368; RRID: AB_2895114 |
| HRP-labeled Goat Anti-Mouse IgG(H + L) | Beyotime | Cat#A0216; RRID: AB_2860575 |
| VeriBlot for IP Detection Reagent (HRP) | Abcam | Cat#ab131366; RRID: AB_2892718 |
| HRP-labeled Goat Anti-Rabbit IgG(H + L) | Beyotime | Cat#A0208; RRID: AB_2892644 |
| **Oligonucleotides and other sequence-based reagents** | | |
| All shRNA sequences | This study | Table EV1 |
| All qPCR primers | This study | Table EV2 |
| *Ryk* smFISH probes | This study | Table EV2 |
| **Chemicals, enzymes and other reagents** | | |
| CY5-UTP | APExBIO | B8333 |
| Cholera toxin subunit B (recombinant), Alexa Fluor 647 conjugate | Invitrogen | C34778 |
| LysoTracker Red DND-99 | Yeasen | 40739ES50 |
| MitoTracker Deep Red FM | Cell Signaling Technology | 8778S |
| BoNT/A-Hc | This study | N/A |
| 1,6-Hexanediol | Sigma-Aldrich | 240117 |
| NaveniFlex Cell Atto647N | Navinci | 60017 |
| Poly (ethylene glycol) 8000 | Solarbio | P8260 |
| Anti-HA Nanobody Magarose Beads | Alpalife | KTSM1335 |
| Anti-Flag M2 Affinity beads | Sigma-Aldrich | A2220 |
| Anti-Flag magnetic beads | Selleck | B26102 |
| Ni Sepharose 6 Fast Flow histidine-tagged protein purification resin | Cytiva | 17531802 |

| Reagent/resource | Reference or source | Identifier or catalog number |
|---|---|---|
| Glutathione Sepharose 4 Fast Flow GST-tagged protein purification resin | Cytiva | 17513202 |
| HiLoad Superdex 75 pg preparative size-exclusion chromatography columns | Cytiva | 28989333 |
| SpinDesalt Column | Smart-Lifesciences | SEC02301 |
| SYTO RNASelect Green Fluorescent Cell Stain | Invitrogen | S32703 |
| Atto 647 N NHS ester | Sigma-Aldrich | 05316-1MG-F |
| Vari Fluor 568 SE | MedChemExpress | HY-D1799 |
| iFluor 488 succinimidyl ester | AAT Bioquest | 1023 |
| Fluoroshield™ | Sigma-Aldrich | F6182 |
| DAPI | Sigma-Aldrich | D9542 |
| Propidium Iodide | Invitrogen | P1304MP |
| Nocodazole | Sigma-Aldrich | M1404 |
| TRIzol reagent | Ambition | 15596026 |
| DEPC-water (DNase, RNase free) | Beyotime | R0021 |
| HiScript III 1st Strand cDNA Synthesis Kit (+gDNA wiper) | Vazyme | R312 |
| ChamQ Universal SYBR qPCR Master Mix | Vazyme | Q711 |
| YSFluor 594-conjugated Streptavidin | Yeasen | 35107ES60 |
| **Software** | | |
| FIJI | https://imagej.net/software/fiji/ | |
| Graphpad | https://www.graphpad.com/ | |
| Imaris | https://imaris.oxinst.com/ | |
| Huygens | https://svi.nl/Download | |
| "Wu Kong" platform | https://www.omicsolution.com/wkomics/wkold/ | |
| Metascape | https://metascape.org | |
| Bioinformatics | https://www.bioinformatics.com.cn | |
| BioGRID | https://thebiogrid.org | |
| PrDOS | https://prdos.hgc.jp/cgi-bin/top.cgi | |
| Nikon Elements AR | https://www.microscope.healthcare.nikon.com | |
| Tsingke Biotechnology cloud platform | https://cloud.tsingke.com.cn | |

## Primary neuronal culture and transfection

Hippocampal or cortical tissues were derived from embryonic day 18 (E18) Sprague–Dawley rat brains, following relevant guidelines and regulations as approved by the Animal Ethics Committees of ShanghaiTech University (approval number: 20230217002). Then neurons were dissociated, suspended in plating medium (DMEM with 10% FBS, 10% F-12 and 1% Penicillin-Streptomycin), and seeded on Poly-L-Lysine coated glass coverslip, 29 mm glass-bottom dish (#D29-20-1.5-N, Cellvis) at $3.2 \times 10^4$ cells/cm$^2$, or into polydimethylsiloxane (PDMS) microfluidic device at $2 \times 10^5$ cells per reservoir, as previously described (Pan et al, 2024; Pan et al, 2022; Wang and Meunier, 2022). Plating medium was half changed to maintain medium (Neurobasal Medium with 2% B27 and 1% L-GlutaMax) on DIV1, and on DIV3-4, 10 μM 5-fluoro-2'-deoxyuridine (FDU) was added to suppress non-neuronal cell growth. All plasmids used for transfection were maintained in and purified from *E. coli* TOP10 competent cells. For hippocampal neurons, DIV6 neurons were transfected with 1–2 μg indicated plasmids using Lipofectamine 2000. Cortical neurons were electroporated using Nucleofector 2b (Lonza) before seeding and harvested on DIV8-11 for western blot analysis.

## Analysis of granule trafficking in live axons

To fluorescently label RNPs in axons, DIV6 rat hippocampal neurons cultured in 29 mm glass-bottom dishes were co-transfected with 0.2 nmol of CY5-UTP (#B8333, APExBIO) for total RNA labeling and 1 μg of EGFP-TIA1. Live-imaging was then conducted on DIV8 by replacing the medium with imaging buffer (15 mM HEPES, 145 mM NaCl, 5.6 mM KCl, 2.2 mM CaCl$_2$, 0.5 mM MgCl$_2$, 5.6 mM D-glucose, 0.5 mM ascorbic acid, 0.1% BSA, pH 7.4). Time-lapse confocal images were acquired using a Nikon TI2-E inverted microscope equipped with a Yokogawa spinning confocal disc head (CSU-W1 2 camera) with a 60 × 1.4 NA oil objective, with 1–5 s interval. Acquired time stacks were analysed in ImageJ (v2.3.0/1.53 f, NIH). Kymographs of RNP movement were generated using the multi-Kymograph plugin. Directions were assigned based on relative location to the soma.

For co-labeling of TIA1 granules with axon-derived lysosomes, signaling endosomes, and synaptic vesicle–related vesicles, DIV6 rat hippocampal neurons cultured in microfluidic devices were transfected with EGFP-TIA1 or TIA1-mCherry and co-labeled on DIV8–9 using a pulse-chase labeling assay as previously described (Pan et al, 2024; Wang and Meunier, 2022). Briefly, the axon terminal chamber was incubated with imaging buffer containing 50 ng/ml Alexa Fluor 647–conjugated recombinant CTB (#C34778, Invitrogen), 100 nM BoNT/A-Hc-Atto647N, or 50 nM LysoTracker Red (#40739ES50, Yeasen) for 10–30 min at 37 °C.

For other organelle marker labeling, neurons cultured in petri dishes were either co-transfected with fluorescently tagged TIA1 and Rab5 or DIC1B on DIV6, or live-stained with 500 nM MitoTracker Deep Red FM (#8778S, Cell Signaling Technology) on DIV8–9. Axons of live neurons were imaged under the same conditions by spinning disc microscopy with 4–20 s interval. The total number of TIA1 granules and their co-trafficking proportions with dynein or membranous markers were manually counted in kymographs using ImageJ.

For axon-derived RNP labeling, similar to other axon-derived marker assays, neurons were cultured in microfluidic devices until DIV8. RNA-select dye (1 μM; SYTO RNASelect Green Fluorescent Cell Stain, Cat#S32703, Invitrogen) was applied to the axonal compartment for 30 min, following procedures described in (Wang and Meunier, 2022). Briefly, to ensure spatial restriction of the dye,

the volumes of the soma and axon chambers were maintained at a 10:7 ratio. After labeling, the dye was either replaced with live-imaging buffer for immediate imaging or with conditioned medium followed by incubation in a $CO_2$ incubator for 60 min to allow tracking of axon-derived RNPs back to the soma prior to fixation and immunofluorescence staining.

For TIA1 granule axon trafficking analysis, rat hippocampal neurons at DIV6 were cultured in a microfluidic device or a glass-bottom dish and transfected with EGFP-TIA1 or TIA1-mCherry or other indicated plasmids. DIV8 cells were imaged under the same conditions with a 6–19 s interval and analyzed using ImageJ, as previously described (Pan et al, 2024, Wang and Meunier, 2022). Briefly, axonal TIA1 granules were traced using the Trackmate plugin (v7.11) with an estimated object diameter of 0.8 μm. Granules used for statistical analysis were filtered based on track duration (over two frames) and track speed (0–2 μm/s). The mean speeds of the tracks were exported, and directions were assigned as described above.

## 1,6-Hex-induced granule diffusion analysis

To assess the response of TIA1 granules to 1,6-Hex, DIV6 rat hippocampal neurons were transfected with TIA1-mCherry. Live imaging was performed using a confocal microscope under the same conditions above on DIV12. Time-lapse images were captured before and after 1,6-Hex (#240117, Sigma-Aldrich) treatment with 11–16 s interval and analysed using ImageJ. Kymographs of axonal TIA1 granules were generated, and the TIA1 heterogeneity index was calculated as detailed in Appendix Fig. S1E.

## OptoDroplet assay

Hippocampal neurons were cultured in glass-bottom dishes and transfected with Opto-Control, Opto-TIA1, and indicated plasmids on DIV6. Live-cell imaging was performed on DIV9 using a previous confocal microscope with a 40× 1.3 NA oil objective. Neurons were exposed to combined laser excitation at 561 nm for mCherry imaging and 488 nm for blue light activation of Cry2. Time-lapse images were continuously acquired over 20 min span with 5–12 s interval, and ~120 s were sufficient for Opto-TIA1 granule formation. The acquired time stacks were analysed in ImageJ. Heterogeneity indexes were calculated as described in Appendix Fig. S1E. The first frame was designated as the "Before" state, while the subsequent frames as the "After" state. Axonal granule number was manually assessed and normalized to axon length. Following 120 s of blue light activation, time-lapse images were analysed using the TrackMate plugin to trace and analyse the movement of axonal Opto-TIA1 granules, following the protocol described above.

## Axonal Ca²⁺ -elevation assays in live neurons

### For high K⁺-induced persistent Ca²⁺ elevation
On DIV12-13, rat hippocampal neurons expressing the Ca²⁺ sensor GCaMP6f were subjected to high K⁺ stimulation following previously established protocols (Wang et al, 2015). Briefly, the culture medium was replaced with a warm high K⁺ buffer (same as the imaging buffer except that it contained 95 mM NaCl and 56 mM KCl), whereas control neurons were treated with an imaging buffer. Imaging was performed as above using a confocal microscope with a 60× 1.4 NA oil objective, and time-lapse images were captured continuously immediately after the medium was replaced, with a 12–20 s interval. The resulting time stacks were analysed using ImageJ, adhering to previously outlined analysis steps.

### For flux-induced transient Ca²⁺ elevation
The flux induced mild mechanical stress was applied selectively to axonal regions by injecting conditioned medium via an AoC device as previously described (Pan et al, 2024; Pan et al, 2022). Specifically, conditioned culture medium was injected into the central injury channel at 50 μL/min for 3 min using a program-mable syringe pump (Pump 11 Elite; #704505, Harvard Apparatus). High-temporal-resolution imaging was performed on a Nikon Ti2-E inverted microscope equipped with a Yokogawa CSU-W1 spinning-disk confocal unit and a 60× objective (NA 1.4; WD 219.15 μm; 0.1826 μm/pixel resolution; 1200×1200 px). The conditioned medium consisted of phenol red–free Neurobasal medium (Thermo Fisher Scientific, #12348017) supplemented with the glutamatergic blockers DNQX (10 μM) and D-AP5 (40 μM) to suppress spontaneous activity. The acquired live-imaging stacks were processed and quantified using ImageJ as described previously.

## Lattice SIM and analysis

To examine the co-localization of endogenous TIA1 with ANXA7 or DIC1B in axons, rat hippocampal neurons transfected with the indicated plasmids on DIV6 were fixed and stained on DIV12. Imaging was conducted using Lattice SIM on a ZEISS Elyra 7 microscope with a 63× 1.4 NA oil objective, utilizing a grid size of 27.5 μm with 12 rotations. Raw SIM images were processed with Fourier transformation in Zen software (version 16.0.13.306, ZEN 3.0 SR black edition; Zeiss), followed by the application of a sharpness filter and fast fit advanced filter. The processed 3D-SIM images were analysed in ImageJ. Co-localization rates between the two channels were calculated as Pearson's correlation coefficients using the JACoP plugin.

## Protein expression and purification

His-TEV-Flag-DIC1B and His-TEV-Myc-ANXA7 proteins were expressed in *E. coli* BL21(DE3) cells. Cultures were induced at an $OD_{600}$ of 0.6–0.7 with IPTG (1 mM and 0.5 mM, respectively) at 16 °C for 16 h. The cells were resuspended in lysis buffer (50 mM phosphate buffer, 300 mM NaCl, 50 mM L-arginine, 2 mM $MgCl_2$, 2 mM imidazole, pH 7.0) with 0.2 mM PMSF, 2 mM DTT, 1 mM protease inhibitor, and 20 U/mL DNase I. After sonication, soluble proteins were separated by centrifugation at 18,000×*g* for 30 min. The soluble fraction was incubated with Ni Sepharose 6 Fast Flow resin (#17531802, Cytiva), washed with wash buffer (50 mM phosphate buffer, 300 mM NaCl, 10% glycerin, 2 mM DTT, 50 mM imidazole, pH 7.0), and eluted with elution buffer (50 mM phosphate buffer, 300 mM NaCl, 0.5 M L-arginine, 200 mM imidazole, pH 7.0, 2 mM DTT). The eluted fractions were dialyzed against TEV cleavage buffer (50 mM phosphate buffer, 300 mM NaCl, 0.5 M L-arginine) to reduce imidazole concentration

to below 0.2 mM. Protein concentration was estimated by SDS-PAGE using BSA standards. TEV protease was added to the protein solution at a 1:30 enzyme-to-protein mass ratio and incubated at 4 °C for 24 h. Following cleavage, the mixture was treated with Ni Sepharose 6 Fast Flow resin to remove the tag, and the target protein was collected from the flow-through. The purified protein was concentrated and stored in aliquots at −80 °C.

The expression and purification of GST-TEV-TIA1 followed a similar protocol, substituting Glutathione Sepharose 4 Fast Flow resin (#17513202, Cytiva) for affinity purification. Imidazole was excluded from all buffers, and the elution buffer contained 10 mM reduced glutathione. TEV protease was used to cleave the GST tag from TIA1, and the resulting proteins were purified using size-exclusion chromatography on a HiLoad 16/600 Superdex 75 pg column (#28989333, Cytiva) with an ÄKTA Pure system (Cytiva).

## Co-IP, in vitro pull-down and western blot

For the Co-IP assay, HEK293T cells were harvested 48 h post-transfection, while primary cortical neurons were collected 11 days post-electroporation. Cells were washed with cold PBS and lysed in NP40 lysis buffer (50 mM Tris-HCl, pH 8.0, 150 mM NaCl, 5 mM MgCl$_2$, and 0.5% NP40) containing protease inhibitors. Lysates were then centrifuged at 21,400×$g$ for 10 min at 4 °C, and the supernatants were collected for IP. IP was performed using anti-Flag M2 affinity beads (#A2220, Sigma-Aldrich) or anti-HA magnetic beads (#B26202, Bimake) with a 3-h incubation at 4 °C. Beads were subsequently washed with NP40 wash buffer (50 mM Tris-HCl, pH 8.0, 300 mM NaCl, 5 mM MgCl$_2$, and 0.1% NP40), and the bound proteins were eluted for western blot detection.

For the in vitro pull-down assay, 2 μg Flag-DIC1B was incubated with anti-Flag magnetic beads (#B26102, Selleck) in NP40 lysis buffer with 0.5 M L-arginine at 4 °C for 1 h. Following this, 2 μg each of Myc-ANXA7 and TIA1 were added, and the incubation was continued at 4 °C for an additional hour. After incubation, the supernatants were removed, and the beads were washed five times with NP40 wash buffer with 0.5 M L-arginine. The beads were then mixed with 1× SDS loading buffer. For experiments involving Ca$^{2+}$, 1 mM CaCl$_2$ was added to both the NP40 lysis and wash buffers. Interactions were quantitatively analysed by Western blot.

Western blot samples were separated on 8% or 10% Tris-glycine polyacrylamide SDS-PAGE and transferred to PVDF membranes. Membranes were then blocked with 5% non-fat milk in TBST (0.05% Tween) for 1 h at room temperature, followed by overnight incubation with primary antibodies at 4 °C. The membranes were washed and then incubated with secondary antibodies for 1 h at room temperature. Primary antibodies were diluted at 1:2000, while secondary antibodies were diluted at 1:5000, except for Veriblot, which was diluted at 1:1000. Blots were detected immediately using the Amersham Imager 680 (Cytiva) or the Touch Imager (e-BLOT Life Science).

## Proximity ligation assay

PLA was conducted using the NaveniFlex Cell MR Atto647N kit (Navinci) according to the manufacturer's instructions. DIV6 rat hippocampal neurons were transfected with the indicated siRNA sequences. On DIV12, neurons were fixed with 4% paraformaldehyde and 4% sucrose in PBS for 30 min. Following fixation, cells

were permeabilized and blocked in a solution containing 0.1% saponin, 1% BSA, and 0.2% gelatin for 1 h at room temperature. Primary antibodies were diluted 1:500 in the antibody dilution buffer provided in the kit and incubated overnight at 4 °C. The following day, Navenibody M1 and R2 were added at a 1:40 dilution and incubated at 37 °C for 1 h. Ligation and rolling circle amplification reactions were subsequently performed using Reaction 1 and Reaction 2 reagents, respectively, with the latter containing the Atto647N fluorescent dye. After PLA staining, neurons were immunolabelled with chicken anti-βIII-tubulin antibody to visualize neuronal morphology. Samples were mounted on glass slides and subjected to z-stack confocal imaging.

For quantitative analysis, a consistent intensity threshold was applied across all samples within each experimental set to exclude background fluorescence, and this threshold was maintained throughout the analysis. Images were binarized, and puncta within the size range of 0–10 μm$^2$ were automatically quantified using the "Analyze Particles" function in ImageJ.

## Proteomic analysis of brain interactomes

BioGRID analysis was conducted by downloading interactor data from the BioGRID database (https://thebiogrid.org). Overlapping interactors of TIA1 and DIC1B were identified using the "Conditional Formatting > Highlight Cells Rules > Duplicate Values" feature in Excel.

Cortical tissues from P14 rats were homogenized on ice in homogenization buffer (0.32 M sucrose, 10 mM HEPES, pH 7.4) and lysed in four volumes of RIPA buffer (50 mM Tris-HCl, 150 mM NaCl, 1% NP40, 0.25% sodium deoxycholate, pH 7.4) with protease inhibitors. The lysates were centrifuged at 15,000×$g$ for 40 min at 4 °C to remove debris. Supernatants were quantified using the Bradford assay and subsequently pre-cleared with glutathione resin. The purified proteins were blocked with 1% BSA for 1 h at 4 °C, then added to the brain lysates and incubated at 4 °C for an additional 2 h. After this incubation, glutathione resin beads were introduced and incubated under the same conditions for another 2 h. The beads were then pelleted by centrifugation at 1000×$g$ for 5 min and washed four times with NP40 wash buffer. Western blot analysis was subsequently performed as previously described.

For proteomic analysis following the GST pull-down, the prepared samples were processed for mass spectrometry. Gel strips were cut into 1.5 mm pieces and washed, then decolorized using a 25 mM NH$_4$HCO$_3$/acetonitrile (1:1) solution, dehydrated with acetonitrile, and vacuum-dried. The proteins were reduced with 10 mM DTT at 56 °C for 1 h and then alkylated with 25 mM iodoacetamide (IAM) for 45 min in the dark. Following sequential washes with 25 mM NH$_4$HCO$_3$, a 25 mM NH$_4$HCO$_3$/acetonitrile (1:1) solution, and acetonitrile, the samples were vacuum-dried again. Proteins were digested by adding an enzyme at a 50:1 protein-to-enzyme ratio. The samples were incubated at 4 °C for 20 min and then at 37 °C overnight. The resulting peptides were extracted with 50% acetonitrile/0.5% formic acid, combined, and vacuum-dried. The peptides were redissolved in 0.1% formic acid and desalted using Stage-Tips.

Proteomic analysis was conducted using a Q Exactive HF-X mass spectrometer (Thermo Fisher Scientific). The mass spectrometry (MS) data were analysed using Proteome Discoverer (version PD2.2) and normalized by total protein intensity. Seq-k-nearest

neighbor (Seq-Knn) imputation was applied for missing values using the "Wu Kong" platform (https://www.omicsolution.com/wkomics/wkold/). Differential analysis between GST control and GST-TIA1/GST-ANXA7 was based on a fold change (FC) ≥2.3 ($\log_2$FC ≥1.2) and a P value ≤0.05 ($-\log_{10}P \geq 1.3$). GO and KEGG pathway analyses were performed using Metascape (Zhou et al, 2019), and enrichment dot bubble plots were generated on https://www.bioinformatics.com.cn.

## FRAP assay

Forty-eight hours after being transfected with EGFP-TIA1 plasmids on DIV6, rat hippocampal neurons in glass-bottom dishes were placed on the previously described confocal microscope with a 60× 1.4 NA oil objective for live-imaging, with 1 s interval. EGFP signals were bleached using a 488-nm laser set at 90% intensity for 100 ms, following the acquisition of six prebleach images. Then, the neurons were allowed to recover and recorded for 5 min after photobleaching.

Time-lapse images were processed and analyzed using Nikon Elements AR, and time measurement results were exported. FRAP efficiency (E) was calculated using the Eq. (1):

$$E = \frac{F_t - F_0}{F_c - F_0} \times 100\% \tag{1}$$

where $F_t$ is the intensity at time t, $F_0$ is the intensity immediately after photobleaching, and $F_c$ is the intensity before photobleaching (corrected by Exponential One phase decay in GraphPad).

## FLIM-FRET assay

Primary hippocampal neurons were transfected with the indicated plasmids (Donor: EGFP-TIA1, Acceptor: DIC1B-mRFP) on DIV6. On DIV9, neurons were imaged in prewarmed imaging buffer using a Leica STELLARIS 8 FALCON microscope with a 63× 1.4 NA oil objective. The tunable white light laser was set to 489 nm excitation at an 80 MHz frequency. Emission from 494 to 540 nm was collected using a HyD X1 detector, and laser power was adjusted to achieve approximately 1 photon per laser pulse, following the published method (Cuevas-Velazquez et al, 2021). Confocal settings included a $512 \times 512$ pixel resolution with a 4.0 optical zoom, resulting in a 0.09-μm pixel size. FLIM images were processed using LAS X 4.4.0.24861 software (Leica Microsystems). Lifetime decay curves were fitted with an n-Exponential Reconvolution model, selecting the number of components based on $\chi^2$ values closest to 1. FLIM images were analysed using ImageJ.

FRET efficiency ($E_{FRET}$) was calculated in Eq. (2):

$$E_{FRET} = \frac{tD - tDA}{tD} \times 100\% \tag{2}$$

where $t_{DA}$ is the donor lifetime (EGFP-TIA1) in the presence of the acceptor (DIC1B-mRFP), and $t_D$ is the donor lifetime without the acceptor.

## In vitro phase separation assays

For $Ca^{2+}$- or $Mg^{2+}$-induced sedimentation assays, purified Myc-ANXA7 and TIA1 proteins were desalted, quantified, and mixed at 5 μM in LLPS

buffer (50 mM Tris-HCl, pH 8.2; 100 mM NaCl; 1 mM DTT) supplemented with 0–10 mM $CaCl_2$ or $MgCl_2$. For $Na^+$-induced assays, reactions were prepared in modified LLPS buffer (50 mM Tris-HCl, pH 8.2; 1 mM DTT) supplemented with 0–500 mM NaCl. Samples were incubated at 37 °C for 20 min, then centrifuged at 17,000×g for 15 min at 25 °C. After centrifugation, 50 μL of the supernatant (S) was collected, and an equal volume of LLPS buffer containing 8 M urea was added to the pellet (P). Aliquots (10 μL) from both fractions were analyzed by SDS-PAGE followed by Coomassie staining. Phase separation was quantified by measuring band intensity.

For fluorescently labeled phase separation assays, fluorescent dyes (Atto 647 N NHS ester (#05316-1MG-F, Sigma-Aldrich), Vari Fluor 568 SE (#HY-D1799, MedChemExpress), or iFluor 488 succinimidyl ester (#1023, AAT Bioquest)) were conjugated to purified proteins (2 mg/ml) in bicarbonate buffer (0.1 M, pH 8.3) at a dye-to-protein ratio of 10:1. After a 1-h incubation at room temperature, labeled proteins were desalted using SpinDesalt columns, and their concentrations were measured. Proteins were then mixed in phase separation buffer (10 mM HEPES, 150 mM NaCl, 0.1 mM EDTA, 2 mM DTT, pH 7.4), with PEG-8000 (#P8260, Solarbio) or $Ca^{2+}$ in some cases. Samples were injected into homemade imaging chambers, consisting of a coverslip and glass slide held together by double-sided tape. After a 15-min incubation at room temperature, the chamber was imaged using the Leica Thunder Imager (DMi8) with HC PL APO 63 × 1.40 NA oil objective (Fig. 3D), Nikon TI2-E inverted microscope equipped with a Yokogawa spinning confocal disc head (CSU-W1) and a 60 × 1.4 NA oil objective (Fig. EV3A,E), or Olympus SpinSR with a UPLXAPO 20 × objective (Figs. 3C,E and EV3F).

For the microscopy-based phase separation assay, to generate the DIC-647 coated beads, 20 μl of Anti-Flag M2 Affinity beads were incubated with Atto 647-labbled Flag-DIC1B (DIC-647, 5 μM) at 4 °C for 1 h and washed by 200 μL imaging pull-down buffer (25 mM Tris-HCl pH 7.5, 150 mM NaCl, and 1 mM DTT). The DIC-647-coated beads were centrifuged at 1500×g for 2 min at 4 °C and resuspended in 10 μL of imaging buffer. Then, 2 μL of resuspended beads were added to 20 μL of imaging mixture containing 5 μM ANXA7-568 and 5 μM TIA1-488. After a 30-min dark incubation, the 20 μL imaging mixture was placed in the center of a slide with double-sided tape, and a coverslip was placed on top. The slide was inverted and immediately imaged using an Olympus SpinSR with a UPLXAPO 20× objective to obtain 3D stacks of the coated beads. All experiments were conducted in darkness to minimize bleaching.

## IF staining and 3D rendering analysis

Primary rat hippocampal neurons at DIV9 or DIV12 were fixed in 4% PFA and 4% sucrose in PBS for 30 min at room temperature. After blocking in antibody diluting buffer (0.1% saponin, 1% BSA, and 0.2% gelatin in PBS) for 1 h at room temperature, cells were incubated with primary antibodies (1:500) overnight at 4 °C, followed by secondary antibodies (1:5000) for 1 h in the dark at room temperature. DAPI in PBS was added for 10 min at room temperature, and cells were mounted with Fluoroshield mounting medium. For PI staining, Propidium Iodide (#P1304MP, Invitrogen) was added after DAPI incubation according to the product manual.

Z-stack images were captured using a Nikon TI2-E inverted microscope equipped with a Yokogawa spinning confocal disk head

(CSU-W1 2 camera) with a 60 × 1.4 NA oil objective. Images were analyzed in ImageJ. Co-localization rates between two channels were measured using Pearson's coefficient with the JACoP plugin. For granule detection, axon shafts with lengths ≥50 μm were selected and straightened. Granules were identified using the "Analyze Particles" function with a circularity >0.6 and area between 0.05 and 20 μm². Detected granule sizes were exported, and granule numbers were normalized to axon length.

To analyze the axonal co-localization of TIA1 with G3BP1 or p62, the 3D stacks of confocal images were deconvoluted using 40 cycles of Huygens Professional software (v18.10, Scientific Volume Imaging) and imported into Imaris software (v9.7.2, Bitplane) for morphology fitting using the "surface" function. For precise 3D renderings, "Surface Grain Size" was set at 0.100 μm and "Diameter of Largest Sphere" at 1.00 μm for neurons co-labeled with TIA1 and G3BP1, while for neurons co-labeled with TIA1 and p62, parameters were 0.00100 and 1.00 μm, respectively.

## RNA-sequencing (RNA-seq) and RT-qPCR

Primary rat cortical neuron suspensions were electroporated with shANXA7 or vector plasmids using a Nucleofector 2b device (Lonza) and then seeded on transwell inserts with 3 or 8 μm pore size membranes (#TCS019006 or #TCS020006, Jet Biofil). On DIV12, axons adhering to the bottom surface of the inserts were collected and lysed in TRIzol reagent (#15596026, Ambion). Lysates from three biological replicates were sent to Tsingke Biotechnology (China) for total RNA extraction, library preparation and sequencing. Sequencing data were generated using the DNBSEQ platform. Raw reads were quality-filtered using *fastp* (v0.20.0), and aligned to the mRatBN7.2 reference genome with *HISAT2* (v2.2.1). Differential expression analysis between shANXA7 and control groups was performed using *DESeq2* (v1.26.0), applying thresholds of fold change ≥1.2 or ≤−1.2 ($\log_2$ fold change ≥0.26 or ≤−0.26) and adjusted $p$ value ($p$adj) ≤0.05 ($-\log_{10} p$adj ≥1.3).

Total RNA from the TRIzol-lysed axon samples was further purified using the chloroform extraction method. For cDNA synthesis, 1 μg of total RNA was reverse transcribed using the HiScript III 1st Strand cDNA Synthesis Kit (+gDNA wiper) (#R312, Vazyme). Quantitative real-time PCR was performed on a QuantStudio 7 system (Thermo Fisher Scientific) using ChamQ Universal SYBR qPCR Master Mix (#Q711, Vazyme) with gene-specific primers for *Ryk*, *Neurod2*, *Car7*, and *Gapdh*. Primer sequences are provided in Table EV2.

## smFISH

Custom Stellaris RNA FISH probes were designed against *Ryk* (NM_080402.3) using the Stellaris RNA FISH Probe Designer (LGC, Biosearch Technologies, Petaluma, CA) available online at www.biosearchtech.com/stellarisdesigner (version 4.2). A set of 48 probes labeled with biotin was synthesized by GENEWIZ (China), and the sequences are provided in Table EV2.

On DIV6, rat hippocampal or cortical neurons were transfected with EGFP-N1, shANXA7-1#, or shANXA7-1#/ANXA7-res constructs. On DIV12, neurons were fixed with 4% PFA and 4% sucrose in PBS for 30 min, then hybridized with the *Ryk* Stellaris RNA FISH probe set following the protocol from www.biosearchtech.com/stellarisprotocols. Briefly, neurons were

permeabilized with 70% ethanol for 1–2 h at 4 °C, then washed twice with wash buffer (10% formamide in 2× SSC) for 5 min each. Hybridization was performed overnight at 37 °C in hybridization buffer (100 mg/mL dextran sulfate and 10% formamide in 2× SSC) containing 125–250 nM probes targeting *Ryk*. The following day, cells were washed twice in wash buffer for 10 min at 29 °C, then incubated in hybridization buffer containing 1.8 μg/mL YSFluor 594-conjugated streptavidin (#35107ES60, Yeasen) for 1 h at 37 °C in the dark. After two additional washes in wash buffer for 10 min at 29 °C, DAPI diluted in 2× SSC was added, followed by incubation for 10 min. Cells were mounted with Fluoroshield mounting medium and imaged using a confocal microscope.

For image analysis, z-stack images were processed in ImageJ. Axon shafts ≥50 μm were straightened, and *Ryk* mRNA smFISH granules were identified using the "Analyze Particles" function with an area threshold ≥0.05 μm². Granule counts were normalized to axon length to calculate granule density.

## AAV packaging and ICV injection

The pAAV-U6-hSyn-EGFP plasmid was constructed by cloning the U6 promoter into the pAAV-hSyn-EGFP vector using specific primers (forward: 5'-CGGCCGCACGCGTGTGTGAGGGCC-TATTTCCCATGAT-3'; reverse: 5'-CAGGGCCCTCTGCAGTC-TAGAGGTGTTTCGTCCTTTCCAC-3'). shRNA targeting mouse ANXA7 (shANXA7-3# and shANXA7-4#; Table EV1) was synthesized and inserted into the pAAV-U6-hSyn-EGFP plasmid. In addition, a rescue construct was generated by fusing shANXA7-4# with a P2A sequence and the cDNA encoding the rat ANXA7 homolog, which was then inserted into the pAAV-U6-hSyn-EGFP plasmid. AAV particles were packaged into serotype 9 capsids and purified as previously described (Pan et al, 2024). Briefly, the target plasmids purified from *E. coli* Stbl3 cells, together with pHelper and pPHP.s plasmids, were co-transfected into HEK293T cells using PEI. After 72 h, cells were harvested, lysed with buffer (150 mM NaCl, 20 mM Tris-HCl, and pH 8.0), treated with Benzonase for 45 min, and centrifuged to collect supernatant. AAV particles were purified by OptiPrep density gradient ultracentrifugation (40% fraction) and titrated by qPCR targeting the EGFP region.

All animal procedures were conducted under the ethical guidelines of the Institutional Animal Care and Use Committee of ShanghaiTech University (approval numbers: 20230217002 and 20250401001). C57BL/6J mice were housed on a 12-h light-dark cycle. Both male and female pups at postnatal day 1 were cryoanesthetized, and AAV were bilaterally injected into the cerebral ventricles using Drummond™ PCR Micropipets, pulled with a P-97 Flaming/Brown micropipette puller. Each ventricle received 1 μL of virus mixed with Fast Green for visualization.

## Rotarod test

Motor coordination of 8-week-old mice was assessed using a Rotarod machine (Ugo Basile, Model 47650). Mice were trained over 3 days (P56-P58) and tested on the 4th day (P59). Training involved constant speed at 4 rpm on P56, 4–20 rpm acceleration over 5 min on P57, and 4–40 rpm on P58. On P59, mice performed three trials at 4–40 rpm acceleration over 5 min, with a 20 min rest between trials. The average time to fall off from three trials was recorded for each mouse.

## Tissue sectioning, staining, imaging, and analysis

P60 mice were deeply anesthetised with isoflurane and perfused with 0.9% saline, followed by 4% PFA in PBS. The brain and spinal cord were dissected, post-fixed in 4% PFA at 4 °C overnight, and then dehydrated in 30% sucrose in PBS at 4 °C until sinking. Tissues were embedded in OCT (#4583, Sakura) and sectioned using a cryostat microtome, with brains coronally at 40 µm and spinal cords longitudinally at 25 µm. For IF staining, sections were permeabilized with 1% SDS in PBS for 4 min, blocked with 1% BSA in PBS for 1 h at room temperature, and incubated with primary antibodies (1:500) overnight at 4 °C. Then, they were incubated with secondary antibodies (1:5000) for 3 h at room temperature, and stained with DAPI for 10 min. Sections were mounted with Fluoroshield mounting medium. All steps were conducted in the dark. Images were captured using an Olympus VS120-S6-W slide scanner with a 20× 0.5 NA objective. Focused imaging was performed using a Nikon TI2-E inverted microscope equipped with a Yokogawa spinning confocal disk head (CSU-W1 2 camera) with a 60× 1.4 NA oil objective.

For axon morphological analysis, EGFP-labeled axons were quantified by measuring the width-to-height ratio of bounding rectangles for each signal using ImageJ. Axons with width-to-height ratios of 0.5–2 were classified as degenerating fragments. The total areas of infected and degenerating axons were used to calculate the percentage of degeneration. For the percentage of activated microglia in the motor cortex, Iba1$^+$ microglia and DAPI$^+$ cells were quantified using the "Analyse Particles" function, with an area threshold >5.27 µm² and normalized to the number of DAPI$^+$ cells.

## Statistical information

All data were illustrated and analyzed using GraphPad Prism (v8.3.0). Results are presented as mean ± SEM. Datasets that followed a normal distribution between two groups were analyzed using a two-tailed unpaired *t*-test to determine statistical significance. For paired data comparisons, a two-tailed paired *t*-test was used. One-way ANOVA was used for comparisons among multiple groups against a control group. When the control group was normalized to 1, a one-sample *t*-test was used to compare experimental groups to this normalized value. Statistical significance was defined as a *p* value of less than 0.05. See the Appendix file for all exact *P* values. For Appendix Fig. S1A',S1F', outliers were identified and excluded using the ROUT method with Q = 1%, and the results were confirmed to be consistent when including or excluding these points. Sample size adequacy was determined based on preliminary data or through discussion. ROI selection was randomized to avoid bias. Data collection and analysis were performed by independent operators who were blinded to the experimental conditions.

## Data availability

All RNA sequencing data of this study are deposited in the National Center for Biotechnology Information Sequence Read Archive (SRA) with the accession number BioProject: PRJNA1290857.

The source data of this paper are collected in the following database record: biostudies:S-SCDT-10_1038-S44318-025-00609-8.

## Peer review information

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

## Acknowledgements

We thank Professor F.A. Meunier, Professor Lei Li, on their constructive comments. We thank the Microscopy Core Facility, the Core Omic Facility, the Molecular and Cellular Core Facility, and the Animal Center of Westlake University for the facility support and technical assistance; Cartoons were created with BioRender.com. This work was supported by the Science and Technology Commission of Shanghai Municipality (24490713800 to T. Wang), National Natural Science Foundation of China (32271001 to T. Wang; 32100777 to Y. Chu). Liu acknowledges the support from the Double First-Class Initiative Fund of ShanghaiTech University (SYLPOC0022022 and SYLDX0302022). K. Dou acknowledges support from the National Natural Science Foundation of China (32370604).

## Author contributions

**Yu Feng**: Conceptualization; Data curation; Formal analysis; Validation; Investigation; Visualization; Methodology; Writing—original draft; Writing—review and editing. **Tongshu Luan**: Data curation; Formal analysis; Validation; Investigation; Visualization; Methodology; Writing—original draft; Writing—review and editing. **Zhenda Zhang**: Validation; Investigation; Visualization; Methodology. **Wei Wang**: Formal analysis; Validation; Investigation; Visualization; Methodology. **Yuanyuan Chu**: Funding acquisition; Investigation; Methodology. **Sijia Wan**: Formal analysis; Visualization. **Xiaorong Pan**: Methodology. **Jie Li**: Methodology. **Yifan Liu**: Resources; Supervision; Funding acquisition. **Yaqian Xu**: Methodology. **Kun Dou**: Resources; Supervision; Funding acquisition; Methodology. **Tong Wang**: Conceptualization; Resources; Data curation; Formal analysis; Supervision; Funding acquisition; Visualization; Writing—original draft; Project administration; Writing—review and editing.

Source data underlying figure panels in this paper may have individual authorship assigned. Where available, figure panel/source data authorship is listed in the following database record: biostudies:S-SCDT-10_1038-S44318-025-00609-8.

## Disclosure and competing interests statement

The authors declare no competing interests.

# Expanded View Figures

**Figure EV1.   TIA1 granules undergo retrograde trafficking in axons.**

(A) Cy5-UTP co-localization with indicated RNP markers. Arrowheads indicate RNPs with these markers. Scale ba = 5 µm. (B) Schematic diagram of the pulse-chase labeling assay to specifically label retrograde membranous axonal organelles in neurons cultured in a microfluidic device. See also the Methods for details. Scale bar = 50 µm. (C, D) Key frames from time-lapse images showing TIA1 granules trafficking with axon-derived CTB (C) or BoNT/A-Hc (D) in a microfluidic device. Arrowheads indicate moving TIA1 granules. Scale bar = 10 µm. (E, F) Key frames from time-lapse images showing TIA1 granules trafficking with whole cell stained MitoTracker (E) or co-expressed EGFP-Rab5 (F). Arrowheads indicate moving TIA1 granules and organelles. Scale bar = 5 µm. (G) Representative confocal images of endogenous TIA1 with organelle markers (Rab5 for endosomes, LC3 for autophagosomes, LAMP1 for lysosomes, DIC1B for dynein) in axons, with intensity profiles shown below. Scale bars = 10 µm (top), 5 µm (bottom). (G′-G‴) Quantification of (G), with (G′) showing the Pearson's coefficient of endogenous TIA1 granules with indicated markers, and (G″) showing the ratio of TIA1 co-localized with the indicated markers, and (G‴) showing the ratio of the markers co-localized with TIA1 ($n$ = 56, 56, 53, 56 axons from three biological replicates). (H) Screening of shDIC1B constructs for knockdown efficiency in cultured rat cortical neurons. shRNA sequences of 1# and 2# are available in Table EV1. (I–I′) Key frames from time-lapse images showing axon trafficking of TIA1-mCherry granules with or without 10 µM, 15 min nocodazole treatment, with arrowheads in different colors marking distinct granules (I), and corresponding quantification (I′) ($n$ = 334 and 497 granules from three biological replicates). Scale bar = 10 µm. Data represent mean ± SEM; two-tailed unpaired $t$-test in (I′); one-way ANOVA in (G′, G″, G‴); **$P < 0.01$, ***$P < 0.001$. See appendix for exact $P$ values. Source data are available online for this figure.

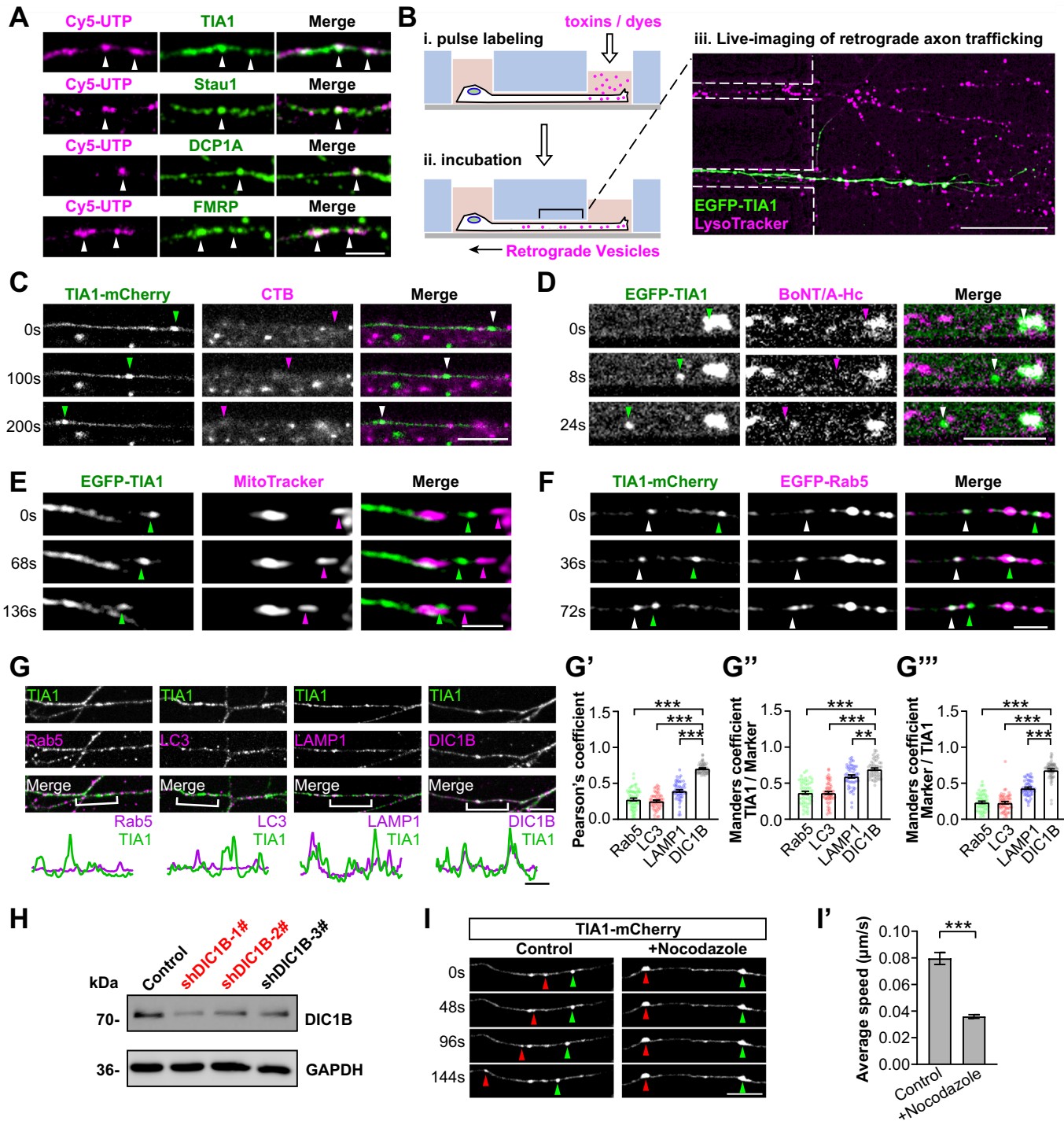

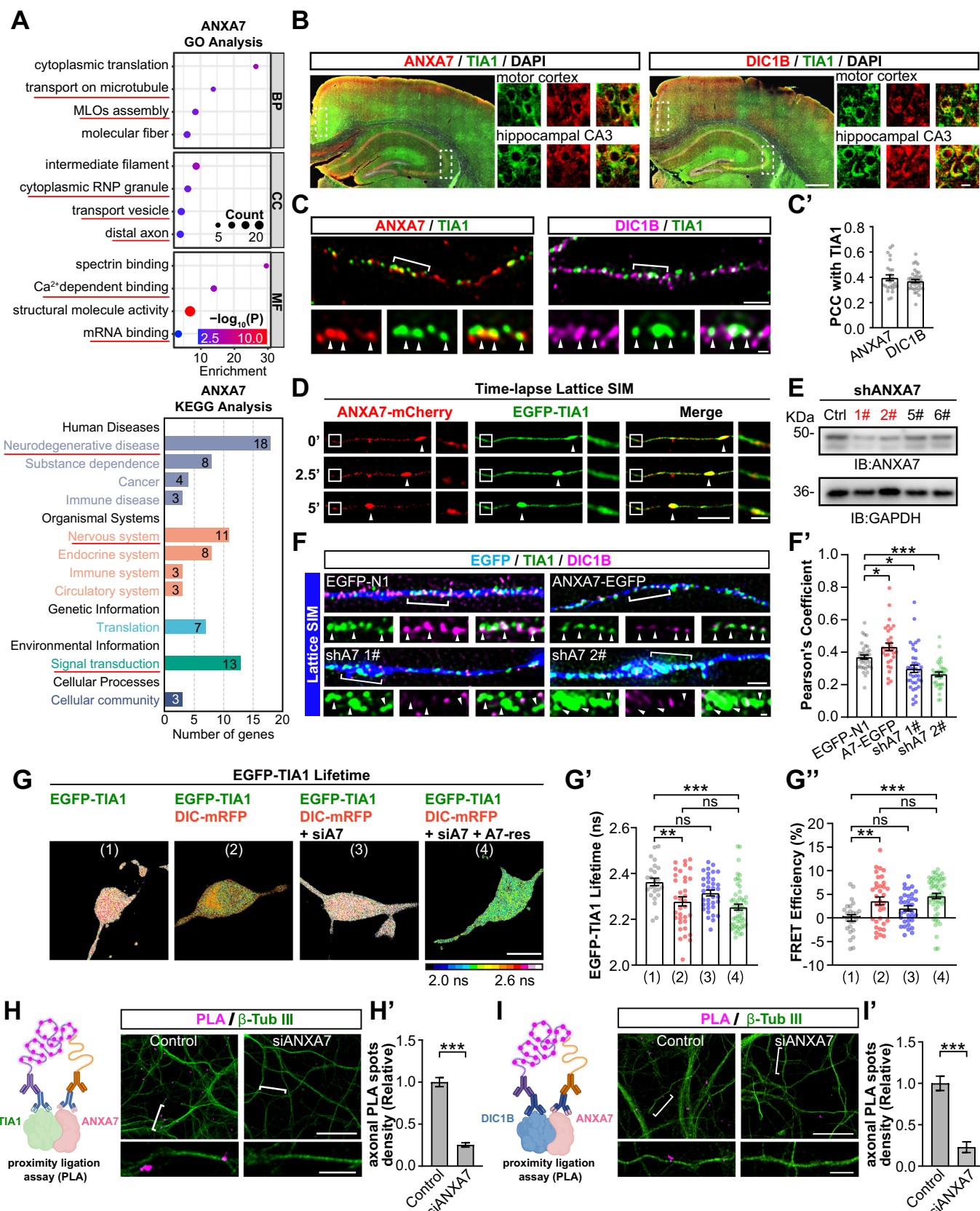

◄ **Figure EV2.   ANXA7 promotes the recruitment of TIA1 granules to dynein.**

(A) GO and KEGG pathway analysis of GST-ANXA7 interactors, including biological processes (BP), cellular components (CC), and molecular functions (MF). $P$ values were computed using the hypergeometric test, and $q$ values (adjusted $P$ values) were derived using the Benjamini–Hochberg FDR method. Only categories with $q < 0.05$ are shown, ranked by descending enrichment score ($-\log_{10}(P)$). (B) Confocal images of endogenous TIA1 (green) and ANXA7 or DIC1B (red) in the cortex and hippocampus of P34 mouse brain. Scale bars = 500 and 10 μm. (C) In DIV12 rat hippocampal neurons, 3D-Lattice SIM images of TIA1 (green) and ANXA7 (red) or DIC1B (magenta) along axons. Arrowheads indicate co-localization. Scale bars = 1 μm and 200 nm. (C') Pearson's coefficient for co-localization between TIA1 and ANXA7 or DIC1B from (C) ($n = 26$, 38 axons from three biological replicates). (D) Arrows in key frames from time-lapse SIM images showing co-trafficking of granules in DIV13 hippocampal neurons. One newly formed TIA1/ANXA7 granule (boxed) are amplified in the right panels. Scale bars = 5 and 1 μm. (E) Western blot showing shRNA-mediated knockdown of endogenous ANXA7 in rat cortical neurons. shRNA sequences (#1 and #2) are listed in Table EV1. (F) 3D-lattice-SIM images showing the distribution of endogenous TIA1 (green) and DIC1B (magenta) in axons of DIV12 cultured rat hippocampal neurons under conditions of endogenous ANXA7 knockdown (shA7 1# and 2#) or ANXA7-EGFP overexpression. Arrowheads indicate co-localized TIA1 and DIC1B spots. Scale bars = 1 μm (top) and 200 nm (bottom). (F') Pearson's coefficient quantifying co-localization between TIA1 and DIC1B ($n = 38$, 32, 37, 35 axons from three biological replicates). (G) Color-coded EGFP-TIA1 lifetime in the soma of transfected neurons, with lifetime (G') and FRET efficiency (G") quantified and compared across the indicated groups ($n = 25$, 35, 35, 53 neurons from 4 biological replicates). Scale bar = 10 μm. (H–I') Left: schematics illustrating PLA detection of endogenous TIA1/ANXA7 (H) and DIC1B/ANXA7 (I) interactions. Right: Confocal images showing PLA signals ANXA7 knockdown (siANXA7) neurons, with bracketed axons enlarged below. Scale bars = 50 μm (top), 10 μm (bottom). Quantification of axonal PLA density shown in (H') and (I') ($n = 100$, 72 ROIs for (H') from four biological replicates; $n = 60$ ROIs for (I') from three biological replicates). Data represent mean ± SEM; one-sample $t$-test in (H', I'), one-way ANOVA in (F', G', G"); $*P < 0.05$, $**P < 0.01$, $***P < 0.001$, ns non-significant. See appendix for exact $P$ values. Source data are available online for this figure.

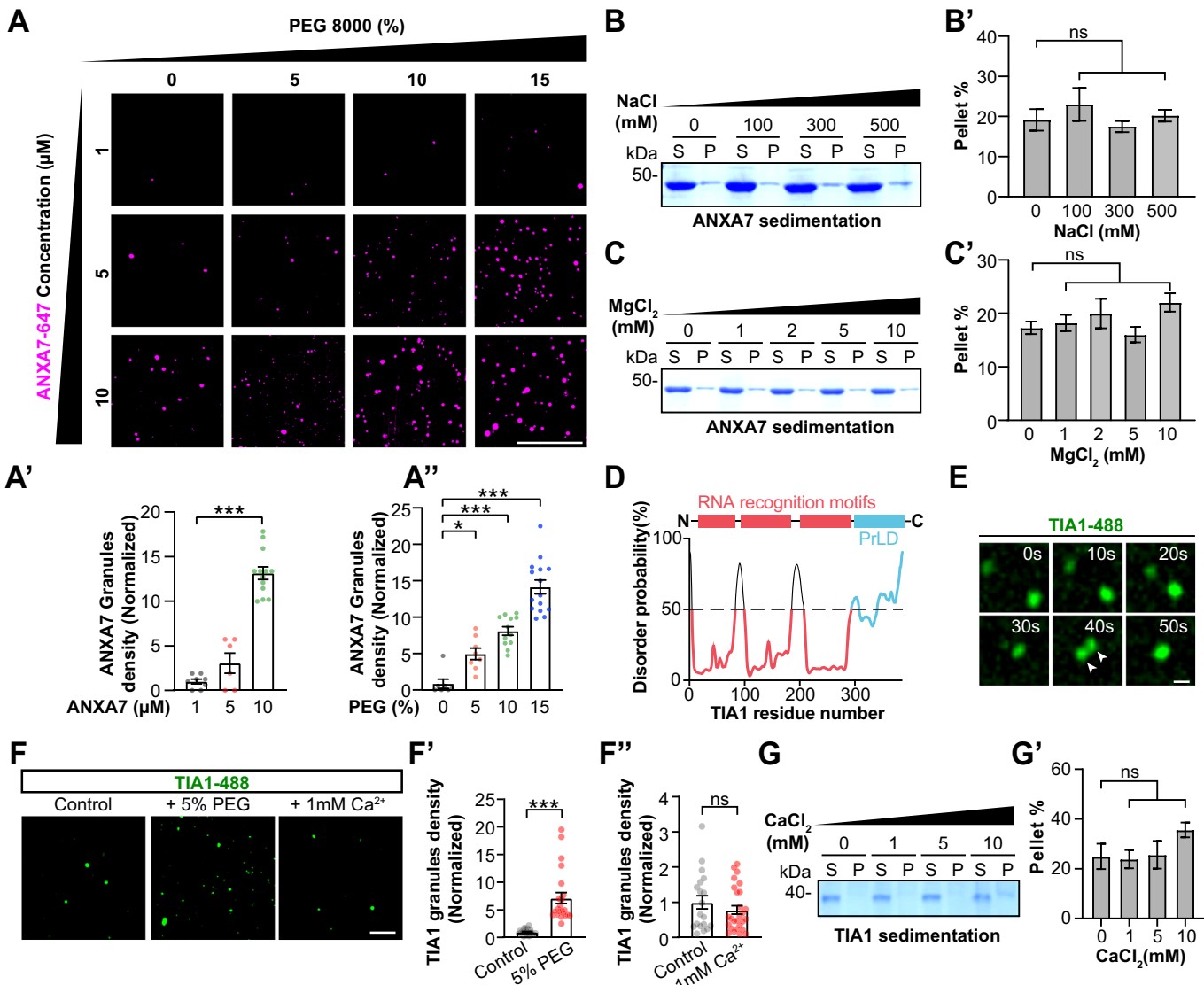

**Figure EV3.  The TIA1 phase separation is not affected by Ca²⁺ elevation.**

(A) In vitro assay demonstrating the phase separation of purified ANXA7-647 induced by PEG-8000 addition. The concentrations of PEG and ANXA7-647 are indicated. Scale bar = 50 µm. (A'–A") Quantification of (A), showing the ANXA7 granules density under different concentrations of protein (A') and PEG (A") (for (A'): PEG concentration = 0%, $n = 8, 6, 13$ ROIs; for (A"): ANXA7 concentration = 5 µM, $n = 7, 8, 12, 15$ ROIs from three biological replicates). (B–C') In vitro sedimentation assays detected by SDS-PAGE showing the distribution of purified ANXA7 (5 µM) between supernatant (S) and pellet (P) fractions at the indicated concentrations of NaCl (B) or MgCl₂ (C). Quantification shown in (B') and (C') (for (B'): $n = 5$ technical replicates from three biological replicates; for (C'): $n = 4$ biological replicates). (D) Schematic diagram of the TIA1 protein domain structure with PrDOS analysis, showing the C-terminal PrLD. (E) Key frames from time-lapse images showing the in vitro LLPS process of purified TIA1 protein (TIA1-488), with fusion events of droplets indicated by arrowheads. Scale bar = 2 µm. (F) In vitro LLPS assay using purified TIA1 protein (TIA1-488), illustrating the effects of adding 5% PEG or 1 mM Ca²⁺ on phase separation. Scale bar = 10 µm. (F'–F") Quantification of (F), with the effect of 5% PEG (F') or 1 mM Ca²⁺ (F") on TIA1 granules density compared to those of control groups, respectively (for (F'): $n = 20, 23$ ROIs; for (F"): $n = 19, 27$ ROIs from three biological replicates). (G) In vitro sedimentation assay detected by SDS-PAGE showing the distribution of purified TIA1 protein (5 µM) in the supernatant (S) and pellet (P) fractions at the indicated concentration of CaCl₂. (G') Quantification of the sedimentation assay results in (G) ($n = 3$ biological replicates). Data represent mean ± SEM; two-tailed unpaired $t$-test in (F', F"); one-way ANOVA in (A', A", B', C', G'); *$P < 0.05$, ***$P < 0.001$, ns non-significant. See appendix for exact $P$ values. Source data are available online for this figure.

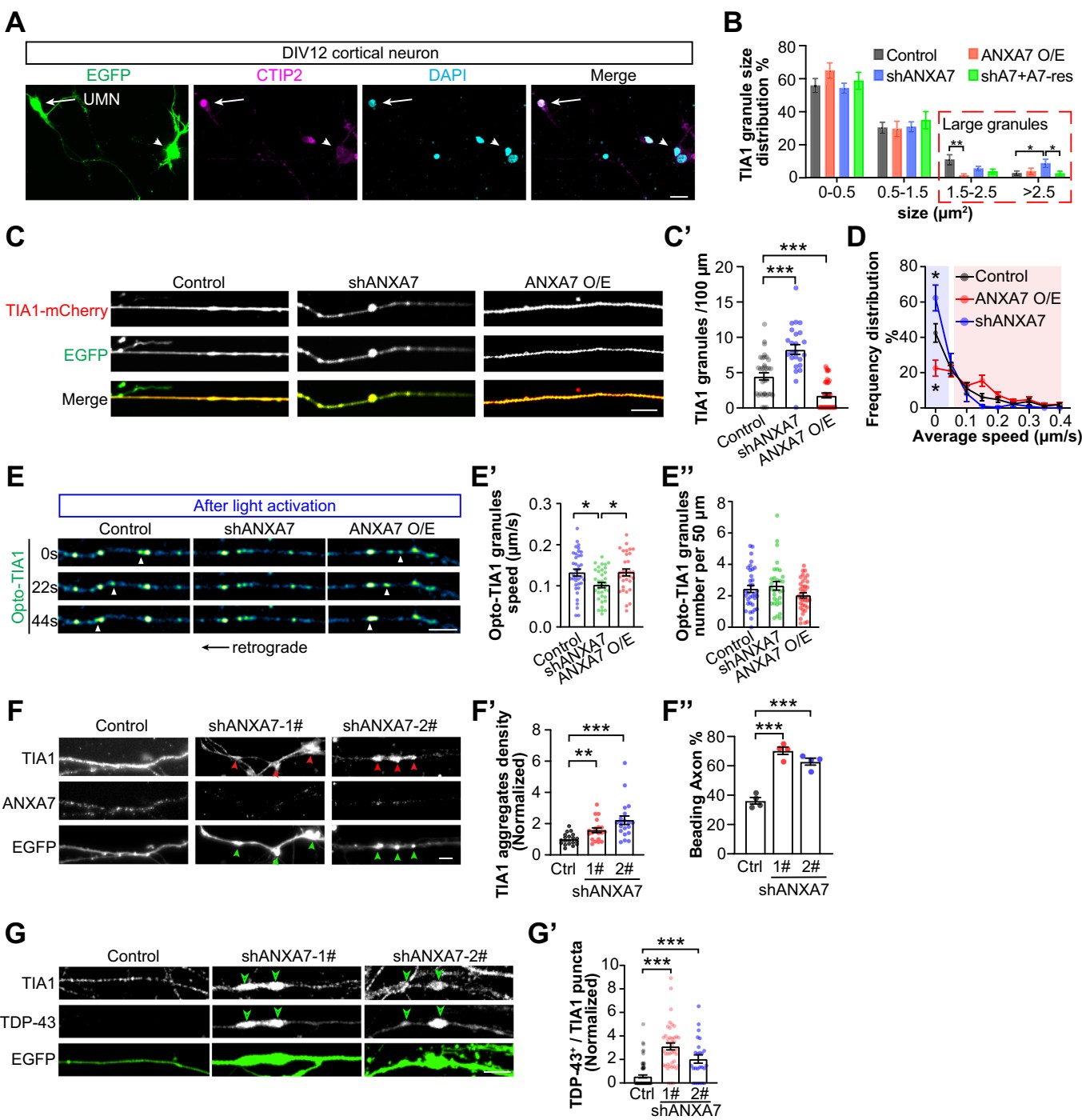

**Figure EV4. ANXA7 regulates TIA1 axon trafficking and LLPS.**

(A) Identification of upper motor neurons (UMNs) in DIV12 cortical cultures. CTIP2-positive UMNs are indicated by arrows; non-UMNs by arrowheads. Scale bar = 20 μm. (B) Related to Fig. 6A,A'. Quantification of TIA1 granule size from (Fig. 6A, **HN**). The red dotted box indicates large granules (≥2 μm², circularity 0.6–1) ($n$ = 43, 43, 46, 40 axons from three biological replicates). (C) Distribution of TIA1-mCherry in axons of DIV11 rat hippocampal neurons with endogenous ANXA7 knockdown (shANXA7) or ANXA7-EGFP overexpression (ANXA7 O/E). Scale bar = 10 μm. (C') Quantification of TIA1 aggregates per 100 μm axon under conditions in (C) ($n$ = 34, 24, 35 axons from four biological replicates). (D) Distribution of the EGFP-TIA1 granule speeds from (Fig. 6C). Blue shadow indicates stationary granules (≤0.05 μm/s), red shading indicates mobile (>0.05 μm/s) granules ($n$ = 21, 23, 23 axons from three biological replicates). (E) Key frames from time-lapse images of Opto-TIA1 in DIV9 rat hippocampal neurons with endogenous ANXA7 knocked down (shANXA7) or ANXA7-EGFP overexpression (ANXA7 O/E), showing retrograde trafficking of light-induced Opto-TIA1 granules after 11–20 min blue light exposure. Arrowheads indicate mobile Opto-TIA1 granules, and arrows indicate the retrograde direction. Scale bar = 10 μm. (E'-E") Quantification from (E), showing the speed (E') and density (E") of Opto-TIA1 granules in the axons of indicated groups (E': $n$ = 37, 36, 31 granules; E": $n$ = 34, 30, 36 axons; all from three biological replicates). (F) IF staining images of TIA1 and ANXA7 in axons of DIV9 rat HN with ANXA7 knockdown using two shRNA sequences (shANXA7-1# and 2#). EGFP shows axon morphology, with arrowheads indicating beading structures. Scale bar = 5 μm. (F'-F") Quantification of (F), showing the density of TIA1 aggregates (F') and the percentage of beading axons (F") (F': $n$ = 20 axons from four biological replicates; F": $n$ = 4 biological replicates). (G) Distribution of endogenous TIA1 with TDP-43 in axons of HN. EGFP depicts axon morphology. Scale bar = 10 μm. (G') Quantification from (G) showing the number of TIA1 puncta co-localized with TDP-43 per 100 μm axon ($n$ = 70, 42, 25 axons from four biological replicates). Data represent mean ± SEM; one-sample $t$-test in (F'); two-tailed unpaired $t$-test in (B); one-way ANOVA in (C', D, E', E", F", G'); *$P$ < 0.05, **$P$ < 0.01, ***$P$ < 0.001. See appendix for exact $P$ values. Source data are available online for this figure.

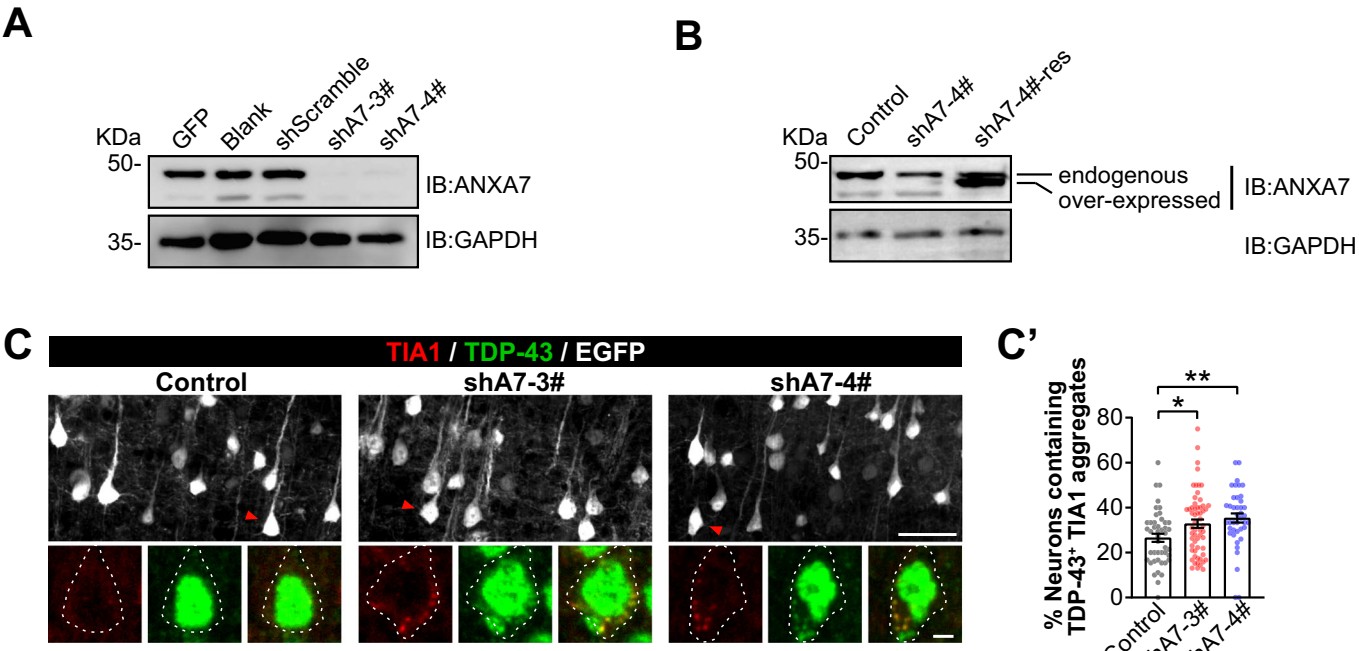

**Figure EV5.  ANXA7 knockdown leads to TIA1 aggregation in layer V neurons of the mouse motor cortex.**

(A) Western blot validating the knockdown efficiency of two different shRNA sequences targeting ANXA7 in mouse brains (shA7-3# and 4#). shRNA sequences are available in Table EV1. (B) Western blot validating the rescue efficiency of shA7-4#-res, which overexpressing rat ANXA7 resistant against shANXA7-4# (targeting the mouse intron sequence) knockdown of endogenous ANXA7 in cultured mouse cortical neurons. (C) Confocal images of P60 mouse cortex showing TIA1 and TDP-43 IF in layer V neurons. Infected neurons marked by EGFP expression, the neurons pointed by red arrows are amplified in lower panels. Dotted lines depict soma shapes. Scale bars = 50 μm (top), 5 μm (bottom). (C') Percentage of neurons containing TDP-43$^+$ TIA1 aggregates ($n$ = 45, 58, 41 ROIs from three mices). Data represent mean ± SEM, one-way ANOVA in (C'); *$P < 0.05$, **$P < 0.01$. See appendix for exact $P$ values. Source data are available online for this figure.

