## [Peer Review File · The EMBO Journal]

Annexin A7 Enhances TIA1 Axonal Trafficking to Counteract Pathological Aggregation in Neurons

Yu Feng, Tongshu Luan, Zhenda Zhang, Wei Wang, Yuanyuan Chu, Sijia Wan, Xiaorong Pan, Jie Li, Yifan Liu, Yaqian Xu, Kun Dou, and Tong Wang

Corresponding author(s): Tong Wang (wangtong@shanghaitech.edu.cn)

Review Timeline:

Submission Date:	9th Nov 24
Editorial Decision:	21st Feb 25
Revision Received:	16th Jul 25
Editorial Decision:	2nd Sep 25
Revision Received:	5th Sep 25
Accepted:	29th Sep 25

Editor: *Cornelius Schneider*

Transaction Report:

Dear Dr. Wang,

Thank you for submitting your manuscript for consideration by the EMBO Journal and for sharing a preliminary revision plan with me.

Based on your willingness to engage in a major revision as indicated during the pre-decision consultation, I would like to invite you to submit a revised version of the manuscript, addressing the comments of all three reviewers as outlined in your preliminary point-by-point response. I should add that it is EMBO Journal policy to allow only a single round of revision, and acceptance of your manuscript will therefore depend on the completeness of your responses in this revised version. If you have any additional questions or want to discuss the revisions further, I am happy to do so by email or video conferencing.

We generally allow three months as standard revision time, which can be extended to 6 months in case of major revisions, such as the experiments required here. As a matter of policy, competing manuscripts published during this period will not negatively impact on our assessment of the conceptual advance presented by your study. However, we request that you contact the editor as soon as possible upon publication of any related work, to discuss how to proceed. Should you foresee a problem in meeting the deadline, please let us know in advance and we may be able to grant an extension.

Thank you for the opportunity to consider your work for publication. I look forward to your revision.

Yours sincerely,

Cornelius Schneider, PhD
Editor
The EMBO Journal
c.schneider@embojournal.org

We realize that it is difficult to revise to a specific deadline. In the interest of protecting the conceptual advance provided by the work, we recommend a revision within 3 months (22nd May 2025). Please discuss the revision progress ahead of this time with the editor if you require more time to complete the revisions. Use the link below to submit your revision:

Referee #1:

In their study, Feng et al identify a functional association between the neuronal RBP TIA1 and ANXA7 for axonal trafficking of RNPs. They observed that TIA1 is present in axons and predominantly retrogradely transported. They identified the adapter protein ANXA7 as TIA1 interactor and found that ANXA7 mediates the association of TIA1 with dynein (DIC1B). ANXA7 undergoes LLPS, which is further stimulated by Ca²⁺ and can also trigger TIA1 droplet formation. Importantly, Ca²⁺ reduced the association of ANXA7 and, consequently, TIA1 with DIC1B. In cultured neurons, Ca²⁺ elevation through depolarization induced the aggregation of TIA1 and ANXA7 in axons. Knockdown of ANXA7 increased TIA1 granule size and reduced TIA1 motility while ANXA7 overexpression had the opposite effects, preventing TIA1 aggregation and increasing its axonal motility. Both in vitro and in vivo, ANXA7 knockdown induced cytoplasmic TDP-43 aggregation and caused axon degeneration. Thus, ANXA7 plays a vital function for maintaining axon integrity via supporting TIA1-mediated retrograde axonal transport. The manuscript is logically structured and the data are presented well. The authors used a range of sophisticated techniques with appropriate controls to support their findings. The results are of high interest in the field and extend current models of axonal RBP functioning. However, additional data need to be provided verifying the existence of endogenous TIA1-ANXA7-DIC1B complexes.

Major:

The authors used GST-TIA1 to identify TIA1 interactors in P14 rat brain. However, ANXA7 was one of the weakest interactors (Fig. 2B) and DIC1B was not detectable in the interactome data. The validation of the TIA1-ANXA7-DIC1B interaction was subsequently carried out using overexpression constructs in HEK293 cells or neurons. However, to substantiate the findings of the paper, it would be necessary to validate the interactions between endogenous TIA1, ANXA7 and DIC1B rather than overexpressed fusion proteins. Particularly, this interaction needs to be confirmed in axons. The authors performed a FRET assay in Fig. 2O using overexpressed proteins. For endogenous proteins, the axonal interactions could be investigated by immunoprecipitation from axonal lysates or by a proximity ligation assay (PLA).

The authors suggest that the retrograde transport of TIA1 is mediated via its interaction with DIC1B. Does knockdown of DIC1B (or other components of the dynein complex) affect retrograde TIA1 transport?

Minor:

The authors might consider using nocodazole to disrupt microtubules as additional control in Fig. 1. This should prevent axonal transport and reduce the motility of TIA1 particles.

Fig. 1A: The specificity of the CY5-UTP signal is unclear as it appears outside cell bodies and axons. Can the authors clarify?

Fig. 1C: What is the GFP signal, is it fused to another protein or just overexpressed as marker? This needs to be clarified in the text and figure legend. Also, the quality of the compartmentalized neuron culture is not very convincing, there are only very few axons crossing to the axonal side.

Fig. 1P: Does the time indicate blue light activation? If so, wouldn't it make sense to follow newly formed Opto-Tia1 granules rather than pre-existing ones?

Fig. 1G and J legend: it should be "Percentage" not "Ratio"

Fig. 1M and N: Typo "heterogeneity"

Fig. 1O: Opto-Tia1 is fused to mCherry but shown in green

sFig. 2B: There is no HA-TIA1 band present in the second lane in the input.

Fig. 2M: the control values are all 1, therefore a one sample t-test with a hypothetical mean value of 1 should be used

The authors show that Ca²⁺ triggers ANXA7 LLPS. Is this effect specific for Ca²⁺? This could be tested with other ions such as Na⁺ and Mg²⁺.

Define "PI" (propidium iodide) in the text

Referee #2:

The authors undertake a very thorough analysis of Annexin A7 (ANXA7) as an adaptor for retrograde transport of T-cell intracellular antigen 1 (TIA1) containing RNPs by linking TIA1 to the dynein motor. The work is overall well done, despite the caveats noted below, however the novelty of the finding is not high. The groups of Michael Ward and Jennifer Lippincot-Schwartz published a similar role for Annexin 11 in linking RNPs to axonal transport mechanisms (Liao et al., Cell, 2019), and there have been several follow-up studies since then. Others have shown direct interactions of RNA binding proteins to kinesin complexes. Hence, the question is whether the findings presented here move the field forward sufficiently to justify publication in EMBO Journal. My personal assessment is that more is needed, specifically to address the issue of the molecular/cellular role of retrograde transport of TIA1 containing RNPs.

1) Specifically, does the mechanism enable a functional role for this retrogradely transported RNP (most axonal RNP transport is thought to be in the opposite direction, to provide RNAs for axonal functions)? If the authors characterize the RNA contents of this RNP, that might shed light on whether it is a functional complex, or is it a "garbage disposal" or recycling route for RNAs used and no longer needed in the axon.

2) Another major issue is that most of the analyses were done in cultured hippocampal neurons, while the in vivo analysis in the last figure was on motor neurons. Some key experiments from the first part of the manuscript should be replicated in motor neurons to bridge this gap.

3) A third major issue concerns Figure 6 - there is no control for potential off-target effects of the shRNAs. Key aspects of this figure must be repeated with co-expression of an shRNA-resistant form of TIA1, to verify that shRNA effects are attenuated or reversed.

Additional specific issues:

1) Figure 2 - Panels 2G and 2K require quantification with statistics, to support the claims in the text of significant differences. More generally, the overall data as shown in the figure cannot discriminate between ANXA7 increasing affinity of a weak interaction between dynein and TIA1, versus it acting as an obligatory linker, as suggested by the authors. Cross-linking experiments to test for direct interactions between the three partners might be informative here.

2) Figure 3 is a weak point in the story overall. At first, I was puzzled why the blot in 3B does not reflect the graph in 3C, until I realized that 3C is plotted with 50% as baseline - this is truly misleading and inappropriate. It is difficult to believe that shifts from 51% to 54-55% are biologically meaningful, even if they happen to pass a statistical test.

3) The claims for effects of elevated calcium (Fig. 4) rely entirely on challenges with high K⁺, which is an artificial challenge likely to have numerous additional effects. The authors should use other approaches to induce calcium elevation to validate key findings, and/or attenuate calcium effects by use of chelators, to strengthen this aspect of the study.

4) Figure 5 - panel B does not show what it is claimed to show in the text.

5) Discussion - there are additional cases of direct motor complex-RNP interactions in the literature that are not cited here, e.g. ZBP1/PAT1 with kinesin-1 (Wu et al., 2020), APC/KAP3 with kinesin-2 (Baumann et al, 2020), nucleolin-GAR with kinesins (Doron-Mandel et al, 2021) etc.

Referee #3:

In the manuscript "Axon Trafficking Counteracts Aberrant Protein Aggregation in Neurons" by Feng Y. et al., the authors identify Annexin A7 as a novel adaptor that facilitates the interaction and retrograde axonal transport of TIA1 positive RNPs with dynein intermediate chain (DIC). The authors provide evidence that Annexin A7 also regulates TIA1 RNP liquid condensate properties and transport in a calcium dependent manner. Although previous studies had shown that the cargo binding tail region of KIF5 interacts with several RNA-binding proteins, facilitating their transport, only a few adapters linking RNP granules to dynein or kinesins have been identified in mammalian neurons to date. Several recent studies indicate that RNP granules can also "hitchhike" on membrane-bound organelles. However, the relative contribution of these two mechanisms of RNP transport in the axon is not entirely clear. In this context, the findings of (1) little to no co-trafficking of TIA1 RNP granules with the subset of membrane bound organelles that were examined; (2) identification of Annexin A7 as a linker of TIA1 and DIC1B; and (3) data that of Annexin A7 as a modulator of TIA1 condensate dynamic properties are novel and of broad interest to neuronal cell biology and neurodegeneration fields. The experiments are mostly well-controlled, though some additional experiments and clarification of the data presented are needed (please see below). Similarly, although the conclusions are generally supported by the data, there are specific claims that need to be softened (or additional data needed to substantiate these). Overall, the manuscript would be suitable for publication in EMBO, provided the following points are addressed and the new data confirm the

initial findings / support the conclusions:

1. The title suggests that axon trafficking is a general mechanism to counteract abnormal protein aggregation in neurons - while this may be true, the manuscript only looks at specific examples of TIA1 and the role of Annexin A7. Therefore, a more specific title would more accurately represent the manuscript.
2. In Fig.1 various n's are listed - in most cases n seems to represent the number of TIA1 granules analyzed. However, it is unclear how many neurons were analyzed and from how many independent replicates? In the graphs, what does each individual data point represent?
3. For timelapse imaging montages in Fig. 1H and 1I, the time intervals shown are too far apart; with the 200 and 500+ second intervals between frames shown, it is difficult to know if the granules are the same or different objects in each image. Also, the acquisition frame rates for live cell imaging experiments are not clear from the description in the Methods.
4. In lines 125-129, the authors claim: "These results suggest that the axon trafficking of TIA1 granules is unlikely to depend on membranous axon carriers, instead occurring primarily via a direct link with the retrograde motor dynein". The authors have not examined a complete list of membrane-bound organelles, including mitochondria and non-acidified endosome populations, so this statement should be softened or additional experiments should be performed with early and late endosomes, mitochondria, etc.
5. There is no description of the BioGRID analysis performed in the methods or anywhere in the text. Supplementary data tables S1 and S2 are described but were not provided for review.
6. In line 144, the authors write, "Collectively, most TIA1 granules in axons are retrogradely transported RNPs that are directly linked to dynein." However, thus far, the data presented only show co-trafficking and not a direct link, so this statement should be softened. Furthermore, the next set of experiments with GST pull down followed by proteomic mass spec analysis shows an interaction, but again, this is not necessarily a "direct" link.
7. In fig. 3, 1mM Ca²⁺ is not a physiologic concentration, but this is a minor point since additional work is also shown in the neurons (fig. 4).

Referee #1 (Report for Author)

In their study, Feng et al identify a functional association between the neuronal RBP TIA1 and ANXA7 for axonal trafficking of RNPs. They observed that TIA is present in axons and predominantly retrogradely transported. They identified the adapter protein ANXA7 as TIA1 interactor and found that ANXA7 mediates the association of TIA1 with dynein (DIC1B). ANXA7 undergoes LLPS, which is further stimulated by Ca^{2+} and can also trigger TIA1 droplet formation. Importantly, Ca^{2+} reduced the association of ANXA7 and, consequently, TIA1 with DIC1B. In cultured neurons, Ca^{2+} elevation through depolarization induced the aggregation of TIA1 and ANXA7 in axons. Knockdown of ANXA7 increased TIA1 granule size and reduced TIA1 motility while ANXA7 overexpression had the opposite effects, preventing TIA aggregation and increasing its axonal motility. Both *in vitro* and *in vivo*, ANXA7 knockdown induced cytoplasmic TDP-43 aggregation and caused axon degeneration. Thus, ANXA7 plays a vital function for maintaining axon integrity via supporting TIA1-mediated retrograde axonal transport. The manuscript is logically structured and the data are presented well. The authors used a range of sophisticated techniques with appropriate controls to support their findings. The results are of high interest in the field and extend current models of axonal RBP functioning. However, additional data need to be provided verifying the existence of endogenous TIA1-ANXA7-DIC1B complexes.

We sincerely thank the reviewer for their evaluation and constructive feedback on our manuscript. In response, we have addressed all the concerns raised by the reviewer through a combination of new experiments, data reanalysis, and textual revisions, as outlined in our point-by-point rebuttal below.

Major:

1. The authors used GST-TIA1 to identify TIA1 interactors in P14 rat brain. However, ANXA7 was one of the weakest interactors (Fig. 2B) and DIC1B was not detectable in the interactome data. The validation of the TIA1-ANXA7-DIC1B interaction was subsequently carried out using overexpression constructs in HEK293 cells or neurons. However, to substantiate the findings of the paper, it would be necessary to validate the interactions between endogenous TIA1, ANXA7 and DIC1B rather than overexpressed fusion proteins. Particularly, this interaction needs to be confirmed in axons. The authors performed a FRET assay in Fig. 2O using overexpressed proteins. For endogenous proteins, the axonal interactions could be

investigated by immunoprecipitation from axonal lysates or by a proximity ligation assay (PLA).

We appreciate the reviewer’s insightful feedback. To address this critical concern, we performed proximity ligation assay (PLA) experiments to assess the interactions between endogenous TIA1, ANXA7, and DIC1B in cultured cortical neurons. Specifically, we examined the following:

(1) Validate the direct interactions (TIA1/ANXA7 and ANXA7/DIC1B): We further validated the endogenous interactions between TIA1 and ANXA7, as well as ANXA7 and DIC1B, using PLA in axons. We found knockdown of ANXA7 significantly reduced PLA signal for both pairs, supporting that ANXA7 interacts with both TIA1 and DIC1B.

These results are included as Fig. EV2H–I’ and described in the Results (lines 273–276) in the revised manuscript.

(2) TIA1/DIC1B interaction under ANXA7 manipulation: We tested whether ANXA7 influences the interaction between TIA1 and DIC1B by comparing PLA signal densities under different ANXA7 expression conditions. As shown below, knockdown of endogenous ANXA7 led to a significant reduction in PLA signal in axons, while overexpression of siRNA-resistant ANXA7 restored the PLA signal between TIA1 and DIC1B. These data suggest that ANXA7 significantly enhances the interaction between TIA1 and DIC1B in axons.

This analysis has been included as Fig. 2K, K’, with text added to the Results section (lines 276–280) of the updated manuscript.

(3) Absence of DIC1B in GST-TIA1 interactome: We appreciate the opportunity to clarify why DIC1B was not detected in our GST-TIA1 pulldown mass spectrometry data. The absence of DIC1B from the GST-TIA1 interactome likely reflects that the TIA1-DIC1B interaction depends strongly on ANXA7 acting as an affinity enhancer. Without sufficient ANXA7, this association is too transient or weak for efficient retention on GST-TIA1 beads, thereby falling below the detection threshold of mass spectrometry.

We have included a discussion of this point in the revised Discussion section (lines 561–565).

2. The authors suggest that the retrograde transport of TIA1 is mediated via its interaction with DIC1B. Does knockdown of DIC1B (or other components of the dynein complex) affect retrograde TIA1 transport?

We thank the reviewer for this constructive and insightful comment. We conducted additional experiments to test whether DIC1B is required for retrograde axonal trafficking of TIA1 granules in cultured neurons.

First, we screened multiple shRNA constructs targeting endogenous DIC1B, the neuron-specific isoform of the dynein intermediate chain (1). Based on knockdown efficiency in rat cortical neurons, we selected two effective sequences (shDIC1B-1# and shDIC1B-2#) for further experiments. This result is now included in the revised manuscript as Fig. EV1H.

Fig. EV1H

Next, we knocked down endogenous DIC1B and assessed its impact on retrograde axonal trafficking of TIA1 granules. Both shDIC1B constructs significantly reduced the average trafficking speed, supporting a critical role for DIC1B in this process. These data are presented in Fig. 1G, G' and described in the Results section (lines 148–151) of the revised manuscript.

Fig. 1G

G'

Minor:

3. The authors might consider using nocodazole to disrupt microtubules as additional control in Fig. 1. This should prevent axonal transport and reduce the motility of TIA1 particles. We thank the reviewer for this valuable suggestion. We performed nocodazole treatment experiments to assess the impact of microtubule disruption on TIA1 granule axon trafficking. As shown below, nocodazole treatment markedly reduced the trafficking speed of TIA1 granules in axons, confirming that their transport is microtubule-dependent. These results have been included in the revised manuscript as Fig. EV11, I', and are described in the Results section (lines 151–152).

4. Fig. 1A: The specificity of the CY5-UTP signal is unclear as it appears outside cell bodies and axons. Can the authors clarify?

We thank the reviewer for raising this important concern. Below, we clarify the observed localization of CY5-UTP and outline an additional experiment we performed to validate the specificity of the signal:

(1) Clarification of CY5-UTP localization: CY5-UTP and EGFP-TIA1 were co-transfected, but due to variable transfection efficiencies, their signals may arise from different cells. CY5-UTP, having a much smaller molecular weight, transfects more efficiently than the EGFP-TIA1 plasmid. As a result, the observed CY5-UTP signal outside axons or neuronal cell bodies may originate from CY5-UTP-transfected but EGFP-TIA1-non-transfected adjacent neurons or non-neuronal cells.

(2) Validation of CY5-UTP specificity: As shown below, to further assess the specificity of CY5-UTP labelling for RNPs, we performed colocalization analyses of Cy5-UTP with several well-characterized endogenous RBPs that label distinct types of RNP granules: TIA1, Staufen1, DCP1A, and FMRP. We observed significant colocalization between CY5-UTP and each of these markers, confirming that CY5-UTP reliably labels RNP structures.

Fig. EV1A

These results are now included in the revised manuscript as Fig. EV1A, with corresponding description in the Figure legends.

5-1. Fig. 1C: What is the GFP signal, is it fused to another protein or just overexpressed as marker? This needs to be clarified in the text and figure legend.

We thank the reviewer for this insightful comment. In the original Fig. 1C, the GFP signal corresponds to a co-expressed cytosolic EGFP vector, which serves as a morphological marker to visualize neuronal shape and identify transfected cells. It is not fused to any other protein. We have revised the Results section (lines 119–121) to clarify this information.

5-2. Also, the quality of the compartmentalized neuron culture is not very convincing, there are only very few axons crossing to the axonal side.

We appreciate the reviewer for raising this concern. We would like to clarify that the limited number of visible axons in the axonal compartment reflects *low transfection efficiency* rather than poor culture quality. In neurons cultured in microfluidic devices, Lipofectamine-based transfection typically yields efficiencies below 1%, resulting in only a small fraction of axons that are EGFP-positive. In contrast, the total number of axons, including those from untransfected neurons, is substantially higher, indicating that the overall neuronal culture remains healthy and dense.

To illustrate this, we have added a new panel (Appendix Fig. S1C) showing the sparse distribution of TIA1-mCherry-expressing axons (boxes 3#, right) and their corresponding somas (boxes 1#, left) within an otherwise dense neuronal population marked by the microtubule marker \$\beta\$ -Tubulin III.

Appendix Fig. S1C

In addition, Fig. EV1B demonstrates numerous untransfected axons labelled by LysoTracker (magenta puncta, arrowheads) in the axonal compartment. Although only one EGFP-positive axon is visible (arrows), the abundance of unlabelled axons further confirms the health and density of the cultures.

Fig. EV1B

These data have been incorporated into the revised manuscript (Appendix Fig. S1C, Fig. EV1B), and clarifying text has been added to the Results section (lines 121–123) to emphasize the sufficient quality of the microfluidic neuronal cultures.

6-1. Fig. 1P: Does the time indicate blue light activation?

We thank the reviewer for this important clarification request. The time labels in the original Fig. 1P indicate the starting frame of the video capturing Opto-TIA1 granule movement and do not represent the onset of blue light activation. The video was recorded after 19 minutes of continuous blue light exposure.

To avoid confusion, we have revised Fig. 1I', I'' by adding a timeline that clearly marks the onset of blue light activation as well as the start and end points of the video acquisition. The corresponding figure legend and Results section (lines 166–170) have been updated accordingly.

Additionally, to further clarify the timing of blue light activation throughout the manuscript, we have revised the panel labels in Fig. 2F, Fig. EV4E, and Appendix Fig. S1F, to explicitly include the label “After light activation.”

6-2. If so, wouldn't it make sense to follow newly formed Opto-Tia1 granules rather than pre-existing ones?

The marked granules (old Fig. 1P, arrows; now Fig. 1J, arrows) are indeed newly formed, light-induced Opto-TIA1 condensates, as very few micron-scale condensates were present prior to light activation. To support this, in the revised manuscript, we have added a pre-activation frame (new Fig. 1I', "Before"), clearly demonstrating the absence of large condensates before light exposure. To further clarify this point, we have revised the Results section (lines 166–170) as follows:

"Some of these light-induced Opto-TIA1 granules underwent rapid fusion, as indicated by the merging of two droplets (Fig. 1I', 1I''; Movie EV6; magenta- and green-arrowheads) into a larger granule (white-arrowheads). Interestingly, this fused granule subsequently underwent retrograde axon trafficking (white-arrowheads)."

7. Fig. 1G and J legend: it should be "Percentage" not "Ratio"

We thank the reviewer for pointing out this error. We have corrected the terminology in the figure legends of Fig. 1D', E', F' in the revised manuscript accordingly.

8. Fig. 1M and N: Typo "heterogeneity"

We appreciate the reviewer for noting these typographical errors. The misspelling of “heterogeneity” has been corrected in the revised version of the manuscript (Fig. 1H’, H’’), as shown below.

9. Fig. 1O: Opto-Tia1 is fused to mCherry but shown in green

We thank the reviewer for highlighting this important point. To avoid confusion, we have revised the pseudo-colouring of the original Fig. 1O to display the signal in red, accurately reflecting that Opto-TIA1 is fused to mCherry. The corrected panel has now been incorporated as Fig. 1I, I’’ in the revised manuscript.

10. sFig. 2B: There is no HA-TIA1 band present in the second lane in the input.

We thank the reviewer for noticing this error. The absence of the HA-TIA1 band in the last two lanes of the input in sFig. 2B was due to an *unintentional* shift of the masking box (red boxes) during figure assembly in Adobe Illustrator. As a result, the HA-TIA1 bands were mistakenly excluded. In the revised manuscript, we have corrected this issue as follows:

(1) **Updated panel:** The corrected sFig. 2B panel is now included, with the masking box repositioned to accurately display both HA-TIA1 bands. This updated figure has replaced the original version in Appendix Fig. S2B of the revised manuscript. In addition, the uncropped scans have been uploaded as source data for Appendix Fig. S2B.

(2) **Provided raw data:** To ensure clarity and transparency, we have uploaded the following files to the DRYAD server:

- The original and corrected Adobe Illustrator files of sFig. 2B.
- Full scans of the uncropped blots for all panels in sFig. 2B, with cropping boxes clearly indicated.

Reviewers can access these files at:

http://datadryad.org/stash/share/7RtZ9VtS5_CUy09HJQ7LGhTrObd-B9w2XvpQ0TYNR2A

11. Fig. 2M: the control values are all 1, therefore a one sample t-test with a hypothetical mean value of 1 should be used.

We sincerely thank the reviewer for pointing out this issue. We agree that a one-sample *t*-test is more appropriate for this analysis. Using our original dataset (old Fig. 2M), the one-sample *t*-test yielded:

Myc-ANXA7: $p = 0.0991$

shANXA7: $p = 0.073$

While these values suggest a trend toward an effect of ANXA7 on the DIC–TIA1 interaction, they do not reach statistical significance.

To further support our conclusion, we performed two additional biological replicates. With the updated dataset, the revised one-sample *t*-test results are:

Myc-ANXA7: $p = 0.0267$

shANXA7: $p = 0.0474$

These updated results support a statistically significant role for ANXA7 in neurons.

Fig. 2H

H'

We have incorporated the revised statistical analysis into the manuscript as Fig. 2H'.
Additionally, detailed statistical information has been provided in the Appendix.

12. The authors show that Ca^{2+} triggers ANXA7 LLPS. Is this effect specific for Ca^{2+} ? This could be tested with other ions such as Na^+ and Mg^{2+} .

We thank the reviewer for this insightful suggestion. We performed additional experiments testing the effects of Na^+ and Mg^{2+} on ANXA7 LLPS using the same sedimentation assay employed for Ca^{2+} in our original study. We found altering Na^+ concentrations (0–500 mM; new Fig. EV3B, B') or Mg^{2+} concentrations (0–10 mM; new Fig. EV3C, C') did not alter LLPS of purified ANXA7. In contrast, Ca^{2+} robustly promoted LLPS (new Fig. 3B, B'). These results suggest that the phase separation of ANXA7 is specifically triggered by Ca^{2+} .

Fig. EV3B

B'

C

C'

We have now incorporated these findings into the revised manuscript as Fig. EV3B–C', and updated the Results section (lines 298–300).

See also our replies to R2, Q5 and R3, Q7 for more details.

13. Define "PI" (propidium iodide) in the text

We thank the reviewer for pointing out this omission. In the revised manuscript, we have addressed this by defining "PI" as *propidium iodide* in the updated figure legend for Fig. 6G.

Referee #2 (Report for Author)

The authors undertake a very thorough analysis of Annexin A7 (ANXA7) as an adaptor for retrograde transport of T-cell intracellular antigen 1 (TIA1) containing RNPs by linking TIA1 to the dynein motor. The work is overall well done, despite the caveats noted below, however the novelty of the finding is not high. The groups of Michael Ward and Jennifer Lippincott-Schwartz published a similar role for Annexin 11 in linking RNPs to axonal transport mechanisms (Liao et al., Cell, 2019), and there have been several follow-up studies since then. Others have shown direct interactions of RNA binding proteins to kinesin complexes. Hence, the question is whether the findings presented here move the field forward sufficiently to justify publication in EMBO Journal. My personal assessment is that more is needed, specifically to address the issue of the molecular/cellular role of retrograde transport of TIA1 containing RNPs. We sincerely thank the reviewer for acknowledging the thoroughness of our work and for the constructive comments to improve it.

We fully recognize the important contribution by Liao et al. (2), who identified Annexin A11 as a membrane-tethered adaptor linking RNPs to the axonal transport machinery. However, our study differs in several key aspects: *First*, we identify a distinct subset of RNPs, specifically TIA1 granules, which undergo retrograde axon trafficking independent of membrane tethering. *Second*, we show that ANXA7 enhances the interaction between TIA1-RNPs and the dynein subunit DIC1B, facilitating their retrograde transport. *Third*, and in contrast to the Ca²⁺-activated nature of ANXA11, we demonstrate that the enhancer function of ANXA7 is inactivated by Ca²⁺ overload, which **disrupts** the ANXA7–TIA1–dynein interaction and compromises RNP trafficking in axons.

Together, our findings advance the field by (i) uncovering a membrane-independent mechanism of retrograde RNP axon trafficking, (ii) elucidating a Ca²⁺ overload triggered switch-off mechanism, and (iii) linking disruption of this pathway to axon degeneration. In response to the reviewer's suggestion, we have revised the manuscript, as detailed in the following point-by-point rebuttal, to more clearly highlight these conceptual advances and their implications for axon health.

1. Specifically, does the mechanism enable a functional role for this retrogradely transported RNP (most axonal RNP transport is thought to be in the opposite direction, to provide RNAs for axonal functions)? If the authors characterize the RNA contents of this RNP, that might shed light on whether it is a functional complex, or is it a "garbage disposal" or recycling route for RNAs used and no longer needed in the axon.

We thank the reviewer for raising this insightful question regarding the functional significance of retrogradely transported RNPs. To directly address this point, we performed additional experiments to characterize both the fate of these RNPs and their mRNA cargo composition: **(1) Pulse-chase labelling to define the fate of retrograde RNPs.**

We adapted the SYTO™ RNASelect Green Fluorescent Cell Stain (RNA-Select, Cat# S32703, ThermoFisher), an efficient RNA-specific dye that permeabilizes the plasma membrane to rapidly label intracellular RNA. Using microfluidic devices that physically separate axons and somas, we selectively applied RNA-Select to axonal compartments to establish a two-step pulse-chase assay (new Fig. 5A), similar to our previously described retrograde tracing approach (3). We observed that axon-derived RNA was labelled initially in axon terminals (new Fig. 5B, top) and then accumulated in soma of live neurons (new Fig. 5B, bottom).

In axons, the trajectories of RNA-Select-labelled RNA transcripts were clearly visualized. Tracking their movement revealed that most RNPs were either retrogradely transported or stationary during the observation window (new Fig. 5C, C'; Mov. EV12), confirming their retrograde identity.

Finally, to assess their fate, we fixed neurons 1 hour after labelling and performed colocalization analyses in the soma. The majority of retrogradely transported RNA localized to degradative compartments, with 33.8% colocalizing with lysosomes (LAMP1) and 21.6% with autophagosomes (LC3), while only a small fraction associated with early endosomes (2.5%)

EEA1) or processing bodies (1.4% DCP1A) (new Fig. 5D, D'). These findings suggest a “garbage disposal” role for axon-derived RNA transported to the soma.

Fig. 5D

D'

(2) Characterizing the mRNA content of retrograde TIA1-RNPs.

We cultured ANXA7 knockdown and control neurons in Boyden chambers (Fig. 5E) to enrich axon-derived RNA transcripts, as described previously (4). Axon-enriched RNA from both groups was sequenced and compared (Fig. 5F, F'), identifying 232 transcripts with increased abundance in ANXA7 knockdown neurons. We validated three of these mRNAs implicated in neuronal development using RT-qPCR (Fig. 5F'').

Fig. 5E

F

F'

F''

Among the candidates, we focused on *Ryk* mRNA, previously identified as a TIA1 cargo (ENCSR057DWB; ENCSR623VEQ). *Ryk* encodes a receptor for Wnt5a that plays a critical role in Wnt5a-induced axon growth/repulsion (5, 6). We therefore used smFISH (7, 8) to examine axonal *Ryk* mRNA density in DIV12 hippocampal and cortical neurons. We found that ANXA7 knockdown significantly increased axonal *Ryk* mRNA puncta compared to control (new Fig. 5G, G'), while ANXA7 rescue restored *Ryk* mRNA levels to control levels.

Fig. 5G

G'

Together, these new data on RNA fate and content suggest that the retrograde RNPs likely serve as a degradative mechanism for RNAs that "no longer needed in the axon," highlighting the critical role of this pathway in maintaining neuronal homeostasis and health.

To address this major concern, we have added a new figure (Fig. 5) and corresponding results under the subtitle “Identify the fate and RNA composition of retrograde RNPs” (lines 376–412), along with an expanded discussion (lines 572–587) in the revised manuscript.

2. Another major issue is that most of the analyses were done in cultured hippocampal neurons, while the *in vivo* analysis in the last figure was on motor neurons. Some key experiments from the first part of the manuscript should be replicated in motor neurons to bridge this gap.

We thank the reviewer for raising this important point. To address this concern and strengthen the relevance between our *in vitro* and *in vivo* data, we have replicated two of the key functional experiments in cortical upper motor neurons (UMNs):

(1) Validation of UMN identity: We confirmed the identity of UMNs in cultured rat cortical neurons by immunostaining for CTIP2, a well-established UMN marker specifically labelling nuclei (new Fig. EV4A).

Fig. EV4A

(2) Replication of key functional experiments in UMNs: Using DIV12 cortical neurons with UMNs identified by CTIP2 staining, we performed the following experiments in CTIP2-positive UMNs:

- *Axonal TIA1 aggregation upon modulation of ANXA7 expression:*

Consistent with results in hippocampal neurons (HNs) (Fig. 6A, top), altering ANXA7 levels in UMNs (Fig. 6A, bottom; uncropped images in Appendix Fig. S4A) produced similar effects on TIA1 granule aggregation in axons. Specifically, ANXA7 overexpression reduced TIA1 aggregation, whereas ANXA7 knockdown increased aggregation, which was rescued by expressing an shRNA-resistant ANXA7 variant (Fig. 6A’).

Fig. 6A

A'

Appendix Fig. S4A

- Axonal TIA1 co-aggregation with neurodegenerative markers:

Consistent with our observations in HNs (new Fig. 6E), ANXA7 knockdown in UMN axons significantly increased the accumulation of TIA1 aggregates colocalized with the neurodegeneration marker SQSTM1/p62 along axons (Fig. 6F, F'; uncropped images in Appendix Fig. S4B).

Fig. 6F

F'

Appendix Fig. S4B

Results from these two additional experiments demonstrate that the functional roles of ANXA7 in regulating TIA1 granule dynamics and preventing aggregation are conserved between hippocampal and cortical upper motor neurons, thereby bridging the gap between our *in vitro* and *in vivo* analyses.

These data are now incorporated into Fig. 6A, A', F, F'; Fig. EV4A; and Appendix Fig. S4A, B, and are described in the Results section (lines 418-428; 463-465) of the revised manuscript.

3. A third major issue concerns Figure 6 - there is no control for potential off-target effects of the shRNAs. Key aspects of this figure must be repeated with co-expression of an shRNA-resistant form of TIA1, to verify that shRNA effects are attenuated or reversed.

We thank the reviewer for raising this critical point. To address this concern, we have performed key *in vivo* rescue experiments using an shRNA-resistant form of ANXA7 (rather than TIA1, as clarified), to exclude their potential off-target effects:

(1) Validation of the rescuing effect of the ANXA7-res construct in mouse neurons: We overexpressed the wildtype rat ANXA7 homologue lacking the sequence targeted by shANXA7-4# to rescue the knockdown effect towards the endogenous ANXA7 in mouse neurons. As shown in the new Fig. EV5B, expression of this exogenous shRNA-resistant ANXA7 (ANXA7-res) restored overall ANXA7 levels in cultured mouse cortical neurons.

Fig. EV5B

(2) Rescue of neurodegenerative phenotypes caused by ANXA7 knockdown:

To assess functional rescue *in vivo*, we generated AAV co-expressing shANXA7-4# alongside the shRNA-resistant ANXA7 rat variant (ANXA7-res) (new Fig. 7A) and delivered to mouse brain via intracerebroventricular (ICV) injection at postnatal day 1 (P1) (new Fig. 7B).

Fig. 7A

B

Co-expression of ANXA7-res effectively rescued the neurodegenerative phenotypes induced by ANXA7 knockdown, including:

- behavioural deficits (Fig. 7B')
- loss of layer V cortical neurons (Fig. 7D, D')
- axon degeneration in the corticospinal tract (Fig. 7E, E')
- accumulation of pathological TIA1/p62 aggregates (Fig. 7F, F')
- activated microglia (Fig. 7G, G')

These findings demonstrate that the observed phenotypes are specifically targeted down-regulation of ANXA7 rather than off-target effects.

These data are now incorporated into the new Fig. 7 and Fig. EV5B, and are described in the Results section (lines 479–509) of the revised manuscript.

Additional specific issues:

4. Figure 2 - Panels 2G and 2K require quantification with statistics, to support the claims in the text of significant differences. More generally, the overall data as shown in the figure cannot discriminate between ANXA7 increasing affinity of a weak interaction between dynein and TIA1, versus it acting as an obligatory linker, as suggested by the authors. Cross-linking experiments to test for direct interactions between the three partners might be informative here. We thank the reviewer for these valuable suggestions. To address this critical concern, we have made the following revisions:

(1) Quantification of the original Fig. 2G and Fig. 2K: In the revised manuscript, we have performed new quantitative analyses for these panels, including:

- *A new statistical chart the original Fig. 2G (new Fig. 2G').* In addition, as we performed further repeats of this assay, we have replaced the previous Fig. 2G with a more representative case, while retaining the original as Appendix Fig. S2F.

- *A new statistical chart for Fig. 2K (new Appendix Fig. S2C').*

Appendix Fig. S2C

C'

These analyses confirm that ANXA7 expression significantly *enhances* the observed DIC–TIA1 interactions.

(2) Clarification that ANXA7 serves as an affinity enhancer in text:

We appreciate the reviewer’s important point regarding whether ANXA7 functions as an obligatory linker or an affinity enhancer. Our revised data of *in vitro* pull-down assay using purified recombinant proteins (new Fig. 2G, G’; Appendix Fig. S2F) show that a low baseline interaction between recombinant DIC1B and TIA1 is detectable even without ANXA7 (red arrows in new Fig. 2G, G’ and Appendix Fig. S2F), supporting the existence of a pre-existing weak association. In contrast, the addition of recombinant Myc-ANXA7 significantly increased the amount of TIA1 pulled down with DIC1B by over four-fold (new Fig. 2G’).

Fig.2G

G’

Appendix Fig. S2F

These findings demonstrate that ANXA7 strongly enhances this weak direct interaction *in vitro* rather than acting as an essential linker. Accordingly, we have revised the manuscript text throughout to describe ANXA7 as an *affinity enhancer* rather than an *obligatory linker* between TIA1 and dynein.

These updates are now incorporated into Fig. 2G’, Appendix Fig. S2C’, and the Results section (lines 215–217; 235–241) of the revised manuscript.

5. Figure 3 is a weak point in the story overall. At first, I was puzzled why the blot in 3B does not reflect the graph in 3C, until I realized that 3C is plotted with 50% as baseline - this is truly misleading and inappropriate. It is difficult to believe that shifts from 51% to 54-55% are biologically meaningful, even if they happen to pass a statistical test.

We thank the reviewer for this important critique. To address this concern, we optimized the assay and significantly improved its dynamic range, thereby strengthening the evidence:

(1) Optimization of sedimentation assay conditions:

- Following protocols described in (9) and (10), we increased the NaCl concentration from 50 mM to 100 mM and raised the centrifugation force from $14,000 \times g$ to $17,000 \times g$ to improve pellet separation.
- Following protocols described in (2), we also extended the Ca^{2+} treatment range up to 10 mM, which elicited a substantially stronger sedimentation response.

These adjustments markedly improved the assay's dynamic range and reproducibility, providing more robust evidence for Ca^{2+} -induced phase separation (please also see our responses to Reviewer 1, Q12, and Reviewer 3, Q7).

(2) Replotting and additional visualization:

Using this optimized sedimentation assay, we repeated the sedimentation experiments and replotted the pellet percentage on a full 0–100% scale (new Fig. 3B, B'). Across 0–10 mM Ca^{2+} , the fraction of ANXA7 in the pellet increased by over 30%, providing much stronger support for a biologically meaningful effect. To further illustrate this response, we added a new image panel showing visible emulsification of ANXA7 solutions upon Ca^{2+} elevation (new Fig. 3B, bottom).

These updates are incorporated into Fig. 3B, B', and the Results section (lines 296–298). We have also revised the Methods (lines 919–928) to describe the improved conditions in detail.

6. The claims for effects of elevated calcium (Fig. 4) rely entirely on challenges with high K^+ , which is an artificial challenge likely to have numerous additional effects. The authors should use other approaches to induce calcium elevation to validate key findings, and/or attenuate calcium effects by use of chelators, to strengthen this aspect of the study.

We thank the reviewer for raising this critical concern. To address it, we performed both an alternative Ca^{2+} elevation assay and a Ca^{2+} attenuation assay. The key findings are summarized below:

(1) Transient Ca^{2+} elevation induces reversible ANXA7/TIA1 aggregation in axons.

To more specifically assess the role of axonal Ca^{2+} elevation in ANXA7/TIA1 aggregation, we employed a microfluidic-based approach to generate transient, localized Ca^{2+} surges triggered by mild mechanical stress, as developed in our recent works ((11, 12); see also Methods, lines 755–766). By applying precisely controlled transverse microflows of culture medium selectively to axonal compartments in neurons cultured within our custom-designed Axon-on-a-Chip device (new Fig. 4H, H', “② Flux”), we induced a transient and focal Ca^{2+} increase in “beading” regions of axons. This method enables paired comparisons of LLPS dynamics before, during, and after Ca^{2+} elevation within the same axonal region of interest (ROI), thereby avoiding artifacts associated with high K^+ depolarization, which causes persistent and global Ca^{2+} elevation throughout the neuron.

This elevation reached ~2–3 fold over baseline and lasted less than 10 min before returning to resting levels (new Fig. 4I, I').

During this Ca^{2+} transient, we monitored the LLPS behaviour of ANXA7–mCherry and TIA1–mCherry, using co-transfected GCaMP6f as the real-time sensor to localize Ca^{2+} hotspots (new Fig. 4J, J'). Both proteins showed prominent accumulation at Ca^{2+} -elevated axon ROIs (Fig. 4J, arrowheads), as quantified by a significant increase in the heterogeneity index (new Fig. 4J', orange bars). Notably, once Ca^{2+} levels returned to baseline, the aggregation of ANXA7 and TIA1 decreased markedly, reducing the heterogeneity index (Fig. 4J', blue bars).

These new results provide spatially and temporally resolved evidence that the assembly and disassembly of ANXA7/TIA1 aggregates are directly linked to local Ca^{2+} fluctuations, further supporting our hypothesis that Ca^{2+} elevation drives their axonal aggregation.

These updates are incorporated into Fig. 4H–J' and the Results section (lines 359–374). We have also revised the Methods (lines 755–766) to describe the microfluidic assay in detail.

(2) Attenuation of Ca^{2+} elevation prevents ANXA7/TIA1 aggregation.

We also used the Ca^{2+} chelator EDTA to attenuate Ca^{2+} elevation. EDTA efficiently reduced the high K^+ -induced persistent Ca^{2+} rise (Appendix Fig. S3D–D''; new Fig. 4A').

Consistently, EDTA treatment suppressed the axonal aggregation of both ANXA7 and TIA1, as reflected by the significantly reduced heterogeneity index in ANXA7–mCherry neurons (new Fig. 4C–C'') and TIA1–mCherry neurons (new Fig. 4D–D'').

Together, these additional experiments provide strong evidence that elevated Ca²⁺ is both necessary and sufficient to drive TIA1/ANXA7 dynamics in axons.

These updates are incorporated into Fig. 4A'; Fig. 4C–D''; Appendix Fig. S3D–D'', and the Results section (lines 335–337; 342–344).

7. Figure 5 - panel B does not show what it is claimed to show in the text.

We thank the reviewer for pointing out this issue. In Fig. 5B, we intended to show that ANXA7 levels are negatively correlated with large TIA1 granule formation in axons: overexpression leads to fewer large TIA1 granules, whereas ANXA7 KD (shANXA7) results in more. We agree with the reviewer that our initial description in the text and Fig. 5B were not sufficiently clear. To address this concern, we have:

(1) Modified the Fig. 5B. We replaced the line chart with a grouped bar chart and explicitly defining the groups of large granules (size $\geq 2 \mu\text{m}^2$) using red dashed boxes to highlight the key groups for comparison (now Fig. EV4B), as shown below.

Old Fig. 5B

New Fig. EV4B

(2) **Added quantification of granule density:** To further clarify the effect, we quantified the density of large TIA1 granules (size $\geq 2 \mu\text{m}^2$) along axons by counting the number of large granules per 100 μm of axon length in both HNs and UMNs. This additional analysis is shown below and included as Fig. 6A' in the revised manuscript.

(3) **Revised the text for clarity.** We have updated the Results section to improve clarity and align with the revised figures. The new text reads:

"In both HNs (Fig. 6A, top) and UMNs (Fig. 6A, bottom; Appendix Fig. S4A), altering ANXA7 expression significantly affected the extent of TIA1 granule aggregation within axons. Overexpression reduced the accumulation of large granules ($\geq 2 \mu\text{m}^2$) (Fig. 6A, arrows; Fig. EV4B, boxed columns), decreasing both their percentage (Fig. EV4B) and density (Fig. 6A'). In contrast, knockdown of endogenous ANXA7 using shANXA7 led to a significant increase in large TIA1 granules within axons (Fig. 6A, arrows), elevating both their percentage (Fig. EV4B) and density (Fig. 6A'). Notably, overexpression of an shRNA-resistant ANXA7 variant (shA7+A7-res) reversed the phenotype caused by ANXA7 knockdown, reducing the size of TIA1 granules (Fig. EV4B; Fig. 6A, A')."

The new data are now incorporated into Fig. EV4B and Fig. 6A', and the revised Results text appears on lines 419–428 of the updated manuscript.

(8) Discussion - there are additional cases of direct motor complex-RNP interactions in the literature that are not cited here, e.g. ZBP1/PAT1 with kinesin-1 (Wu et al., 2020), APC/KAP3 with kinesin-2 (Baumann et al, 2020), nucleolin-GAR with kinesins (Doron-Mandel et al, 2021) etc.

We thank the reviewer for highlighting these important studies. In the revised manuscript, we have now cited these references as suggested, along with a recent review by Abraham & Fainzilber (2022), to provide a broader and more comprehensive context for axonal RNP transport mechanisms. The updated Discussion text (lines 545–552) now reads:

*“...ZBP1/PAT1 links β -actin mRNA to kinesin-1 (Wu, Zhou et al., 2020); nucleolin–GAR motifs engage multiple kinesins (Doron-Mandel et al., 2021); and splicing factor proline/glutamine-rich (SFPQ) binds kinesin light chain 1 (KLC1) of kinesin family member 5A (KIF5A) (Fukuda, Pazyra-Murphy et al., 2021). Particularly, recent advances in RNP transport in axons were summarized in (Abraham & Fainzilber, 2022). However, although the retrograde motor dynein has been shown to drive RBP trafficking in cultured *Drosophila* S2 cells or embryo lysates (McClintock et al., 2018, Sladewski et al., 2018), the mechanism underlying dynein-driven axon transport of RNPs in mammalian neurons remains elusive.”*

Referee #3 (Report for Author)

In the manuscript "Axon Trafficking Counteracts Aberrant Protein Aggregation in Neurons" by Feng Y. et al., the authors identify Annexin A7 as a novel adaptor that facilitates the interaction and retrograde axonal transport of TIA1 positive RNPs with dynein intermediate chain (DIC). The authors provide evidence that Annexin A7 also regulates TIA1 RNP liquid condensate properties and transport in a calcium dependent manner. Although previous studies had shown that the cargo binding tail region of KIF5 interacts with several RNA-binding proteins, facilitating their transport, only a few adapters linking RNP granules to dynein or kinesins have been identified in mammalian neurons to date. Several recent studies indicate that RNP granules can also "hitchhike" on membrane-bound organelles. However, the relative contribution of these two mechanisms of RNP transport in the axon is not entirely clear. In this context, the findings of (1) little to no co-trafficking of TIA1 RNP granules with the subset of membrane bound organelles that were examined; (2) identification of Annexin A7 as a linker of TIA1 and DIC1B; and (3) data that of Annexin A7 as a modulator of TIA1 condensate dynamic properties are novel and of broad interest to neuronal cell biology and neurodegeneration fields. The experiments are mostly well-controlled, though some additional experiments and clarification of the data presented are needed (please see below). Similarly, although the conclusions are generally supported by the data, there are specific claims that need to be softened (or additional data needed to substantiate these). Overall, the manuscript would be suitable for publication in EMBO, provided the following points are addressed and the new data confirm the initial findings / support the conclusions:

We thank the reviewer for the positive and constructive comments on our manuscript and for recognizing the novelty and potential significance of our findings. As detailed in the point-by-point rebuttal below, we have carefully addressed each of the concerns raised by performing additional experiments, reanalysis, and providing clarifications where requested. We believe that these revisions and new data further strengthen the manuscript.

1. The title suggests that axon trafficking is a general mechanism to counteracts abnormal protein aggregation in neurons - while this may be true, the manuscript only looks at specific examples of TIA1 and the role of Annexin A7. Therefore, a more specific title would more accurately represent the manuscript.

We thank the reviewer for raising this constructive point regarding the title. We agree with the reviewer that the manuscript specifically focuses on the regulatory role of Annexin A7 in TIA1-

related RNPs. To better reflect the scope of the study, we have revised the title to: “*Annexin A7 Enhances TIA1 Axon Trafficking to Counteract Pathological Aggregation in Neurons*” in the revised manuscript.

2. In Fig.1 various n's are listed - in most cases n seems to represent the number of TIA1 granules analyzed. However, it is unclear how many neurons were analyzed and from how many independent replicates? In the graphs, what does each individual data point represent? We thank the reviewer for raising this important point. To address the concern, we have updated all figure legends, including that of Fig. 1, to clearly specify the number of neurons, mice, axons, or ROIs analyzed, the number of independent biological replicates, and what each individual data point represents. These changes are detailed in the Figure Legends section (lines 1374–end) of the revised manuscript.

Furthermore, we have included a detailed table in the Appendix listing exact *P* values for transparency and reference.

3. For timelapse imaging montages in Fig. 1H and 1I, the time intervals shown are too far apart; with the 200 and 500+ second intervals between frames shown, it is difficult to know if the granules are the same or different objects in each image. Also, the acquisition frame rates for live cell imaging experiments are not clear from the description in the Methods.

We sincerely thank the reviewer for highlighting this important point. To address these concerns, we have made several clarifications and additions:

(1) Clarification of time intervals in panels: The intervals indicated in the original Fig. 1H and 1I correspond to *key frames* selected from the full time-lapse series (acquired at 4–20 second intervals). These selected frames were intended to highlight representative motility events. We have uploaded the original videos (previously sMov. 3 and sMov. 4) with full temporal resolution to the Dryad server for reference:

http://datadryad.org/stash/share/7RtZ9VtS5_CUy09HJQ7LGhTrObd-B9w2XvpQ0TYNR2A.

To clarify this, we have now specified in the legend of Fig. 1E, F (lines 1388–1390) that the displayed panels are *key frames* extracted from time series data. Additionally, we have revised the figure legends throughout the manuscript to provide this clarification for all time-lapse panels.

2. New showcases with higher time-resolution: To further address the concern regarding the large gaps between frames in the montages, we have also included an additional time-lapse sequence with higher temporal resolution (9.9 seconds interval), as new Fig. 1E (see below) and Mov. EV3 in the revised manuscript, which provide a clearer view of TIA1 granule trafficking dynamics relative to lysosomes.

Fig. 1E

4. In lines 125-129, the authors claim: "These results suggest that the axon trafficking of TIA1 granules is unlikely to depend on membranous axon carriers, instead occurring primarily via a direct link with the retrograde motor dynein". The authors have not examined a complete list of membrane-bound organelles, including mitochondria and non-acidified endosome populations, so this statement should be softened or additional experiments should be performed with early and late endosomes, mitochondria, etc.

We thank the reviewer for raising this critical point. In response, we have both softened our conclusion and performed additional experiments to further evaluate whether TIA1 granules co-traffic with membrane-bound organelles.

(2) Low co-trafficking with early endosome and mitochondria: To further assess whether TIA1 granule transport depends on membranous axon carriers, we examined their co-trafficking with mitochondrial and early endosomal markers. In live neurons expressing EGFP-TIA1, TIA1 granules did not co-move with MitoTracker-labeled mitochondria (Fig. EV1E). Similarly, in neurons co-expressing TIA1-mCherry and EGFP-Rab5, TIA1 granules showed minimal association with Rab5-positive vesicles (Fig. EV1F), as reflected by the low co-trafficking percentages (Fig. 1F'). These data have been added to Fig. 1F' and Fig. EV1E-F, with Results section (lines 141-142) updated in the revised manuscript.

Fig. EV1E

F

Fig. 1F'

(2) Softened conclusion: While these data show consistently low co-trafficking of TIA1 granules with membrane-bound organelles compared to their higher co-trafficking with dynein, we agree that potential interactions with other compartments cannot be fully excluded. Accordingly, we have softened the conclusion in the revised manuscript (lines 152–155):

“These data demonstrate that axon transport of TIA1 granules is unlikely to depend primarily on tethering to membranous carriers. Instead, a major fraction occurs via association with the microtubule-based retrograde motor dynein.”

For further clarification, please also see our detailed response to your Question 6.

5. There is no description of the BioGRID analysis performed in the methods or anywhere in the text. Supplementary data tables S1 and S2 are described but were not provided for review. We thank the reviewer for pointing out these omissions. To address this concern, we have made the following updates:

(1) Description of the BioGRID analysis: The BioGRID analysis was performed by downloading the TIA1 and DIC1B interactor datasets from the BioGRID database (<https://thebiogrid.org>). We then identified overlapping interactors between these datasets using the “Conditional Formatting > Highlight Cells Rules > Duplicate Values” function in Excel. We have now added this description to the revised Methods section (lines 850–853).

(2) Supplementary Data Tables S1 and S2:

These tables were originally uploaded as the Source Data for Fig. 2A, 2B, and 2D during the initial submission. However, they were not incorporated into the compiled PDF by the submission system. For the revised manuscript, we have re-uploaded these files as Dataset EV1 and Dataset EV2, designated as Source Data for Fig. 2A, 2B, and 2D. Additionally, to facilitate immediate access, we have deposited them on the Dryad server, where they can be downloaded directly via the following link:

http://datadryad.org/stash/share/7RtZ9VtS5_CUy09HJQ7LGhTrObd-B9w2XvpQ0TYNR2A

6. In line 144, the authors write, "Collectively, most TIA1 granules in axons are retrogradely transported RNPs that are directly linked to dynein." However, thus far, the data presented only show co-trafficking and not a direct link, so this statement should be softened.

Furthermore, the next set of experiments with GST pull down followed by proteomic mass spec analysis shows an interaction, but again, this is not necessarily a "direct" link.

We thank the reviewer for highlighting this important point regarding the interpretation of our data. We agree that the evidence presented in Figure 1 demonstrates association and co-trafficking between TIA1 granules and dynein, rather than a direct molecular link. Consequently, we have removed the phrase “directly linked” from the conclusion. This revision has also been applied in the Results section (lines 172–175) of the revised manuscript.

“In summary, these co-trafficking data suggest that the majority of TIA1 granules in axons are retrogradely transported as membrane-less RNPs that may primarily associate with dynein rather than rely on tethering to membranous organelles, although potential interactions with other membrane-bound compartments cannot be fully excluded.”

For further clarification, please also see our detailed response to your Question 4.

7. In fig. 3, 1 mM Ca^{2+} is not a physiologic concentration, but this is a minor point since additional work is also shown in the neurons (fig. 4).

We agree with the reviewer that 1 mM Ca^{2+} is well above the physiological resting intracellular concentration in neurons, which is around 100–200 nM. However, under pathological conditions, local Ca^{2+} concentrations can transiently increase to micromolar or even millimolar levels, comparable to the extracellular environment, due to rupture of the plasma membrane. This phenomenon has been described in studies of wound-healing responses and ANXA7 regulation in non-neuronal cells (13, 14). Accordingly, *in vitro* biochemical assays investigating phase separation of Annexins often use supraphysiological Ca^{2+} concentrations to model the local Ca^{2+} overload. For example, previous studies have used 2–5 mM Ca^{2+} to study ANXA7 (14, 15) and up to 10 mM Ca^{2+} to investigate ANXA11 (2). See also our replies to R1, Q12 and R2, Q5 for more details. To clarify this rationale for using millimolar Ca^{2+} in our *in vitro* assays, we have now added the following explanation to the Discussion section (lines 610–616) in the revised manuscript:

*“Under pathological conditions, local Ca^{2+} levels can transiently rise to millimolar concentrations, similar to extracellular Ca^{2+} levels, likely due to the rupture of the plasma membrane (Sønder et al., 2019). To model the acute effects of such localized Ca^{2+} surges on the LLPS of ANXA family proteins, *in vitro* experiments commonly apply millimolar Ca^{2+} concentrations (Liao et al., 2019, Sønder et al., 2019, Yu et al., 2023). Guided by the Ca^{2+} ranges used in these studies, we examined the impact of Ca^{2+} elevations (0–10 mM) on ANXA7 phase separation.”*

Reference:

1. J. Ha *et al.*, A neuron-specific cytoplasmic dynein isoform preferentially transports TrkB signaling endosomes. *J Cell Biol* **181**, 1027-1039 (2008).
2. Y. C. Liao *et al.*, RNA Granules Hitchhike on Lysosomes for Long-Distance Transport, Using Annexin A11 as a Molecular Tether. *Cell* **179**, 147-164.e120 (2019).
3. T. Wang, F. A. Meunier, Live-Cell Superresolution Imaging of Retrograde Axonal Trafficking Using Pulse-Chase Labeling in Cultured Hippocampal Neurons. *Methods Mol Biol* **2473**, 101-128 (2022).
4. E. Doron-Mandel *et al.*, The glycine arginine-rich domain of the RNA-binding protein nucleolin regulates its subcellular localization. *Embo j* **40**, e107158 (2021).
5. E. R. Hollis, 2nd *et al.*, Ryk controls remapping of motor cortex during functional recovery after spinal cord injury. *Nature neuroscience* **19**, 697-705 (2016).
6. X. Duan, Y. Gao, Y. Liu, Ryk regulates Wnt5a repulsion of mouse corticospinal tract through modulating planar cell polarity signaling. *Cell Discov* **3**, 17015 (2017).
7. Z. Jin *et al.*, Structural basis of thymidine-rich DNA recognition by Drosophila P75 PWWP domain. *Commun Biol* **8**, 445 (2025).
8. A. Raj, P. van den Bogaard, S. A. Rifkin, A. van Oudenaarden, S. Tyagi, Imaging individual mRNA molecules using multiple singly labeled probes. *Nat Methods* **5**, 877-879 (2008).
9. X. Huang *et al.*, ROS regulated reversible protein phase separation synchronizes plant flowering. *Nature chemical biology* **17**, 549-557 (2021).
10. X. Wu *et al.*, RIM and RIM-BP Form Presynaptic Active-Zone-like Condensates via Phase Separation. *Molecular cell* **73**, 971-984.e975 (2019).
11. X. Pan *et al.*, Axons-on-a-chip for mimicking non-disruptive diffuse axonal injury underlying traumatic brain injury. *Lab on a Chip* **22**, 4541-4555 (2022).
12. X. Pan *et al.*, Actomyosin-II protects axons from degeneration induced by mild mechanical stress. *J Cell Biol* **223** (2024).
13. V. Gerke *et al.*, Annexins—a family of proteins with distinctive tastes for cell signaling and membrane dynamics. *Nature communications* **15**, 1574 (2024).
14. S. L. Sønner *et al.*, Annexin A7 is required for ESCRT III-mediated plasma membrane repair. *Scientific Reports* **9**, 6726 (2019).
15. C. Yu, S. L. Nelson, G. Meisl, R. Ghirlando, L. Deshmukh, Phase Separation and Fibrillization of Human Annexin A7 Are Mediated by Its Proline-Rich Domain. *Biochemistry* **62**, 3036-3040 (2023).

Dear Dr. Wang,

Thank you for submitting a revised version of your manuscript. Your study has now been seen by two of the three original referees, who find that their previous concerns have been addressed and now recommend publication of the manuscript with one of the referees asking for only minor textual edits which I think are fair and reasonable. The third referee was unfortunately not able to reassess the manuscript. In addition to the requested textual edits there remain only a few mainly editorial points that have to be addressed before I can extend formal acceptance of the manuscript:

- On the abstract page of the manuscript, please include 4-5 general keyword terms to enhance searchability.
- Please adjust the format of the reference list and of the in-text citations according to EMBO Journal format (alphabetical order, author name et al + year.../up to 10 author names in the reference list before et al / please refer to our Guide to Authors for additional information on EMBO J reference format).
- As we are switching from a free-text author contribution statement towards a more formal statement based on Contributor Role Taxonomy (CRediT) terms, please remove the present Author Contribution section and instead specify each author's contribution(s) directly in the Author Information page of our submission system during upload of the final manuscript. See <https://casrai.org/credit/> for more information.
- Synopsis image: Please make sure that the aspect ratio of the synopsis image conforms to our website's format - it should be exactly 550 pixels wide and between 300-600 pixels high and in either jpeg or tif format.
- Please provide the specific URL for the PRJNA1290857 dataset in the data availability statement.
- Table EV1-EV2 - legends for EV tables should be included above the tables in each Excel file
- "Materials & Correspondence" should be removed, and corr. author's email should be listed on the title page
- Section order should be corrected: Title page - Abstract - Keywords - Introduction - Results - Discussion - Methods - Data Availability - Acknowledgements - Disclosure and Competing Interests Statement - References - Figure Legends - Table(s) - Expanded View Figure Legends.
- Figure Legends (main + EV):
 1. Please note that the legends for figures 2 is not provided in the sequential manner (legend for figure D is provided before legend of figure C).
 2. Please note that the exact p values are not provided in the legends of figures 1B', E', D', F', G, H', H', 2G', H', J', J', K'; 3B', C', E', H', I', I'; 4B, C', D', E', F, G', I', J'.
 3. Please indicate the statistical test used for data analysis in the legends of figures 2B, D; 5F, EV2 A
 4. Please note that the arrow heads are not defined in the legend of figure 1E, F, H, EV1 I.

Please let me know if you have any questions regarding any of these points. You can use the link below to upload the revised files. Thank you again for giving us the chance to consider your manuscript for The EMBO Journal. I look forward to receiving the final version.

With best regards,

Cornelius Schneider

Cornelius Schneider, PhD
Editor | The EMBO Journal
c.schneider@embojournal.org

Please refer to our figure preparation guideline in order to ensure proper formatting and readability in print as well as on screen:

See also figure legend guidelines:

<https://www.embopress.org/page/journal/14602075/authorguide#figureformat>

Use the link below to submit your revision:

Referee #1:

The authors have addressed all my comments thoroughly and have added new experimental data that further clarify their results and strengthen their conclusions. Therefore, I strongly suggest publication of their manuscript.

Referee #3:

In the manuscript "Annexin A7 enhances TIA1 Axon Trafficking to Counteract Pathological Aggregation in Neurons" by Feng Y. et al., the authors identify Annexin A7 as a novel adaptor that facilitates the interaction and retrograde axonal transport of TIA1 positive RNPs with dynein intermediate chain (DIC). In the substantially revised manuscript, the authors provide additional biochemical evidence for ANXA7 as an adapter between TIA1 granules and DIC, including new data providing evidence for association of the endogenous proteins (proximity ligation assay). Furthermore, the evidence for calcium dependent regulation of this interaction is far more robust than previously, with addition of important controls. My prior concerns and critiques have been addressed.

I raised only a few additional points:

(1) In the introduction, authors write, "...it remains largely unknown whether retrograde transport of RNPs occurs within axons and, if so, how specific RNPs engage with dynein remains largely unknown." (Introduction, lines 48-49).

This reviewer would agree that the mechanisms linking specific RNPs with the dynein motor are not fully elucidated, especially in mammalian neurons. Prior work by Liao et al 2019 on Annexin A11 focuses on the linkage of RNPs to LAMP1 positive organelles and does not investigate the motors per se.

However, there are several studies that have shown both retrograde and anterograde transport of RNPs in the axon - so this part of the statement should be modified.

(2) In the revision, authors have performed experiments with addition of nocodazole and found reduced transport of TIA-1 granule transport. While this result supports dependence upon microtubules and microtubule-based motor transport, it does not exclude tethering to membranous carriers (which also use microtubule-based motors and would also be disrupted by nocodazole). The authors' interpretation of these findings should be revised (lines 153-155).

(3) Minor typographical errors are noted, including the following -

a. Line 99: "crutial"

b. Line 532: a word is missing from this sentence "...plays a direct in preventing RBP aggregation..."

Referee #3:

(1) In the introduction, authors write, "...it remains largely unknown whether retrograde transport of RNPs occurs within axons and, if so, how specific RNPs engage with dynein remains largely unknown." (Introduction, lines 48-49).

This reviewer would agree that the mechanisms linking specific RNPs with the dynein motor are not fully elucidated, especially in mammalian neurons. Prior work by Liao et al 2019 on Annexin A11 focuses on the linkage of RNPs to LAMP1 positive organelles and does not investigate the motors per se.

However, there are several studies that have shown both retrograde and anterograde transport of RNPs in the axon - so this part of the statement should be modified.

We thank the reviewer for this insightful comment, with which we fully agree. To better describe the knowledge gap, we have revised the Introduction to tone down the original statement. The updated text (lines 47-51) now reads, with modified sentences shown in bold:

*"Although motor-dependent trafficking is crucial for ensuring the correct localization and function of RNPs within neurons (Abouward & Schiavo, 2021; Abraham & Fainzilber, 2022; Dalla Costa et al, 2021; Das et al., 2021), **the mechanisms underlying dynein-mediated retrograde transport of RNPs in mammalian axons remain largely unknown.**"*

(2) In the revision, authors have performed experiments with addition of nocodazole and found reduced transport of TIA-1 granule transport. While this result supports dependence upon microtubules and microtubule-based motor transport, it does not exclude tethering to membranous carriers (which also use microtubule-based motors and would also be disrupted by nocodazole). The authors' interpretation of these findings should be revised (lines 153-155).

We thank the reviewer for this important point. We agree that nocodazole treatment alone cannot distinguish between trafficking events directly driven by dynein and those indirectly mediated through tethering to membranous carriers. To address this, we have revised the summary of this section:

- The updated text (lines 155-156) now emphasizes that nocodazole treatment together with DIC1B knockdown demonstrates the dependence of TIA1-RNP axonal trafficking on microtubule-based, dynein-driven transport.

- The conclusion that TIA1-RNPs are not exclusively dependent on membranous carriers has been moved to lines 149-151, where it now concludes the colocalization analyses of TIA1-RNPs with axonal carriers.

The revised text reads, with modified sentences shown in bold:

*“...These data suggest that retrograde trafficking of TIA1 granules in axons is likely driven by dynein, rather than relying exclusively on tethering to membranous carriers. Moreover, knockdown of endogenous DIC1B using two independent shRNA constructs (shDIC1B-1# and 2#) (Fig. EV1H) significantly impaired the retrograde trafficking of TIA1-mCherry granules in neurons (Fig. 1G, G’). Similarly, disruption of microtubule tracks with nocodazole markedly reduced the axonal transport of TIA1 granules (Fig. EV1I, I’). **Together, these results demonstrate that TIA1-granule trafficking in axons depends on the microtubule-based retrograde motor dynein.**”*

(3) Minor typographical errors are noted, including the following -

a. Line 99: "crutial"

b. Line 532: a word is missing from this sentence "...plays a direct in preventing RBP aggregation..."

We thank the reviewer for noticing these points. These minor typos have been corrected in the updated manuscript:

- “Crutial” has been corrected to “crucial” (line 100).
- The missing word “role” has been added (line 530).

Dear Dr. Wang,

I am pleased to inform you that your manuscript has been accepted for publication in the EMBO Journal.

Yours sincerely,

Cornelius Schneider, PhD
Editor
The EMBO Journal
c.schneider@embojournal.org
